# Optimal Robust Memorization with ReLU Neural Networks

**Lijia Yu**
Institute of Software, Chinese Academy of Sciences
Academy of Mathematics and Systems Science, Chinese Academy of Sciences
yulijia@ios.ac.cn

**Xiao-Shan Gao**[*]
Academy of Mathematics and Systems Science, Chinese Academy of Sciences
University of Chinese Academy of Sciences
xgao@mmrc.iss.ac.cn

**Lijun Zhang**
Institute of Software, Chinese Academy of Sciences
University of Chinese Academy of Sciences
zhanglj@ios.ac.cn

## Abstract

Memorization with neural networks is to study the expressive power of neural networks to interpolate a finite classification dataset, which is closely related to the generalizability of deep learning. However, the important problem of robust memorization has not been thoroughly studied. In this paper, several basic problems about robust memorization are solved. First, we prove that it is NP-hard to compute neural networks with certain simple structures, which are robust memorization. A network hypothesis space is called optimal robust memorization for a dataset if it can achieve robust memorization for any budget less than half the separation bound of the dataset. Second, we explicitly construct neural networks with $O(Nn)$ parameters for optimal robust memorization of any dataset with dimension $n$ and size $N$. We also give a lower bound for the width of networks to achieve optimal robust memorization. Finally, we explicitly construct neural networks with $O(Nn \log n)$ parameters for optimal robust memorization of any binary classification dataset by controlling the Lipschitz constant of the network.

## 1 Introduction

Memorization with neural networks is to study the expressive power of neural networks to interpolate a given dataset. The main focus of the study is to determine the number of parameters that a neural network needs to memorize the dataset. To be precise, for a classification dataset $\mathcal{D} = \{(x_i, y_i)\}_{i=1}^N$ with data $x_i \in \mathbb{R}^n$ and labels $y_i \in [L] = \{1, \ldots, L\}$, a network $\mathcal{F} : \mathbb{R}^d \to \mathbb{R}$ is called a *memorization* for $\mathcal{D}$ if $\mathcal{F}(x_i) = y_i$ for all $i \in [N]$. Many important advances have been made in the memorization with neural networks by Huang & Babri (1998); Yun et al. (2019); Huang (2003); Vershynin (2020); Daniely (2020); Bubeck et al. (2020); Park et al. (2021); Zhang et al. (2021); Vardi et al. (2021). Memorization has been shown to help generalization on

---

[*]Corresponding author.

complex learning tasks (Ma et al., 2018; Belkin et al., 2019; Feldman & Zhang, 2020), because data with the same label have quite diversified features and need to be nearly memorized.

However, in many fields, the required neural networks must not only be able to interpolate the dataset, but also be robust on the dataset. For example, in the field of automatic driving, the automatic driving system must be able to accurately recognize all road traffic signs, but many traffic signs may have noise on themselves, such as graffiti, stickers, and dirty, and networks that lack strong robustness will generate incorrect classification results on them, which can lead to traffic accidents. Therefore, in the field of autonomous driving, a robust network that can withstand all kinds of noise is crucial. The same is true in some other areas, such as autonomous aviation and safety monitoring systems.

Another area closely related to network security is adversarial examples (Szegedy et al., 2013; Goodfellow et al., 2014); that is, it is possible to intentionally make imperceptible modifications to a standard sample so that the network outputs a wrong label. Unfortunately, adversarial examples are hard to eliminate for commonly used networks (Azulay & Weiss, 2019; Shafahi et al., 2019; Bastounis et al., 2021; Gao et al., 2022; Yu et al., 2023). The existence of adversarial examples makes it vulnerable to use neural networks in safety-critical systems, which provides another reason to find robust networks.

So, some natural questions are raised: How many parameters does a network need to give a correct answer for a given dataset even if certain noises exist, and what is the computational complexity to find such a robust network? Strictly speaking, for a classification dataset $\mathcal{D} = \{(x_i, y_i)\}_{i=1}^N \subset \mathbb{R}^n \times [L]$ with separation bound $\lambda_{\mathcal{D}} = \min_{y_i \neq y_j} ||x_i - x_j||_\infty$ and a robust budget $\mu < \lambda_{\mathcal{D}}/2$, a neural network $\mathcal{F} : \mathbb{R}^n \to \mathbb{R}$ is called a *robust memorization* of $\mathcal{D}$ with budget $\mu$ if $\widehat{\mathcal{F}}(x) = \text{argmin}_{l \in [L]} |\mathcal{F}(x) - l| = y_i$ for all $x$ satisfying $||x - x_i||_\infty \leq \mu$. Furthermore, a hypothesis space $\mathbf{H}$ of neural networks is called an *optimal robust memorization (with respect to the robust radius)* of $\mathcal{D}$, if for any $\mu < \lambda_{\mathcal{D}}/2$ there exists an $\mathcal{F} \in \mathbf{H}$ that is a robust memorization of $\mathcal{D}$ with budget $\mu$. Notice that if $\mu \geq \lambda_{\mathcal{D}}/2$, then $\mathbb{B}(x_i, \mu) \cap \mathbb{B}(x_j, \mu) \neq \emptyset$ for some $y_i \neq y_j$, and thus there exist no networks that can robustly memorize $\mathcal{D}$ with a robust budget $\mu$, so $\mu < \lambda_{\mathcal{D}}/2$ is always assumed.

There exist significant works on robust memorization. The existence of robust memorization was proved by Yang et al. (2020); Bastounis et al. (2021), which may have an exponential number of parameters, since the universal approximation theorem was used. In a recent work (Li et al., 2022), a robust memorization was constructed, but this result limits the robust budget $< \lambda/4$ which is not optimal compared to the largest possible budget $\lambda_{\mathcal{D}}/2$. A lower bound for the number of parameters for robust interpolation was also given by (Li et al., 2022). In this paper, a systematic study of robust memorization is presented by answering the following three questions.

**Question Q1**: What is the computational complexity to decide whether a fixed structure neural network is a robust memorization of a given dataset $\mathcal{D}$? This problem is often encountered in reality because we usually train pre-designed networks, such as VGG16 (Simonyan & Zisserman, 2014) and ResNet18 (He et al., 2016).

For this question, we show that, for certain small networks, robust memorization is NP-hard.

**Theorem 1.1** (Informal). *For $\alpha \in \mathbb{R}_+$ and a classification dataset $\mathcal{D}$, it is NP-hard to decide whether there exists a network of depth 2 and width 2, which is a robust memorization of $\mathcal{D}$ with budget $\alpha$.*

This theorem shows that it is computationally difficult to find robust memorization of a given dataset using a network of certain simple structure. To our knowledge, this theorem is the first result on computational complexity for robust memorization. It has been proven that finding memorization for a given dataset using a certain small networks is NP-hard (refer to Section 2 for details). However, the NP-hardness of computing robust memorization for a non-zero budget and the NP-hardness of computing memorization cannot be deduced from each other as shown below. For a given dataset $\mathcal{D}$ and a given robust radius $\mu$, the existence of memorization of $\mathcal{D}$ does not mean the existence or absence of robust memorization of $\mathcal{D}$ with budget $\mu$; the absence of robust memorization of $\mathcal{D}$ with budget $\mu$ does not mean the existence or absence of memorization of $\mathcal{D}$.

A natural question is: Can the network structure in Theorem 1.1 be replaced by a general network structure? This problem is very challenging because it is difficult to relate the problem to NP-hard problems. It should be noted that all NP-hardness results for computing memorization are also for small networks.

**Question Q2**: As said in question Q1, for a given dataset $\mathcal{D}$ and robust radius $\mu$, it is NP-hard to find a robust memorization of $\mathcal{D}$ with budget $\mu$ within a given small network structure, so we want to know what kind of structure (can relate to $\mathcal{D}$ and $\mu$) of network can be a robust memorization of $\mathcal{D}$ with budget $\mu$.

To answer this question, we give some necessary conditions of robust memorization, and we also give a network structure that can be robust for any $\mu < \lambda_{\mathcal{D}}/2$, as following:

**Theorem 1.2** (Informal). *Let $\mathcal{D} \subset \mathbb{R}^n \times [L]$ be a dataset of size $N$ and $\mu < \lambda_{\mathcal{D}}/2$ the robust budget. A network with width smaller than $n$ cannot be robust memorization for some $\mathcal{D}$ and $\mu$. Furthermore, we can explicitly construct a network in polynomial time, which has width $3n + 1$, depth $2N + 1$, $O(Nn)$ non-zero parameters, and is a robust memorization of $\mathcal{D}$ with budget $\mu$.*

This result shows that the networks with constant width used in many works of memorization such as (Vardi et al., 2021) cannot guarantee robustness, and a network with $O(Nn)$ parameters is enough to reach robustness for any $\mathcal{D}$ and $\mu$. In (Li et al., 2022), it was shown that for a binary classification dataset $\mathcal{D}$, there exists a robust memorization network with budget $\mu < \lambda_{\mathcal{D}}/4$, which has $O(Nn \log(\frac{n}{\lambda_{\mathcal{D}}}) + N\text{polylog}(\frac{N}{\lambda_{\mathcal{D}}}))$ parameters. This result has several differences compared to Theorem 1.2. First, their robust memorization does not reach optimal robust budget, since the budget is $< \lambda_{\mathcal{D}}/4$, while our theorem does not have such a limit. Second, the number of parameters $O(Nn)$ given by us is smaller, which does not have factors $\log n$, $\log \lambda_{\mathcal{D}}$ and $\text{polylog}(\frac{N}{\lambda_{\mathcal{D}}})$. Third, our conclusions are not limited to binary classification. Also note that the number of parameters of this robust memorization does not depend on the $\lambda_{\mathcal{D}}$, $L$ and $\mu$, and these values affect the parameter value of the network.

**Question Q3**: As said in question Q2, a network with $O(Nn)$ parameters is enough to ensure robustness for any given dataset $\mathcal{D}$ and $\mu < \lambda_{\mathcal{D}}/2$. Unfortunately, when $\mu$ is very close to $\lambda_{\mathcal{D}}/2$, the value of parameters of such a network will be very large and tend toward $\infty$, which will be far beyond the scale of the computation. So, a natural question is can we find a robust networks with limited parameter value? We try to control the Lipschitz constant of the network to answer this question in the case of binary classification problem.

**Theorem 1.3** (Informal). *Let $\mathcal{D} \subset \mathbb{R}^n \times \{-1, 1\}$ be a dataset of size $N$. Then we can explicitly construct a network $\mathcal{F} : \mathbb{R}^n \to \mathbb{R}$ in polynomial time, which has width $O(n)$, depth $O(N \log(n))$, $O(Nn \log(n))$ non-zero parameters, and is a memorization of $\mathcal{D}$. Furthermore, the Lipschitz constant of $\mathcal{F}$ is optimal to guarantee robust memorization and the value of its parameters is $O(\max_{(x,y) \in \mathcal{D}} ||x||_\infty + 1/\lambda_{\mathcal{D}})$.*

This theorem requires more parameters compared to Theorem 1.2, but is more practical when the budget is close to $\lambda_{\mathcal{D}}/2$ because the values of the parameters are much smaller. Two necessary conditions for robust memorization via Lipschitz are also given.

## 2 RELATED WORK

**Memorization**. Baum (1988) showed that =networks with depth 2 and width $O(\lceil \frac{N}{n} \rceil)$ can memorize a generic binary dataset in $\mathbb{R}^n$ with size $N$. Huang & Huang (1990); Sartori & Antsaklis (1991); Bubeck et al. (2020) showed that networks with depth 2 and $O(N)$ parameters can memorize an arbitrary dataset. Huang & Babri (1998); Zhang et al. (2021); Huang (2003); Yun et al. (2019); Vershynin (2020); Hardt & Ma (2016); Nguyen & Hein (2018) further showed that networks with $O(N)$ (ignoring some small quantities) parameters can be a memorization for various networks and activation functions. Park et al. (2021) gave the first sub-linear result by showing that a network with $O(N^{2/3})$ parameters is enough for memorization under certain assumptions. Vardi et al. (2021) further gave the optimal number of parameters $\widetilde{O}(\sqrt{N})$. Since the VC dimension of

neural networks with $N$ parameters and with ReLU as activation function is $O(N^2)$ (Goldberg & Jerrum, 1993; Bartlett et al., 1998; Goldberg & Jerrum, 1993; Bartlett et al., 2019), memorizing datasets of size $N$ requires at least $\Omega(\sqrt{N})$ parameters. Bartlett et al. (2019) showed that depth helps in memorization; that is, a network with depth $L$ need $\Omega(\frac{N}{L \ln(N)})$ parameters for memorization under certain assumptions. Properties of memorization were studied in (Daniely, 2020; Patel & Sastry, 2021; Xu et al., 2021).

**Robust Memorization.** Existence of accurate and robust networks was proved based on the approximation of functions (Yang et al., 2020; Bastounis et al., 2021). In a recent work (Li et al., 2022), a robust interpolation network was constructed and a lower bound was given for the number of parameters for robust interpolation. Furthermore, it was shown that robust memorization of two disjoint infinite sets requires an exponential number of parameters.

**Computational Complexity.** The first NP-hardness result was given in (Blum & Rivest, 1992), which showed that it is NP-complete to train certain networks with three nodes and with step activation function. The computational complexities for networks with step activation functions were further studied in (Klivans & Sherstov, 2009; Shai & Shai, 2014). The computational complexities of networks with the ReLU activation function were studied (DasGupta et al., 1994; Livni et al., 2014; Arora et al., 2016; Boob et al., 2022; Froese et al., 2022). Recently, it was proven that even training a single ReLU node is NP-hard (Manurangsi & Reichman, 2018; Dey et al., 2020; Goel et al., 2020). It is worth noting that all NP-hard results for memorization with neural networks are for simple networks.

## 3 PRELIMINARIES

For $L \in \mathbb{N}_+$, denote $[L] = \{1, \ldots, L\}$. For a matrix $W$ and a vector $b$, denote by $W^{j,k}$ the element of $W$ at the $j$-th row and the $k$-th column and by $b^{(j)}$ the $j$-th element of $b$. For $\mu \in \mathbb{R}_+$ and $x \in \mathbb{R}^n$, denote $\mathbb{B}_\infty(x, \mu) = \{\overline{x} \in \mathbb{R}^n : ||\overline{x} - x||_\infty \le \mu\}$.

We consider feedforward neural networks $\mathcal{F} : \mathbb{R}^n \to \mathbb{R}$ with $D$ hidden layers and with $\sigma = \text{Relu}$ as the activation function. The $l$-th hidden layer of $\mathcal{F}$ can be written as

$$X_l = \sigma(W_l X_{l-1} + b_l) \in \mathbb{R}^{n_l}, l \in [D], \tag{1}$$

and the output is $\mathcal{F}(X_0) = X_{D+1} = W_{D+1} X_D + b_{D+1} \in \mathbb{R}^{n_{D+1}}$, where $n_0 = n$, $W_l \in \mathbb{R}^{n_l \times n_{l-1}}$, $b_l \in \mathbb{R}^{n_l}$, and $n_{D+1} = 1$. $\mathcal{F}$ is said to have depth $\text{depth}(\mathcal{F}) = D + 1$ and width $\text{width}(\mathcal{F}) = \max_{i=1}^{D+1} n_i$. Denote $\mathcal{F}_l(X_0) = X_l$ as the output of the $l$-th hidden layer of $\mathcal{F}(X_0)$ and $\mathcal{F}_l^j(X_0)$ the $j$-th element of $\mathcal{F}_l(X_0)$. The classification result of the network is $\widehat{\mathcal{F}}(x) = \text{argmin}_{l \in [L]} |\mathcal{F}(x) - l|$, that is, the label closest to $\mathcal{F}(x)$.

In this paper, we will explicitly construct networks from the following hypothesis space.

**Definition 3.1.** Denote the set of neural networks with depth $d$, width $w$, and $p$ parameters by

$$\mathbf{H}_{n,d,w,p} = \{\mathcal{F} : \mathbb{R}^n \to \mathbb{R} : \text{depth}(\mathcal{F}) = d, \text{width}(\mathcal{F}) = w, \text{para}(\mathcal{F}) = p\}, \tag{2}$$

where $\text{para}(\mathcal{F}) = p$ means that there exist two fixed sets $\mathcal{I}_w \subset \mathbb{N}^3$ and $\mathcal{I}_b \subset \mathbb{N}^2$, such that $\mathcal{I}_w \cup \mathcal{I}_b$ has $p$ elements, $W_l^{i,j} \ne 0$ for $(l, i, j) \in \mathcal{I}_w$, $b_l^{(s)} \ne 0$ for $(l, s) \in \mathcal{I}$, and all other parameters are zero. For brevity, we do not explicitly give the sets $\mathcal{I}_w$ and $\mathcal{I}_b$, when we say $\text{para}(\mathcal{F}) = p$. We use $*$ to denote an arbitrary number in $\mathbb{N}$. For example, $\mathbf{H}_{n,d,*,*} = \{\mathcal{F} : \mathbb{R}^n \to \mathbb{R} : \text{depth}(\mathcal{F}) = d\}$ is the set of networks with depth $d$, and the width and number of parameters can be any number in $\mathbb{N}_+$.

**Definition 3.2.** Let $N, n, L \in \mathbb{N}_+$, and $\mathcal{D} = \{(x_i, y_i)\}_{i=1}^N \subset \mathbb{R}^n \times [L]$ a dataset in $\mathbb{R}^n$ with size $N$ and label set $[L]$. **Denote $\mathcal{D}_{n,L,N}$ to be the set of all such datasets.** We exclude non-interesting cases by assuming $n > 9$ and $L \ge 2$. All datasets in this paper are considered to be in $\mathcal{D}_{n,L,N}$ unless otherwise mentioned. The *separation bound* for a dataset $\mathcal{D}$ is defined as

$$\lambda_{\mathcal{D}} = \min\{||x_i - x_j||_\infty : (x_i, y_i), (x_j, y_j) \in \mathcal{D} \text{ and } y_i \ne y_j\}.$$

We assume $\lambda_{\mathcal{D}} > 0$. Otherwise, $\mathcal{D}$ does not have a proper classification.

The *robust accuracy* of a network $\mathcal{F}$ on a dataset $\mathcal{D}$ with respect to a given *robust budget* $\mu \in \mathbb{R}_+$ is

$$\mathrm{RA}_{\mathcal{D}}(\mathcal{F}, \mu) = \frac{1}{|\mathcal{D}|} \sum_{(x,y) \in \mathcal{D}} \mathbf{1}(\forall \overline{x} \in \mathbb{B}_{\infty}(x, \mu), \widehat{\mathcal{F}}(\overline{x}) = y).$$

The problem of *memorization for a dataset* $\mathcal{D} \in \mathcal{D}_{n,L,N}$ is to construct a neural network $\mathcal{F} : \mathbb{R}^n \to \mathbb{R}$, such that $\mathcal{F}(x) = y, \forall (x, y) \in \mathcal{D}$. In this paper, we consider the problem of robust memorization.

**Definition 3.3.** The problem of **robust memorization for a given dataset** $\mathcal{D} \in \mathcal{D}_{n,L,N}$ **with a robust budget** $\mu$ is to build a network $\mathcal{F} : \mathbb{R}^n \to \mathbb{R}$ satisfying $\mathrm{RA}_{\mathcal{D}}(\mathcal{F}, \mu) = 1$. A network hypothesis space $\mathbf{H}$ is said to be an **optimal robust memorization** for a dataset $\mathcal{D}$, if for any $\mu < \lambda_{\mathcal{D}}/2$, there exists an $\mathcal{F} \in \mathbf{H}$ such that $\mathrm{RA}_{\mathcal{D}}(\mathcal{F}, \mu) = 1$.

## 4 OPTIMAL ROBUST MEMORIZATION

In this section, we investigate the existence and computation of robust memorization from three perspectives. First, we show that the computation of robust memorization for certain networks is NP-hard. Second, we provide some necessary conditions for the existence of optimal robust networks. Finally, we show that for any given dataset, optimal robust memorization exists and can be computed in polynomial time. Note that Theorem 1.1 follows from Theorems 4.1, and Theorem 1.2 follows from Theorems 4.3 and 4.8.

### 4.1 ROBUST MEMORIZATION FOR CERTAIN SIMPLE NETWORK IS NP-HARD

In this section, we prove that computing robust memorization with certain simple structures is NP-hard. For $\alpha \in \mathbb{R}_+$ and a dataset $\mathcal{D} \subset \mathcal{D}_{n,L,N}$, denote by $\mathrm{RobM}(\mathcal{D}, \alpha)$ the decision problem for the existence of an $\mathcal{F} \in \mathbf{H}_{n,2,2,*}$, which is a robust memorization of $\mathcal{D}$ with budget $\alpha$. We have the following result.

**Theorem 4.1.** $\mathrm{RobM}(\mathcal{D}, \alpha)$ *is NP-hard. As a consequence, it is NP-hard to compute* $\mathcal{F} \in \mathbf{H}_{n,2,2,*}$, *which is a robust memorization of* $\mathcal{D}$ *with budget* $\alpha$.

We prove the theorem by showing that $\mathrm{RobM}(\mathcal{D}, \alpha)$ is computationally equivalent to the following NPC problem. The proof details are in Appendix B.1.

**Definition 4.2.** Let $\varphi$ be a Boolean formula and $\overline{\varphi}$ the formula obtained from $\varphi$ by negating each variable. The Boolean formula $\varphi$ is called *reversible* if either both $\varphi$ and $\overline{\varphi}$ are satisfiable or both are not satisfiable. The *reversible satisfiability problem* is to recognize the satisfiability of reversible formulae in conjunctive normal form (CNF). By the *reversible 6-SAT*, we mean the reversible satisfiability problem for CNF formulae with six variables per clause. In (Megiddo, 1988), it was shown that the reversible 6-SAT is NPC.

### 4.2 NECESSARY CONDITIONS FOR THE EXISTENCE OF ROBUST NETWORKS

In this section, we give two necessary conditions for the existence of robust neural networks, which imply that robust memorization is essentially harder than memorization in that more complex networks and more parameters are needed. These results are in line with the theoretical and experimental observations that, in order to increase the robustness, the network needs more expressive power, even if the original set is linearly separable (Madry et al., 2017; Gao et al., 2019; Li et al., 2022).

The first necessary condition is in terms of the width of the network.

**Theorem 4.3.** *For any* $d < n$ *and* $N \geq n + 1$, $\mathbf{H}_{n,*,d,*}$ *is not an optimal robust memorization for* $\mathcal{D}_{n,L,N}$. *In other words, there exist a dataset* $\mathcal{D} \in \mathcal{D}_{n,L,N}$ *and a* $\mu < \lambda_{\mathcal{D}}/2$ *such that any* $\mathcal{F}$ *with width smaller than* $n$ *is not a robust memorization of* $\mathcal{D}$ *with radius* $\mu$.

**Proof Sketch**. Let $\mathcal{F} : \mathbb{R}^n \to \mathbb{R}$ be a network and let $W_1$ be the weight matrix of the first layer of $\mathcal{F}$. Then $W_1 \in \mathbb{R}^{K \times n}$. If $\mathcal{F}$ has a width smaller than $n$, then $K < n$, that is, $W_1$ is not of full-row rank. Based on this property, we can construct a dataset $\mathcal{D}$ such that $\mathcal{F}$ is not an optimal robust memorization of $\mathcal{D}$. Details of the proof are in B.2. ∎

*Remark* 4.4. It was widely observed that the width of the network plays a key role in robustness. Gao et al. (2019) showed that increasing the width of the network is necessary for robust memorization. In (Allen-Zhu et al., 2019; Du et al., 2019a;b; Li & Liang, 2018; Zou et al., 2018), it was shown that when the width is large enough, the gradient descent converges to a global minimum point. Theorem 4.3 gives more evidence of the importance of width in robustness, that is, to be a robust memorization for an arbitrary dataset, the width of the network must be at least equal to the dimension of the data.

The lower bound of width given in Theorem 4.3 is still necessary even in more general cases, as shown by the following proposition whose proof is similar to that of Theorem 4.3 and is given in Appendix B.3.

**Proposition 4.5.** *For any $\lambda > 2\mu > 0$, there exists a dataset $\mathcal{D} \in \mathcal{D}_{n,L,N}$ such that $\lambda_{\mathcal{D}} \geq \lambda$ and any network in $\mathbf{H}_{n,*,d,*}$ is not a robust memorization of $\mathcal{D}$ with budget $\mu$ if $d < n$.*

The following proposition shows that depth 2 networks are optimal robust memorization for any dataset. Furthermore, a necessary condition for depth 2 networks to be an optimal robust memorization of $\mathcal{D}_{n,L,N}$ is that the width of the network must be greater than $N$.

**Proposition 4.6.** *For neural networks with depth 2, we have the following results.*

**(1)** $\mathbf{H}_{n,2,*,*}$ *is an optimal robust memorization for $\mathcal{D}_{n,L,N}$.*

**(2)** *For any $L \geq 5$ and $N > 9$, $\mathbf{H}_{n,2,N,*}$ is not an optimal robust memorization for $\mathcal{D}_{n,L,N}$. In other words, there exist a dataset $\mathcal{D} \in \mathcal{D}_{n,L,N}$ and a $\mu < \lambda_{\mathcal{D}}/2$ such that any $\mathcal{F}$ with depth 2 and width $N$ is not a robust memorization of $\mathcal{D}$ with radius $\mu$.*

**Proof Sketch**. For a dataset $\mathcal{D} \in \mathcal{D}_{n,N,L}$ and a budget $\mu < \lambda_{\mathcal{D}}/2$, since $\mathcal{D}$ is finite, we can assume that $\mathcal{D}$ is in $[-C, C]^n$ for some $C \in \mathbb{R}$. There clearly exists a continuous function $E : [-C, C]^n \to \mathbb{R}$ such that $E(\hat{x}) = y$ for $\hat{x} \in \mathbb{B}(x, \mu)$ and $(x, y) \in \mathcal{D}$. Then (1) of Proposition 4.6 can be obtained using the universal approximation theorem (Cybenko, 1989; Leshno et al., 1993) to function $E$. The proof for (2) of Proposition 4.6 can be found in Appendix B.4. ∎

*Remark* 4.7. We give a detailed comparison with memorization networks. In (Vardi et al., 2021), it was shown that for any dataset $\mathcal{D} \in \mathcal{D}_{n,L,N}$, there exists a network $\mathcal{F} \in \mathbf{H}_{n,\widetilde{O}(\sqrt{N}),12,\widetilde{O}(\sqrt{N})}$, which can memorize $\mathcal{D}$; that is, networks with width 12 are enough for memorization. By Theorem 4.3, networks with fixed width cannot be optimal robust memorization. Here, $\widetilde{O}(\cdot)$ hides certain logarithmic factors. In (Zhang et al., 2021), it was shown that for any dataset $\mathcal{D} \in \mathcal{D}_{n,L,N}$, there exists a network $\mathcal{F} \in \mathbf{H}_{n,2,N,O(N+n)}$, which can memorize $\mathcal{D}$; that is, networks with depth 2 width $N$ are enough for memorization. By (2) of Proposition 4.6, networks with depth 2 width $N$ cannot be optimal robust memorization. In summary, robust memorization is essentially harder than memorization.

### 4.3 OPTIMAL ROBUST MEMORIZATION WITH NEURAL NETWORK

The following theorem gives an optimal robust memorization for any given dataset. Notice that the network is constructed explicitly.

**Theorem 4.8.** *The hypothesis space $\mathbf{H}_{n,2N+1,3n+1,O(Nn)}$ is an optimal robust memorization for $\mathcal{D}_{n,L,N}$. More precisely, for any $\mathcal{D} \in \mathcal{D}_{n,L,N}$ and $\mu < \lambda_{\mathcal{D}}/2$, there exists a robust memorization network $\mathcal{F}$ of $\mathcal{D}$ with budget $\mu$ such that $\mathrm{width}(\mathcal{F}) = 3n + 1$, $\mathrm{depth}(\mathcal{F}) = 2N + 1$, $\mathrm{para}(\mathcal{F}) = (N - 1)(12n + 5) + 2$, and the absolute values of the parameters are $O(\max_{(x,y)\in\mathcal{D}} ||x||_{\infty} + \frac{L}{\lambda_{\mathcal{D}}-2\mu})$.*

**Proof sketch.** Let $\mathcal{D} = \{(x_i, y_i)\}_{i=1}^N \subset \mathbb{R}^n \times [L]$. It suffices to show that for any $\mu < \lambda_{\mathcal{D}}/2$, there exists a network $\mathcal{F} : \mathbb{R}^n \to \mathbb{R}$ with depth $2N+1$, width $3n+1$, and $O(Nn)$ parameters, which can robustly memorize $\mathcal{D}$ with robust budget $\mu$. From equation 1, a neural network $\mathcal{F} : \mathbb{R}^n \to \mathbb{R}$ can be constructed as follows: starting from the input $x \in \mathbb{R}^n$, performing affine transformations over $\mathbb{R}$ and taking the activation function $\sigma$ recursively. In what follows, we will give a sketch of the construction of $\mathcal{F}$. The details of the proof are given in Appendix B.5.

Let $x \in \mathbb{R}^n$ be the input and $C \in \mathbb{R}_+$ satisfy $C > |x_i^{(j)}| + \mu > 0$ for all $i \in [N]$ and $j \in [n]$. First, we need to know $x$ in each layer of the network, which can be achieved with neural networks by the fact $\sigma(x^{(j)} + C) = x^{(j)} + C$ for $j \in [n]$, $x \in \mathbb{B}_\infty(x_i, \mu)$, and $i \in [N]$.

Second, for each $i \in [N]$, we construct a neural network $E_i(x) : \mathbb{R}^n \to \mathbb{R}$:

$$E_i(x) = y_i - \frac{y_i}{\lambda_{\mathcal{D}} - 2\mu} \sum_{j=1}^n (\sigma(x_i^{(j)} - x^{(j)} - \mu) + \sigma(x^{(j)} - x_i^{(j)} - \mu))$$

which satisfies the following properties: (1) $E_i(x) = y_i$ for $x \in \mathbb{B}_\infty(x_i, \mu)$; (2) $E_i(x) < y_i$ for $x \notin \mathbb{B}_\infty(x_i, \mu)$; (3) $E_i(x) \le 0$ for $x \in \mathbb{B}_\infty(x_k, \mu)$ and $y_k \neq y_i$.

Finally, the output is $\mathcal{F}(x) = \max_{i \in [N]}\{E_i(x), 0\}$, which can be achieved with neural networks by the property: $\max\{y, z\} = y + \sigma(z - y)$ for $y, z \in \mathbb{R}$.

Now we show that $\mathcal{F}$ is the required network. Let $x \in \mathbb{B}_\infty(x_s, \mu)$ for $s \in [N]$. Then $E_s(x) = y_s$. For $i \neq s$, there exist two cases: if $y_i = y_s$, then $E_i(x) < y_s$; if $y_i \neq y_s$, then $E_i(x) \le 0$. In summary, $\mathcal{F}(x) = \max_{i \in [N]}\{E_i(x), 0\} = E_s(x) = y_s$, that is, $\mathcal{F}$ is robust at $x_s$ with budget $\mu$. ∎

*Remark* 4.9. Note that the network constructed in the proof of Theorem 4.8 satisfies $\mathcal{F}(\overline{x}) = y_i$ for all $\overline{x} \in \mathbb{B}_\infty(x_i, \mu)$ and $i \in [N]$, which is a special type of robust memorization. The following proposition shows that these two types of robust memorization essentially need the same number of parameters. More details on the number of parameters and the proof of the proposition are given in Appendix B.6.

**Proposition 4.10.** *Let $\mathcal{D} = \{(x_i, y_i)\}_{i=1}^N \in \mathcal{D}_{n,L,N}$. If the network $\mathcal{F}$ is a robust memorization of $\mathcal{D}$ with budget $\mu$, then there exists a network $G : \mathbb{R} \to \mathbb{R}$ with $\mathrm{depth}(\mathcal{G}) = 2$, $\mathrm{width}(\mathcal{G}) = 2L$, and $\mathrm{para}(\mathcal{G}) = 6L$, such that $\mathcal{F}_1 = \mathcal{G}(\sigma(\mathcal{F}))$ satisfies $\mathcal{F}_1(\overline{x}) = y_i$ for all $\overline{x} \in \mathbb{B}_\infty(x_i, \mu)$ and $i \in [N]$.*

*Remark* 4.11. In this paper, we focus on robust memorization under the norm $L_\infty$, because adversarial attacks with the norm $L_\infty$ are the most widely used attacking methods in the image classification problem. Theorem 4.8 can be generalized to any norm $p \ge 1$; see Appendix B.7.

## 5  OPTIMAL ROBUST MEMORIZATION VIA LIPSCHITZ

In this section, we construct optimal memorization networks by controlling the Lipschitz constant of the network. Theorem 1.3 follows from Theorem 5.2.

The $L_\infty$ norm Lipschitz constant for $\mathcal{F} : \mathbb{R}^n \to \mathbb{R}$ is defined as

$$\mathrm{Lip}_\infty(\mathcal{F}) = \max_{x, \tilde{x} \in \mathbb{R}^n; \, x \neq \tilde{x}} \left\{ \frac{|\mathcal{F}(x) - \mathcal{F}(\tilde{x})|}{||x - \tilde{x}||_\infty} \right\}.$$

In this section, we consider binary classification problems and the dataset has the following form

$$\mathcal{D} = \{(x_i, y_i)\}_{i=1}^N \subset \mathbb{R}^n \times \{-1, 1\}. \tag{3}$$

**Denote $\mathcal{B}_{n,N}$ to be the set of all $\mathcal{D}$ of the form in equation 3.** In order to achieve robust memorization by controlling the Lipschitz constant of $\mathcal{F}$, the classification result of $\mathcal{F}$ is defined as $\widehat{\mathcal{F}}(x) = \mathrm{Sgn}(\mathcal{F}(x))$, which is the commonly used setting for binary classification data in equation 3.

Because there exist $(x_i, 1), (x_j, -1) \in \mathcal{D}$ such that $||x_i - x_j||_\infty = \lambda_\mathcal{D}$, so if $\mathcal{F}$ memorizes $\mathcal{D}$, then $\text{Lip}_\infty(\mathcal{F}) \geq |\mathcal{F}(x_i) - \mathcal{F}(x_j)|/||x_i - x_j||_\infty = \frac{2}{\lambda_\mathcal{D}}$, which motivates the following definition.

**Definition 5.1.** A network $\mathcal{F}$ is called a *robust memorization of a dataset $\mathcal{D}$ via Lipschitz with budget* $\mu < \lambda_\mathcal{D}/2$, if $\mathcal{F}$ is a memorization of $\mathcal{D}$ and $\text{Lip}_\infty(\mathcal{F}) \leq 1/\mu$. Furthermore, $\mathcal{F}$ is called an *optimal robust memorization of $\mathcal{D}$ via Lipschitz*, if $\mathcal{F}$ is a memorization of $\mathcal{D}$ and $\text{Lip}_\infty(\mathcal{F}) = 2/\lambda_\mathcal{D}$.

## 5.1 Optimal Robust Memorization via Lipschitz

The following theorem gives an optimal robust network via Lipschitz for any binary classification dataset.

**Theorem 5.2.** *For any dataset $\mathcal{D} \in \mathcal{B}_{n,N}$, the hypothesis space $\mathbf{H}_{n,O(N\log(n)),O(n),O(Nn\log(n))}$ contains a network $\mathcal{F}$ that is an optimal robust memorization of $\mathcal{D}$ via Lipschitz, and the values of parameters of $\mathcal{F}$ are $O(\max_{(x,y)\in\mathcal{D}} ||x||_\infty + \frac{1}{\lambda_\mathcal{D}})$.*

**Proof Sketch.** Let $\mathcal{D}$ be defined in equation 3. Let $x \in \mathbb{R}^n$ be the input and $C \in \mathbb{R}_+$ satisfy $C > |x_i^{(j)}| + 0.5\lambda_\mathcal{D} > 0$ for all $i \in [N]$ and $j \in [n]$. The construction of the network $\mathcal{F}$ has three main gradients.

First, we need to know $x$ in each layer of the network, which can be achieved with neural networks by the property $\sigma(x^{(j)} + C) = x^{(j)} + C$ for $j \in [n]$, $x \in \mathbb{B}_\infty(x_i, \mu)$, and $i \in [N]$.

Second, for each $k \in [N]$, we construct a neural network $E_k(x) = ||x - x_k||_\infty$, which can be achieved by the properties $\sigma(x) + \sigma(-x) = |x|$ and $\max\{x, y\} = x + \sigma(y - x)$ for $x, y \in \mathbb{R}$, since $||z||_\infty = \max_{i=1}^n |z^{(i)}|$.

Finally, the network is $\mathcal{F}(x) = y_{w_N} \sigma(1 - \frac{2}{\lambda_\mathcal{D}} ||x - x_{w_N}||_\infty)$, where $w_N = \text{argmin}_{i\in[N]} ||x - x_i||_\infty$. $w_N$ can be computed by the property $\min\{x, y\} = x - \sigma(x - y)$ for $x, y \in \mathbb{R}$.

We now show that $\mathcal{F}$ satisfies the condition of the theorem, that is, $\mathcal{F}$ a memorization of $\mathcal{D}$ and satisfies $\text{Lip}_\infty(\mathcal{F}) = \frac{2}{\lambda_\mathcal{D}}$. If $x = x_k$, then $w_N = \text{argmin}_{i\in[N]} ||x - x_i||_\infty = k$. So, $\mathcal{F}(x_k) = y_k$, that is, $\mathcal{F}$ is a memorization of $\mathcal{D}$. If $x \in \mathbb{B}(x_k, 0.5\lambda_\mathcal{D})$ for some $k \in [N]$, then $w_N = k$ and $\mathcal{F}(x) = y_k(1 - \frac{2}{\lambda_\mathcal{D}} ||x - x_k||_\infty)$, so $\text{Lip}_\infty(\mathcal{F}) = \frac{2}{\lambda_\mathcal{D}}$ over $\mathbb{B}(x_k, 0.5\lambda_\mathcal{D})$, since $y_i \in \{-1, 1\}$. If $x$ is not in $B = \cup_{i=1}^N \mathbb{B}(x_i, 0.5\lambda_\mathcal{D})$, then $||x - x_{w_N}||_\infty > 0.5\lambda_\mathcal{D}$ and hence $\mathcal{F}(x) = 0$. As a consequence, we have $\text{Lip}_\infty(\mathcal{F}) = \frac{2}{\lambda_\mathcal{D}}$. Thus, $\mathcal{F}$ is an optimal robust memorization of $\mathcal{D}$ via Lipschitz. Details of the proof are given in Appendix C.2. ∎

*Remark* 5.3. We compare robust memorization with robust memorization via Lipschitz. By Theorem 4.8, it requires $O(Nn)$ parameters for optimal robust memorization. By Theorem 5.2, optimal robust memorization via Lipschitz needs $O(Nn\log n)$ parameters. Therefore, to achieve optimal robust memorization via Lipschitz, more parameters are required according to these results. However, robust memorization via Lipschitz also has advantages. First, there indeed exists a network that is an optimal robust memorization via Lipschitz, while for robust memorization according to Definition 3.3, optimal robust memorization exists for a hypothesis space of networks. Second, robust networks via Lipschitz have controlled parameter values.

## 5.2 Necessary Conditions for Robust Memorization via Lipschitz

It is easy to see that robust memorization via Lipschitz implies robust memorization, so the necessary conditions given in Section 4.2 are also valid for robust memorization via Lipschitz. In this section, we give more necessary conditions for robust memorization via Lipschitz. In the following proposition, we show that a necessary condition for $\mathcal{F}$ to be an optimal robust memorization via Lipschitz is that $\text{depth}(\mathcal{F}) > 2$, whose proof is given in Appendix C.3.

**Proposition 5.4.** *Assume $N \geq n + 1$. There exists a dataset $\mathcal{T} \in \mathcal{B}_{n,N}$, such that any network $\mathcal{F}$ with depth 2 is not an optimal robust memorization for $\mathcal{T}$ via Lipschitz.*

By (1) of Theorem 4.6, networks of depth 2 is an optimal robust memorization for any dataset, so Proposition 5.4 implies that robust memorization via Lipschitz is harder than robust memorization. The following proposition shows that this fact is also true in a more relaxed case which does not reach optimal robustness. The proof of the proposition is given in Appendix C.4.

**Proposition 5.5.** *Let* $\mathbf{H} = \{\mathcal{F} : \mathbb{R}^n \to \mathbb{R} : \mathrm{depth}(\mathcal{F}) = 2, \mathrm{width}(\mathcal{F}) = 4n[\frac{N}{n+1}]\}$ *and assume* $N \geq n + 1$. *Then there exists a dataset* $\mathcal{D} \in \mathcal{B}_{n,N}$ *such that the following results hold.*

**(1)** $\mathbf{H}$ *is an optimal robust memorization for* $\mathcal{D}$; *that is, for any* $\mu < \lambda_{\mathcal{D}}/2$, *there exists a network* $\mathcal{F} \in \mathbf{H}$, *which is a robust memorization for* $\mathcal{D}$ *with budget* $\mu$.

**(2)** *There exists a* $\mu < \lambda_{\mathcal{D}}/2$ *such that any* $\mathcal{F} \in \mathbf{H}$ *is not a robust memorization for* $\mathcal{D}$ *via Lipschitz with budget* $\mu$; *that is, if* $\mathcal{F}$ *is a memorization for* $\mathcal{D}$ *then* $\mathrm{Lip}_\infty(\mathcal{F}) > 1/\mu$.

Note that (2) of Proposition 5.5 cannot be deduced from Proposition 5.4, because $\mathcal{D}$ in Proposition 5.5 must satisfy both (1) and (2), and the robust budgets for Propositions 5.4 and 5.5 are $\lambda_{\mathcal{D}}/2$ and $\mu$, respectively.

# 6 CONCLUSION

This paper extends previous work on memorization with neural networks to robust memorization. We study robust memorization in three aspects. First, we prove that for depth 2 and width 2 neural networks, it is NP-hard to compute robust memorization. Second, we give explicit construction of optimal robust memorization networks for a given dataset in polynomial time and several necessary conditions for the existence of robust memorization. Third, we give explicit construction of optimal robust memorization networks for any binary classification dataset by controlling the Lipschitz constant of the network.

**Problems for further study.** In this paper, we consider the $L_\infty$ norm due to the fact that the $L_\infty$ norm is the most widely used norm in adversarial attacks for image classification. It is interesting to consider robust memorization for other norms. We already have partial results on this problem. In Appendix B.7, we extend Theorem 4.8 to the case of $L_p$ norm with $p \geq 1$.

In (Li et al., 2022), a lower bound $\Omega(\sqrt{Nn})$ is given for the number of parameters of a robust memorization network. There exists a gap between the current best result $O(Nn)$ of Theorem 4.8 and the lower bound. An open problem is to find the optimal number of parameters needed for optimal robust memorization.

In this paper, we consider the expressive power of the network for robust memorization, and generalizability is not studied. In (Li et al., 2022), it was shown that an exponential number of parameters is needed to robustly memorize a manifold, which was closely related with generalizability. However, the result is not in the standard form of generalization bound, which gives an upper bound for the difference between the loss over the whole data distribution and the loss over a finite training set. The generalizability of memorization networks needs to be studied.

Theorem 1.3 is for binary classification problem, it seems a challenge problem to extend such a result to multiple classification problems.

**Acknowledgments**. This work is supported by CAS Project for Young Scientists in Basic Research, Grant No.YSBR-040, ISCAS New Cultivation Project ISCAS-PYFX-202201, and ISCAS Basic Research ISCAS-JCZD-202302. This work is partially supported by the NSFC grant No.11688101 and NKRDP grant No.2018YFA0306702. The authors thank anonymous referees for their valuable comments.

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

## A   PRELIMINARY RESULTS ABOUT LINEAR REGIONS OF DEPTH 2 NETWORK

In the proofs for Sections 4 and 5, we need results on the linear regions of neural networks with depth 2, which will be given in this section. Note that $\sigma$ is ReLU.

Let $\mathcal{F} : \mathbb{R}^n \to \mathbb{R}$ be a network with depth 2 and width $k$. Since $\sigma(-x) = \sigma(x) - x$ and $\sigma(ax) = a\sigma(x)$ for $a \in \mathbb{R}_+$, $\mathcal{F}(x)$ can be written as the following *normal form*

$$\mathcal{F}(x) = \sum_{i=1}^{k} a_i \sigma(U_i x + b_i) + Q x + b \tag{4}$$

where $a_i \in \{-1, 1\}, b_i \in \mathbb{R}, U_i \in \mathbb{R}^{1 \times n}$ and $U_i \neq 0, Q \in \mathbb{R}^{1 \times n}$, and there exist no $i \neq j$ such that $U_i = q U_j$ and $b_i = q b_j$ for $q \in \mathbb{R}$.

For a network $\mathcal{F}$ in normal-form equation 4, we define the concept of linear region. Let $H_i^+ = \{x \in \mathbb{R}^n : U_i x + b_i > 0\}$ and $H_i^- = \{x \in \mathbb{R}^n : U_i x + b_i < 0\}$. If

$$R = \overline{\cap_{i=1}^{k} H_i^{s_i}} \neq \emptyset,$$

then $R$ is called a *linear region* of $\mathcal{F}$, where $s_i \in \{+, -\}$. The *interior* of a linear region $R$ is the set of points $x \in R$ such that $U_i x + b_i \neq 0$ for all $i \in [k]$. An *edge* of $R$ is the set of points $x \in R$ such that $U_i x + b_i = 0$ for some $i$.

**Definition A.1.** Let $R$ be a linear region. Define $S_i(R) = 1$ if there exists an interior point $x$ of $R$ such that $U_i x + b_i > 0$; and $S_i(R) = -1$ in the opposite case. Two linear regions $R_1$ and $R_2$ are said to be *neighboring linear regions*, if there exists only one $s$ such that $S_s(R_1)S_s(R_2) = -1$. For two neighboring linear regions $R_1$ and $R_2$, the *boundary* of $R_1$ and $R_2$ is $R_1 \cap R_2 \cap \{x : U_s x + b_s = 0\}$, which is said to be defined by $(U_s, b_s)$.

We give several properties for linear regions that will be used in this paper.

**Lemma A.2** (Montufar et al. (2014)). *Let $R$ be a linear region of $\mathcal{F}$.*

**(1)** *Over $R$, $\mathcal{F}(x) = (\sum_{i \in S} a_i U_i + Q)x + \sum_{i \in S} a_i b_i = W_R x + b_R$ is an linear function, where $S = \{i \in [k] : S_i(R) = 1\}, W_R \in \mathbb{R}^{1 \times n}, b_R \in \mathbb{R}$.*

**(2)** *$R$ is a closed and $n$-dimensional convex polyhedron. The set of interior points of $R$ is an open set of dimension $n$. An edge of $R$ is of dimension $\leq n - 1$.*

**(3)** *$\mathbb{R}^n$ is the union of all linear regions.*

**(4)** *For two linear regions $R_1$ and $R_2$, $R_1 \cap R_2$ has dimension $\leq n - 1$ and the sets of interior points of $R_1$ and $R_2$ are disjoint. The boundary of two neighboring linear regions has dimension $n - 1$.*

**Lemma A.3.** *Let $x \in \mathbb{R}^n$ satisfy $U_s x + b_s = 0$ for some $s \in [k]$. Then for any $\epsilon > 0$, there exist two linear regions $R_1$, $R_2$, and two points $x_1$ and $x_2$ satisfying*

**(1)** *$||x_1 - x||_\infty \leq \epsilon$ and $||x_2 - x||_\infty \leq \epsilon$;*

**(2)** *$x_i$ is in the interior of $R_i$ for $i = 1, 2$;*

**(3)** *$R_1$ and $R_2$ are neighboring linear regions and their boundary is defined by $(U_s, b_s)$.*

*Proof.* Let $H_i$ be the hyperplane defined by $U_i x + b_i = 0$. Consider two cases.

**Case 1.** If $x$ satisfies that $U_j x + b_j = 0$ if and only if $j = s$; that is $s$ is the only element of $[k]$ such that $U_s x + b_s = 0$. Let $\eta = \min_{j \in [k]/\{s\}} |U_j x + b_j|$ and

$x_1 = x + \min\{\epsilon/2, \frac{\eta}{2\max_{j\in[k]}||U_j||_2}\}\frac{U_s^\tau}{||U_s||_2}$;

$x_2 = x - \min\{\epsilon/2, \frac{\eta}{2\max_{j\in[k]}||U_j||_2}\}\frac{U_s^\tau}{||U_s||_2}$.

For any $k = \{1, 2\}$, $||x_k - x||_\infty \leq ||x_k - x||_2 \leq \epsilon/2$, so (1) is proved.

Note that $U_s x_1 + b_s = \min\{\epsilon/2, \frac{\eta}{2\max_{j\in[k]}||U_j||_2}\} > 0$. Similarly, we can show $U_s x_2 + b_s < 0$. Thus, $x_1$ and $x_2$ are on the opposite sides of $H_s$. Furthermore, for $j \neq s$ and $x_k \in \{1, 2\}$, we have

$$
\begin{aligned}
& |U_j x_k + b_j| \\
\geq\ & |U_j x + b_j| - |\min\{\epsilon/2, \tfrac{\eta}{2\max_{j\in[k]}||U_j||_2}\}\tfrac{1}{||U_s||_2}U_j \cdot U_s^\tau| \\
\geq\ & \eta - \tfrac{\eta}{2\max_{j\in[k]}||U_j||_2||U_s||_2}U_j \cdot U_s^\tau \\
\geq\ & \eta/2.
\end{aligned}
$$

Therefore, $x_k$ is an interior point of a linear region $R_k$. By Lemma A.2, (2) is proved.

Since for any $j \neq s$ and $k \in \{1, 2\}$, $|U_j x + b_j - U_j x_k - b_j| \leq \eta/2 < |U_j x + b_j|$, we have $U_j x_1 + b_j$ and $U_j x_2 + b_j$ have the same sign with $U_j x + b_j$, so (3) is proved.

**Case 2.** $U_j x + b_j = 0$ for more than one $j \in [k]$. We claim that there exists an $\tilde{x}$ such that $||x - \tilde{x}||_\infty < \epsilon/2$ and $U_j \tilde{x} + b_j = 0$ implies $j = s$.

Let $S = \{j \in [k] : U_j x + b_j = 0\}$. Since for $j \in S$, $H_j$ are hyperplanes passing $x$, any two of them cannot be parallel, or equivalently, $U_j \neq q U_s$ for any $q \in \mathbb{R}$ and $j \in S$, $j \neq s$. Let $V_j^\perp = \{v \in \mathbb{R}^n : U_j v = 0\}$ for $j \in [k]$. Let $v \in V_s^\perp \setminus (\cup_{j\in S, j\neq s} V_j^\perp)$, and let $\eta = \min_{j\in[k]\setminus S} |U_j x + b_j|$.

Define $t_v = \min\{\epsilon/2, \frac{\eta}{2\max_{j\in[k]}||U_j||_2}\}\frac{1}{||v||_2} > 0$ and $\tilde{x} = x + t_v v$. Then $||\tilde{x} - x||_\infty \leq ||\tilde{x} - x||_2 = ||t_v v||_2 \leq \epsilon/2$. We have $U_s \tilde{x} + b_s = 0$ and $U_j \tilde{x} + b_j = U_j t_v v + U_j x + b_j = U_j t_v v \neq 0$ when $j \in S/\{s\}$. $|U_j \tilde{x} + b_j| = |U_j t_v v + U_j x + b_j| \geq \eta - \frac{\eta}{2\max_{j\in[k]}||U_j||_2||v||_2}U_j v \geq \eta/2$ when $j \in [k] \setminus S$. So $\tilde{x} \in H_j$ if and only if $j = s$. Now, combining Case 1 for $\tilde{x}$, we can construct the required $x_1$ and $x_2$. The lemma is proved. $\square$

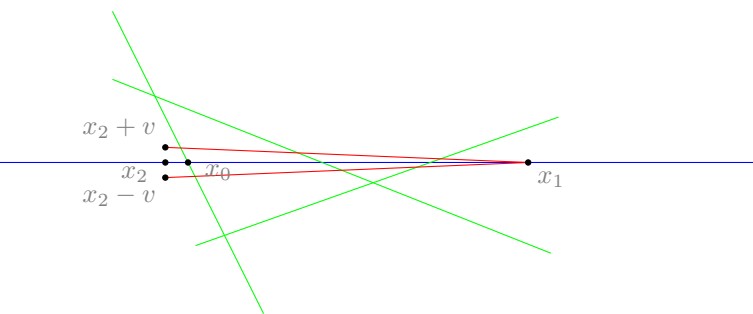

Figure 1: An illustration for Lemma A.4. The blue line is $H_s : U_s x + b_s$. The green lines are $H_j : U_j x + b_j$ ($j \neq s$). The red segments are $\lambda x_1 + (1 - \lambda)(x_2 + v)$ and $\lambda x_1 + (1 - \lambda)(x_2 - v)$.

**Lemma A.4.** *Let $x_1, x_0 \in \mathbb{R}^n$ satisfy $U_s x_1 + b_s = 0$ and $U_s x_0 + b_s = 0$ for some $s \in [k]$. Then for any $\epsilon > 0$ and $\eta > 0$, there exist $x_2, v \in \mathbb{R}^n$ such that*

**(1)** $||x_2 - x_0||_\infty \leq \epsilon$, $||v||_\infty \leq \eta$;

**(2)** $U_j x_2 + b_j = 0$ *if and only if $j = s$;*

**(3)** *Let $p_\lambda = \lambda x_1 + (1 - \lambda)(x_2 + v)$ and $q_\lambda = \lambda x_1 + (1 - \lambda)(x_2 - v)$, where $\lambda \in [0, 1]$. Let $I \subset [0, 1]$ be the set of $\lambda$, such that $p_\lambda$ is in the interior of a linear region $R_{1,\lambda}$, $q_\lambda$ is in the interior of a linear region $R_{2,\lambda}$, and $R_{1,\lambda}$ and $R_{2,\lambda}$ are neighboring linear regions with boundary $(U_s, b_s)$. Then $I$ is a set of intervals with total length $|I| \geq 0.5$.*

*Proof.* Since $H_j = \{x : U_j x + b_j = 0\}$ is an $(n-1)$-dimensional hyperplane in $\mathbb{R}^n$ and $H_i \neq H_j$ for $i \neq j$, $H_i \cap H_j$ is of dimension at most $(n-2)$. Let $x_2$ be a point in $\mathbb{B}_\infty(x_0, \epsilon) \cap H_s \setminus (\cup_{j \neq s} H_j)$. Then $||x_2 - x_0||_\infty \leq \epsilon$ and $U_j x_2 + b_j = 0$ if and only if $j = s$.

For $j \neq s$, since $x_2 \notin H_j$, there exists at most one $\lambda \in \mathbb{R}$ such that $\lambda x_1 + (1 - \lambda) x_2 \in H_j$ and let $\lambda_j$ be such a value if it exists.

Let $\Gamma = [0, 1] \setminus ((1 - \frac{1}{4k}, 1] \cup (\cup_{j \neq s}(\lambda_j - \frac{1}{4k}, \lambda_j + \frac{1}{4k})))$. Then $\Gamma \subset [0, 1)$ is a closed set with length at least $1 - k\frac{1}{2k} = 0.5$.

Let $W = \max_{i \in [k]} ||U_i||_2$, $x_\lambda = \lambda x_1 + (1 - \lambda) x_2$, $\theta_j = \min_{\lambda \in \Gamma}\{|U_j x_\lambda + b_j|\}$, and $\theta = \min_{j \neq s}\{\theta_j\}$. Since $\Gamma$ is a closed set and $\lambda_j \notin \Gamma$, we have $\theta > 0$.

Let $v_0 \in \mathbb{R}^{1 \times n}$ satisfy $v_0 U_j \neq 0$ for all $j \in [k]$ and $v = \min\{\eta, \frac{\theta}{2W}\}\frac{v_0}{||v_0||_2}$. It is easy to see that $||v||_\infty \leq \eta$. Then for all $\lambda \in \Gamma$, we have that

(a1) $x_\lambda + (1 - \lambda)v = p_\lambda$ and $x_\lambda - (1 - \lambda)v = q_\lambda$.

(a2) $U_s(x_\lambda + (1 - \lambda)v) + b_s$ and $U_s(x_\lambda - (1 - \lambda)v) + b_s$ have different signs for all $\lambda \in \Gamma$, because $U_s x_\lambda + b_s = 0$ and $U_s v \neq 0$.

(a3) $|U_j v| \leq \frac{\theta U_j v}{2W||v_0||_2} \leq \theta/2 \leq |U_j x_\lambda + b_j|$ for all $j \neq s$ and $\lambda \in \Gamma$, which means that $U_j(x_\lambda + (1-\lambda)v) + b_j$ and $U_j(x_\lambda - (1 - \lambda)v) + b_j$ have the same sign.

(a2) and (a3) show that if $\lambda \in \Gamma$, then $R_{1,\lambda}$ and $R_{2,\lambda}$ are neighboring linear regions with boundary $(U_s, b_s)$. So $x_2$ and $v$ satisfy the properties in the lemma, and $\Gamma \subset I$ has length at least 0.5. $\qquad \square$

# B  PROOFS FOR SECTION 4

## B.1  PROOF OF THEOREM 4.1

Denote $\mathcal{H}_{n,2} = \mathbf{H}_{n,2,2,*}$. Then an $\mathcal{F}(x) \in \mathcal{H}_{n,2}$ can be written as

$$\mathcal{F} = s_1\sigma(U_1 x + b_1) + s_2\sigma(U_2 x + b_2) + c \tag{5}$$

where $s_i \in \{-1, 1\}$, $U_i \in \mathbb{R}^{1 \times n}$, $b_i \in \mathbb{R}$, $c \in \mathbb{R}$.

For $i \in [n]$, let $\mathbf{1}_i \in \mathbb{R}^n$, whose $i$-th entry is 1 and all other entries are 0.

Denote $\widetilde{\mathcal{F}}(x) = \psi(\mathcal{F}(x))$, where $\psi$ is the sign function. First, we prove a lemma.

**Lemma B.1.** *Let $\mathcal{F} \in \mathcal{H}_{n,2}$ be of the form equation 5 and $z_i = i\mathbf{1}_1 \in \mathbb{R}^n$. If $\widetilde{\mathcal{F}}(z_1) = \widetilde{\mathcal{F}}(z_{-1}) = -1$ and $\widetilde{\mathcal{F}}(z_2) = \widetilde{\mathcal{F}}(z_{-2}) = 1$, then $s_1 s_2 < 0$ when $c \geq 0$ and $s_1 s_2 > 0$ when $c < 0$.*

*Proof.* Let $\mathcal{F}$ be of the form equation 5. We just need to prove the lemma for $n = 1$. Consider two cases.

**(c1):** Assume $c \geq 0$. Since $-1 = \widetilde{\mathcal{F}}(z_1) = \psi(s_1\sigma(U_1 z_1 + b_1) + s_2\sigma(U_2 z_1 + b_2) + c) \geq \psi(s_1\sigma(U_1 z_1 + b_1) + s_2\sigma(U_2 z_1 + b_2))$, at least one of $s_1 = -1$ or $s_2 = -1$ holds. Assume that $s_2 = -1$. We will show that

$s_1 = 1$. If this is not true, then $s_1 = s_2 = -1$. Since $-\sigma(a)/4 - 3/4\sigma(b) \leq -\sigma(a/4 + 3b/4)$, we have

$$
\begin{aligned}
& 0 \\
< \ & \tfrac{1}{4}\mathcal{F}(z_2) + \tfrac{3}{4}\mathcal{F}(z_{-2}) \\
= \ & (-\sigma(U_1 z_2 + b_1) - \sigma(U_2 z_2 + b_2) + c)/4 + 3(-\sigma(U_1 z_{-2} + b_1) - \sigma(U_2 z_{-2} + b_2) + c)/4 \\
= \ & (-\sigma(U_1 z_2 + b_1) - 3\sigma(U_1 z_{-2} + b_1))/4 + (-\sigma(U_2 z_2 + b_2) - 3\sigma(U_2 z_{-2} + b_2))/4 + c \\
\leq \ & -\sigma(U_1 z_{-1} + b_1) - \sigma(U_2 z_{-1} + b_2) + c \\
= \ & \mathcal{F}(z_{-1}) < 0,
\end{aligned}
$$

which is a contradiction.

**(c2):** Assume $c < 0$. Since $1 = \widetilde{\mathcal{F}}(z_2) = \psi(s_1\sigma(U_1 z_2 + b_1) + s_2\sigma(U_2 z_2 + b_2) + c) \leq \psi(s_1\sigma(U_1 z_2 + b_1) + s_2\sigma(U_2 z_2 + b_2))$, at least one of $s_1 = 1$ or $s_2 = 1$ holds. Assume $s_1 = 1$. We will show that $s_2 = 1$. If this is not true, then $s_2 = -1$. Then

$$1 = \widetilde{\mathcal{F}}(z_2) = \psi(\sigma(U_1 z_2 + b_1) - \sigma(U_2 z_2 + b_2) + c) \leq \psi(\sigma(U_1 z_2 + b_1) + c),$$

so $\sigma(U_1 z_2 + b_1) + c > 0$. From $c < 0$, we have $U_1 z_2 + b_1 > 0$. Similarly, we have $U_1 z_{-2} + b_1 > 0$. It is easy to see that $U_1 z_1 + b_1 \geq \min\{U_1 z_2 + b_1, U_1 z_{-2} + b_1\}$, so $\sigma(U_1 z_1 + b_1) + c > 0$, and hence $U_1 z_1 + b_1 > 0$. Similarly, we have $U_1 z_{-1} + b_1 > 0$. From $\widetilde{\mathcal{F}}(z_1) = -1$ and $\sigma(U_1 z_1 + b_1) + c > 0$, we have

$$0 > \mathcal{F}(z_1) = \sigma(U_1 z_1 + b_1) - \sigma(U_2 z_1 + b_2) + c > -\sigma(U_2 z_1 + b_2),$$

so $U_2 z_1 + b_2 > 0$. Similarly, we have $U_2 z_{-1} + b_2 > 0$. Now consider the linear function $L(x) = (U_1 x + b_1) - (U_2 x + b_2) + c$.

Since $c < 0$, $U_1 z_1 + b_1 > 0$, $U_2 z_1 + b_2 > 0$, and $\mathcal{F}(z_1) = -1$, we have $L(z_1) = (U_1 z_1 + b_1) - (U_2 z_1 + b_2) + c = \sigma(U_1 z_1 + b_1) - \sigma(U_2 z_1 + b_2) + c = \mathcal{F}(z_1) < 0$. Similarly, $L(z_{-1}) < 0$.

Since $c < 0$, $U_1 z_2 + b_1 > 0$, and $\mathcal{F}(z_2) = 1$, we have $L(z_2) = (U_1 z_2 + b_1) - (U_2 z_2 + b_2) + c = \sigma(U_1 z_2 + b_1) - (U_2 z_2 + b_2) + c \geq \sigma(U_1 z_2 + b_1) - \sigma(U_2 z_2 + b_2) + c > 0$. Similarly, we have $L(z_{-2}) > 0$.

Hence $L(0) = (L(z_1) + L(z_{-1}))/2 < 0$ and $L(0) = (L(z_2) + L(z_{-2}))/2 > 0$, a contradiction, so $s_2 = 1$. $\qquad\square$

We now give the proof of Theorem 4.1.

*Proof.* Let $\varphi(k, m) = \wedge_{i=1}^{m}\varphi_i(k, m)$ be a 6-SAT for $k$ variables, where $\varphi_i(k, m) = \vee_{j=1}^{6}\tilde{x}_{i,j}$ and $\tilde{x}_{i,j}$ is either $x_s$ or $\neg x_s$ for some $s \in [k]$ (see Definition 4.2).

For $i \in [m]$, define $Q_i^\varphi \in \mathbb{R}^k$ as follows: $Q_i^\varphi[j] = 1$ if $x_j$ occurs in $\varphi_i(k, m)$; $Q_i^\varphi[j] = -1$ if $\neg x_j$ occurs in $\varphi_i(k, m)$; $Q_i^\varphi[j] = 0$ otherwise. Then six entries in $Q_i^\varphi$ are 1 or $-1$ and all other entries are zero.

We define a binary classification dataset $\mathcal{D}(\varphi) = \{(x_i, y_i)\}_{i=1}^{m+4k} \subset \mathbb{R}^k \times [2]$ as follows.

    (1) For $i \in [k]$, $x_i = k\mathbf{1}_i$, $y_i = 1$.

    (2) For $i \in \{k+1, k+2, \ldots, 2k\}$, $x_i = -k\mathbf{1}_{i-k}$, $y_i = 1$.

    (3) For $i \in \{2k+1, 2k+2, \ldots, 3k\}$, $x_i = 2k\mathbf{1}_{i-2k}$, $y_i = 2$.

    (4) For $i \in \{3k+1, 3k+2, \ldots, 4k\}$, $x_i = -2k\mathbf{1}_{i-3k}$, $y_i = 2$.

    (5) For $i \in \{4k+1, 3k+2, \ldots, 4k+m\}$, $x_i = k/4 \cdot Q_{i-4k}^\varphi$, $y_i = 1$.

$\mathcal{D}(\varphi)$ has separation bound $\geq k/4 > 1$, because $k \geq 6$ for 6-SAT problem. Let the robust radius be $\alpha = 0.5 - \gamma$, where $\gamma < \frac{1}{10k}$. It is clear that both $\mathcal{D}$ and $\alpha$ require $\text{poly}(m, k)$ byte representation.

We claim that $\text{RobM}(\mathcal{D}(\varphi), \alpha)$ **has a solution** $\mathcal{F} \in \mathbf{H}_{n,2,2,*}$ **if and only if the reversible 6-SAT** $\varphi(k, m)$ **has a solution** $J = \{x_j = v_j\}_{j=1}^k$. **Furthermore,** $\mathcal{F}$ **and** $J$ **can be deduced from each other in polynomial time; that is,** $\text{RobM}(\mathcal{D}(\varphi), \alpha)$ **is computationally equivalent to** $\varphi(k, m)$. Since reversible 6-SAT is NPC Megiddo (1988), by the claim, $\text{RobM}(\mathcal{D}(\varphi), \alpha)$ is NPC, which implies that $\text{RobM}(\mathcal{D}, \alpha)$ is NP-hard for $\mathcal{D} \in \mathcal{D}_{n,L,N}$. This proves the theorem.

Before proving the claim, we first introduce a notation. Let $J = \{x_j = v_j\}_{j=1}^k$ be a solution to the reversible 6-SAT problem $\varphi$ and $\varphi_i(k, m) = \vee_{j=1}^6 \tilde{x}_{i,j}$ a clause of $\varphi$, where $v_i \in \{-1, 1\}$. Then denote by $q(J, \varphi_i)$ the number of $\tilde{x}_{i,j}$ which has value 1 on the solution $J$. If $q(J, \varphi_i) = 0$, then $\varphi_i$ is not true. If $q(J, \varphi_i) = 6$, then $\neg\varphi_i$ is not true. Since $J$ is a solution to the reversible 6-SAT problem $\varphi$, we have $1 \leq q(J, \varphi_i) \leq 5$. It is easy to see that $q(J, \varphi_i) = |\{j \in [k] : Q_i^\varphi[j] = v_j\}|$.

The claim will be proved in two steps.

**Step 1. We prove that if** $\varphi(k, m)$ **has a solution** $J = \{x_j = v_j\}_{j=1}^k$, **then** $\text{RobM}(\mathcal{D}(\varphi), 0.5 - \gamma)$ **has a solution** $\mathcal{F}$, **where** $v_i \in \{-1, 1\}$. Let $U_1 = (v_1, v_2, \ldots, v_k)$, $U_2 = -(v_1, v_2, \ldots, v_k)$. Define $\mathcal{F} \in \mathcal{H}_{k,2}$ to be $\mathcal{F}(x) = \sigma(U_1 x - 1.5k) + \sigma(U_2 x - 1.5k) + 1.5 - \gamma$. It is clear that $\mathcal{F}$ can be obtained from $J$ in $\text{poly}(k)$. We will show that $\mathcal{F}(x)$ is a robust memorization of $\mathcal{D}(\varphi)$ with budget $\alpha = 0.5 - \gamma$.

It suffices to show that for any $i \in [4k + m]$, $\mathcal{F}$ is robust at $x_i$ with budget $\alpha$. For $x \in \mathbb{B}(x_i, \alpha)$, there exists an $\epsilon \in [-\alpha, \alpha]^k$ such that $x = x_i + \epsilon$. The proof will be divided into three cases: (c1) - (c3).

**(c1).** Assume $i \in [2k]$. It suffices to prove the result for $i \in [k]$ and the result for $i = k+1, \ldots, 2k$ can be proven similarly. Since $v_i \in \{-1, 1\}$, we have $||U_1||_1 = k$. Then for $i \in [k]$, we have $x_i = k\mathbf{1}_i$ and hence $U_1(x_i + \epsilon) - 1.5k \leq k|v_i| + \epsilon||U_1||_1 - 1.5k \leq k|v_i| + (0.5 - \gamma)||U_1||_1 - 1.5k = -\gamma k < 0$. Similarly, we have $U_2(x_i + \epsilon) - 1.5k < 0$. Then,
$$\mathcal{F}(x) = \mathcal{F}(x_i + \epsilon) = \sigma(U_1(x_i + \epsilon) - 1.5k) + \sigma(U_2(x_i + \epsilon) - 1.5k) + 1.5 - \gamma < 1.5.$$
Therefore, $|\mathcal{F}(x) - 1| < |\mathcal{F}(x) - 2|$. Thus $\widehat{\mathcal{F}}(x) = 1$ and $\mathcal{F}$ is robust at $x_i$ with budget $\alpha = 0.5 - \gamma$. Case (c1) is proved.

**(c2).** Assume $i \in \{2k+1, \ldots, 4k\}$. We need only prove $i \in \{2k+1, \ldots, 3k\}$, and the other cases can be proved similarly. Since $U_1 = -U_2$, at least one of the following two equations $U_1 x_i - 1.5k = U_1^{(i)} x_i^{(i-2k)} - 1.5k = 0.5k$ and $U_2 x_i - 1.5k = U_2^{(i)} x_i^{(i-2k)} - 1.5k = 0.5k$ is true, say the first is true. Then, for any $||\epsilon||_\infty \leq 0.5 - \gamma$, we have
$$\mathcal{F}(x_i + \epsilon) = \sigma(U_1(x_i + \epsilon) - 1.5k) + \sigma(U_2(x_i + \epsilon) - 1.5k) - \gamma + 1.5 \geq \sigma(U_1(x_i + \epsilon) - 1.5k) - \gamma + 1.5.$$
We have $U_1(x_i + \epsilon) - 1.5k = 0.5k + U_1\epsilon \geq 0.5k - (0.5 - \gamma)k = \gamma k$. So $\mathcal{F}(x_i + \epsilon) \geq \gamma k - \gamma + 1.5 > 1.5$, since $k > 1$. Therefore, $|\mathcal{F}(x) - 2| < |\mathcal{F}(x) - 1|$. Thus $\widehat{\mathcal{F}}(x) = 2$, so $\mathcal{F}$ is robust at $x_i$ with budget $0.5 - \gamma$. Case (c2) is proved.

**(c3).** Assume $i \in \{4k+1, 4k+2, \ldots, 4k+m\}$. It is clear that $q(J, \varphi_{i-4k}) + q(J, \overline{\varphi}_{i-4k}) = 6$.

Then
$$\frac{k}{4} U_1 Q_{i-4k}^\varphi$$
$$= \sum_{j : x_j \in \varphi_{i-4k}} \frac{k}{4} v_j Q_{i-4k}^\varphi[j]$$
$$= \sum_{j : x_j \in \varphi_{i-4k}, \text{Sgn}(Q_{i-4k}^\varphi[j]) = \text{Sgn}(v_j)} k/4 - \sum_{j : x_j \in \varphi_{i-4k}, \text{Sgn}(Q_{i-4k}^\varphi[j]) \neq \text{Sgn}(v_j)} k/4$$
$$= q(J, \varphi_{i-4k})k/4 - q(J, \overline{\varphi}_{i-4k})k/4$$
$$\in \{0, k/2, k, -k/2, -k\},$$

which means $|\frac{k}{4}U_1 Q^\varphi_{i-4k}| \le k$. Similarly, we also have $|\frac{k}{4}U_2 Q^\varphi_{i-4k}| \le k$. As a consequence, $U_1 x_i - 1.5k = U_1 Q^\varphi_{i-4k} \cdot k/4 - 1.5k \le -0.5k$. Since $||U_1||_1 = k$, for any $||\epsilon||_\infty \le 0.5 - \gamma$, we have $U_1(x_i + \epsilon) - 1.5k \le -0.5k + ||U||_1(0.5 - \gamma) = -\gamma k < 0$. Similarly, $U_1(x_i + \epsilon) - 1.5k < 0$. We thus have

$$\mathcal{F}(x_i + \epsilon) = \sigma(U_1(x_i + \epsilon) - 1.5k) + \sigma(U_2(x_i + \epsilon) - 1.5k) - \gamma + 1.5 \le 0 + 0 - \gamma + 1.5 < 1.5.$$

Thus $\widehat{\mathcal{F}}(x) = 2$ and $\mathcal{F}$ is robust at $x_i$ with budget $0.5 - \gamma$.

From (c1) to (c3), $\mathcal{F}$ is a robust memorization of $\mathcal{D}(\varphi)$ with budget $\alpha = 0.5 - \gamma$, and Step 1 is proved.

**Step 2. We prove that if** $\mathrm{RobM}(\mathcal{D}(\varphi), 0.5 - \gamma)$ **has a solution** $\mathcal{F}(x) = s_1\sigma(U_1 x + b_1) + s_2\sigma(U_2 x + b_2) + C \in \mathcal{H}_{k,2}$ **which is a robust memorization of** $\mathcal{D}(\varphi)$ **with budget** $\alpha = 0.5 - \gamma$, **then** $\varphi(k, m)$ **has a solution** $J$. Since $a\sigma(b) = \mathrm{Sgn}(a)\sigma(|a|b)$, we can assume that $s_1, s_2 \in \{-1, 1\}$.

Let $c = -1.5 + C$, which means $\mathcal{F}(x) - 1.5 = s_1\sigma(U_1 x + b_1) + s_2\sigma(U_2 x + b_2) + c$. Therefore, we have $\widehat{\mathcal{F}}(x) = 1$ if $\mathcal{F}(x) - 1.5 < 0$, and $\widehat{\mathcal{F}}(x) = 2$ if $\mathcal{F}(x) - 1.5 > 0$.

The proof of Step 2 is divided into two sub-steps, and we prove Step 2.1 in two cases.

**Step 2.1. Assuming** $c \ge 0$, **we will show that** $J = \{x_i = \mathrm{Sgn}(U_1^{(i)})\}_{i=1}^k$ **is the solution to the reversible 6-SAT problem** $\varphi(k, m)$. We will prove Step 2.1 by proving six properties: (d1) - (d6).

(d1): We have $s_1 s_2 = -1$. Without loss of generality, we let $s_1 = 1$ and $s_2 = -1$. Since $\mathcal{F}(x_1) - 1.5 = \mathcal{F}(x_{k+1}) - 1.5 < 0$ and $\mathcal{F}(x_{2k+1}) - 1.5 = \mathcal{F}(x_{3k+1}) - 1.5 > 0$, (d1) is a consequence of Lemma B.1.

(d2): We have $-k|U_2^{(q)}| + b_2 + p||U_2||_1 > 0$ for any $p \in [-0.5 + \gamma, 0.5 - \gamma]$ and $q \in [k]$. We just need to prove the case $q = 1$. We first assume $U_2^{(1)} \le 0$. Since $\mathcal{F}$ is a robust memorization of $\mathcal{D}(\varphi)$ with budget $\alpha = 0.5 - \gamma$, we have $\mathcal{F}(x_1 + p\mathrm{Sgn}(U_2)) < 1.5$, so

$$
\begin{aligned}
0 &> \sigma(U_1(x_1 + p\mathrm{Sgn}(U_2)) + b_1) - \sigma(U_2(x_1 + p\mathrm{Sgn}(U_2)) + b_2) + c \\
&\ge -\sigma(U_2(x_1 + p\mathrm{Sgn}(U_2)) + b_2) + c \\
&= -\sigma(-k|U_2^{(1)}| + p||U_2||_1 + b_2) + c \\
&\ge -\sigma(-k|U_2^{(1)}| + p||U_2||_1 + b_2) \quad (by\ c \ge 0)
\end{aligned}
$$

which means $\sigma(-k|U_2^{(1)}| + p||U_2||_1 + b_2) > 0$. Then we have $-k|U_2^{(1)}| + p||U_2||_1 + b_2 > 0$. For the case $U_2^{(1)} \ge 0$, we only need to consider $x_{k+1}$ instead of $x_1$. So property (d2) is proved.

(d3): We have $U_1^{(q)} U_2^{(q)} > 0$ and $|U_1^{(q)}| > |U_2^{(q)}|$ for any $q \in [k]$. We just need to prove it for $q = 1$. First, we show that $U_2^{(1)} \ne 0$. If $U_2^{(1)} = 0$, we first assume $U_1^{(1)} \le 0$. Since $\mathcal{F}(x_1) < 1.5$ and $\mathcal{F}(x_{2k+1}) > 1.5$, we have

$$
\begin{aligned}
0 &> \mathcal{F}(x_1) - 1.5 \\
&= \sigma(U_1 x_1 + b_1) - \sigma(U_2 x_1 + b_2) + c \\
&= \sigma(kU_1^{(1)} + b_1) - \sigma(b_2) + c \quad (by\ U_2^{(1)} = 0) \\
&\ge \sigma(2kU_1^{(1)} + b_1) - \sigma(b_2) + c \quad (by\ U_1^{(1)} \le 0) \\
&= \sigma(2kU_1^{(1)} + b_1) - \sigma(2kU_2^{(1)} + b_2) + c \quad (by\ U_2^{(1)} = 0) \\
&= \sigma(U_1 x_{2k+1} + b_1) - \sigma(U_2 x_{2k+1} + b_2) + c \\
&= \mathcal{F}(x_{2k+1}) - 1.5 \\
&> 0,
\end{aligned}
$$

a contradiction, and thus $U_2^{(1)} \neq 0$. If $U_1^{(1)} \geq 0$, we only need to consider $x_{k+1}$ and $x_{3k+1}$.

Now we prove (d3). Let $h = 1$ if $U_2^{(1)} > 0$, and $h = k + 1$ if $U_2^{(1)} < 0$. Because $x_{h+2k} = 2x_h$ and $\mathcal{F}(x_{h+2k}) - 1.5 > 0 > \mathcal{F}(x_h) - 1.5$, we have

$$
\begin{aligned}
&\mathcal{F}(x_{h+2k}) - 1.5 \\
=\ & \sigma(U_1 x_{h+2k} + b_1) - \sigma(U_2 x_{h+2k} + b_2) + c \\
=\ & \sigma(2U_1 x_h + b_1) - \sigma(2U_2 x_h + b_2) + c \\
=\ & \sigma(2kU_1^{(1)} \mathrm{Sgn}(U_2^{(1)}) + b_1) - \sigma(2k|U_2^{(1)}| + b_2) + c \\
>\ & 0 \quad (by\ \mathcal{F}(x_{h+2k}) - 1.5 > 0) \\
>\ & \sigma(U_1 x_h + b_1) - \sigma(U_2 x_h + b_2) + c \quad (by\ \mathcal{F}(x_h) - 1.5 < 0) \\
=\ & \sigma(kU_1^{(1)} \mathrm{Sgn}(U_2^{(1)}) + b_1) - \sigma(k|U_2^{(1)}| + b_2) + c
\end{aligned}
$$

which means $\sigma(2kU_1^{(1)} \mathrm{Sgn}(U_2^{(1)}) + b_1) - \sigma(2k|U_2^{(1)}| + b_2) > \sigma(kU_1^{(1)} \mathrm{Sgn}(U_2^{(1)}) + b_1) - \sigma(k|U_2^{(1)}| + b_2)$, so we have

$$
\begin{aligned}
0\ <\ & \sigma(2kU_1^{(1)} \mathrm{Sgn}(U_2^{(1)}) + b_1) - \sigma(2k|U_2^{(1)}| + b_2) - (\sigma(kU_1^{(1)} \mathrm{Sgn}(U_2^{(1)}) + b_1) - \sigma(k|U_2^{(1)}| + b_2)) \\
=\ & (\sigma(2kU_1^{(1)} \mathrm{Sgn}(U_2^{(1)}) + b_1) - \sigma(kU_1^{(1)} \mathrm{Sgn}(U_2^{(1)}) + b_1)) - (\sigma(2k|U_2^{(1)}| + b_2) - \sigma(k|U_2^{(1)}| + b_2)) \\
=\ & (\sigma(2kU_1^{(1)} \mathrm{Sgn}(U_2^{(1)}) + b_1) - \sigma(kU_1^{(1)} \mathrm{Sgn}(U_2^{(1)}) + b_1)) - ((2k|U_2^{(1)}| + b_2) - (k|U_2^{(1)}| + b_2)) \quad (by\ (d2)) \\
=\ & (\sigma(2kU_1^{(1)} \mathrm{Sgn}(U_2^{(1)}) + b_1) - \sigma(kU_1^{(1)} \mathrm{Sgn}(U_2^{(1)}) + b_1)) - kU_2^{(1)}.
\end{aligned}
$$

Then $\sigma(2kU_1^{(1)} \mathrm{Sgn}(U_2^{(1)}) + b_1) - \sigma(kU_1^{(1)} \mathrm{Sgn}(U_2^{(1)}) + b_1) > kU_2^{(1)} \geq 0$, which means $2kU_1^{(1)} \mathrm{Sgn}(U_2^{(1)}) + b_1 > 0$. And according to that, we have:

$$
\begin{aligned}
0\ <\ & (\sigma(2kU_1^{(1)} \mathrm{Sgn}(U_2^{(1)}) + b_1) - \sigma(kU_1^{(1)} \mathrm{Sgn}(U_2^{(1)}) + b_1)) - k|U_2^{(1)}| \\
=\ & ((2kU_1^{(1)} \mathrm{Sgn}(U_2^{(1)}) + b_1) - \sigma(kU_1^{(1)} \mathrm{Sgn}(U_2^{(1)}) + b_1)) - k|U_2^{(1)}| \\
\leq\ & ((2kU_1^{(1)} \mathrm{Sgn}(U_2^{(1)}) + b_1) - (kU_1^{(1)} \mathrm{Sgn}(U_2^{(1)}) + b_1)) - k|U_2^{(1)}| \\
=\ & kU_1^{(1)} \mathrm{Sgn}(U_2^{(1)}) - k|U_2^{(1)}|.
\end{aligned}
$$

So we get $U_1^{(1)} \mathrm{Sgn}(U_2^{(1)}) > |U_2^{(1)}| > 0$, which means $\mathrm{Sgn}(U_1^{(1)}) = \mathrm{Sgn}(U_2^{(1)})$, and $|U_1^{(1)}| > |U_2^{(1)}|$. (d3) is proved.

(d4): We have $2k|U_1^{(q)}| + b_1 + p||U_1||_1 > 0$ for any $p \in [-0.5 + \gamma, 0.5 - \gamma]$ and $q \in [k]$.

We just need to prove it for $q = 1$. Let $h = 1$ if $U_1^{(1)} > 0$, and $h = k + 1$ if $U_1^{(1)} < 0$. Because $\mathcal{F}(x_{h+2k} + p\mathrm{Sgn}(U_1)) - 1.5 > 0 > \mathcal{F}(x_h + p\mathrm{Sgn}(U_1)) - 1.5$, we have that:

$$
\begin{aligned}
&\mathcal{F}(x_{h+2k} + p\mathrm{Sgn}(U_1)) - 1.5 \\
=\ & \sigma(U_1(x_{h+2k} + p\mathrm{Sgn}(U_1)) + b_1) - \sigma(U_2(2x_{h+2k} + p\mathrm{Sgn}(U_1)) + b_2) + c \\
=\ & \sigma(2k|U_1^{(1)}| + b_1 + p||U_1||_1) - \sigma(2k|U_2^{(1)}| + b_2 + p||U_2||_1) + c \quad (by\ (d3)) \\
=\ & \sigma(2k|U_1^{(1)}| + b_1 + p||U_1||_1) - (2k|U_2^{(1)}| + b_2 + p||U_2||_1) + c \quad (by\ (d2)) \\
>\ & 0 \quad (by\ \mathcal{F}(x_{h+2k} + p\mathrm{Sgn}(U_1)) - 1.5 > 0) \\
>\ & \sigma(U_1(x_h + p\mathrm{Sgn}(U_1)) + b_1) - \sigma(U_2(x_h + p\mathrm{Sgn}(U_1)) + b_2) + c \quad (by\ 0 > \mathcal{F}(x_h + p\mathrm{Sgn}(U_1)) - 1.5) \\
=\ & \sigma(k|U_1^{(1)}| + b_1 + p||U_1||_1) - \sigma(k|U_2^{(1)}| + b_2 + p||U_2||_1) + c \quad (by\ (d3)) \\
=\ & \sigma(k|U_1^{(1)}| + b_1 + p||U_1||_1) - (k|U_2^{(1)}| + b_2 + p||U_2||_1) + c \quad (by\ (d2))
\end{aligned}
$$

which means $\sigma(2k|U_1^{(1)}| + b_1 + p||U_1||_1) - (2k|U_2^{(1)}| + b_2 + p||U_2||_1) + c > \sigma(k|U_1^{(1)}| + b_1 + p||U_1||_1) - (k|U_2^{(1)}| + b_2 + p||U_2||_1) + c$, so we have

$$\sigma(2k|U_1^{(1)}| + b_1 + p||U_1||_1) > \sigma(k|U_1^{(1)}| + b_1 + p||U_1||_1) + k|U_2^{(1)}| > 0.$$

(d4) is proved.

(d5): We have $\max_{z \in [k]}(|U_1^{(z)}| - |U_2^{(z)}|) < 2(1 - 2\gamma)(||U_1||_1 - ||U_2||_1)$.

For any $z \in [k]$, let $h = z$ if $U_1^{(z)} > 0$, and $h = z + k$ if $U_1^{(z)} < 0$. We have $\mathcal{F}(x_h + (0.5 - \gamma)\mathrm{Sgn}(U_1)) - 1.5 < 0$, which means

$$
\begin{aligned}
& 0 \\
> \quad & \sigma(U_1(x_h + (0.5 - \gamma)\mathrm{Sgn}(U_1)) + b_1) - \sigma(U_2(x_h + (0.5 - \gamma)\mathrm{Sgn}(U_1)) + b_2) + c \\
= \quad & \sigma(k|U_1^{(z)}| + (0.5 - \gamma)||U_1||_1 + b_1) - \sigma(k|U_2^{(z)}| + (0.5 - \gamma)||U_2||_1 + b_2) + c \quad (by\ (d3)) \\
= \quad & \sigma(k|U_1^{(z)}| + (0.5 - \gamma)||U_1||_1 + b_1) - (k|U_2^{(z)}| + (0.5 - \gamma)||U_2||_1 + b_2) + c \quad (by\ (d2)) \\
\geq \quad & (k|U_1^{(z)}| + (0.5 - \gamma)||U_1||_1 + b_1) - (k|U_2^{(z)}| + (0.5 - \gamma)||U_2||_1 + b_2) + c \\
= \quad & k|U_1^{(z)}| - k|U_2^{(z)}| + (0.5 - \gamma)(||U_1||_1 - ||U_2||_1) + b_1 - b_2 + c.
\end{aligned}
$$

We thus have $k|U_1^{(z)}| - k|U_2^{(z)}| < -b_1 + b_2 - c - (0.5 - \gamma)(||U_1||_1 - ||U_2||_1)$. Then we have $\mathcal{F}(x_{h+2k} - (0.5 - \gamma)\mathrm{Sgn}(U_1)) - 1.5 > 0$, which means

$$
\begin{aligned}
& 0 \\
< \quad & \sigma(U_1(x_{2k+h} - (0.5 - \gamma)\mathrm{Sgn}(U_1)) + b_1) - \sigma(U_2(x_{2k+h} - (0.5 - \gamma)\mathrm{Sgn}(U_1)) + b_2) + c \\
= \quad & \sigma(2k|U_1^{(z)}| - (0.5 - \gamma)||U_1||_1 + b_1) - \sigma(2k|U_2^{(z)}| - (0.5 - \gamma)||U_2||_1 + b_2) + c \quad (by\ (d3)) \\
= \quad & (2k|U_1^{(z)}| - (0.5 - \gamma)||U_1||_1 + b_1) - \sigma(2k|U_2^{(z)}| - (0.5 - \gamma)||U_2||_1 + b_2) + c \quad (by\ (d4)) \\
\leq \quad & (2k|U_1^{(z)}| - (0.5 - \gamma)||U_1||_1 + b_1) - (2k|U_2^{(z)}| - (0.5 - \gamma)||U_2||_1 + b_2) + c \\
= \quad & 2k|U_1^{(z)}| - 2k|U_2^{(z)}| - (0.5 - \gamma)(||U_1||_1 - ||U_2||_1) + b_1 - b_2 + c.
\end{aligned}
$$

So, we have $k|U_1^{(z)}| - k|U_2^{(z)}| > \frac{-b_1 + b_2 - c + (0.5 - \gamma)(||U_1||_1 - ||U_2||_1)}{2}$, and thus

$$
\begin{aligned}
& k|U_1^{(z)}| - k|U_2^{(z)}| \\
< \quad & -b_1 + b_2 - c - (0.5 - \gamma)(||U_1||_1 - ||U_2||_1) \\
= \quad & 2\frac{-b_1 + b_2 - c + (0.5 - \gamma)(||U_1||_1 - ||U_2||_1)}{2} - (1 - 2\gamma)(||U_1||_1 - ||U_2||_1) \\
< \quad & 2k|U_1^{(z)}| - 2k|U_2^{(z)}| - (1 - 2\gamma)(||U_1||_1 - ||U_2||_1)
\end{aligned}
$$

which means $k|U_1^{(z)}| - k|U_2^{(z)}| > (1 - 2\gamma)(||U_1||_1 - ||U_2||_1)$ for any $z \in [k]$. Using this inequality, we have

$$
\begin{aligned}
& k|U_1^{(z)}| - k|U_2^{(z)}| \\
= \quad & k(||U_1||_1 - ||U_2||_1) - \sum_{z' \neq z}(k|U_1^{z'}| - k|U_2^{z'}|) \\
< \quad & (k - (1 - 2\gamma)(k - 1))(||U_1||_1 - ||U_2||_1) \\
< \quad & 1.1(||U_1||_1 - ||U_2||_1) \quad (by\ (d3)\ and\ \gamma < 1/(10k)) \\
< \quad & 2 * (1 - 2\gamma)(||U_1||_1 - ||U_2||_1) \quad (by\ (d3))
\end{aligned}
$$

which proves (d5).

(d6): We have that $\{x_i = \mathrm{Sgn}(U_1^{(i)})\}_{i=1}^k$ is a solution to the reversible 6-SAT problem $\varphi(k, m)$.

If this is not valid, then there exists an $i \in [m]$ such that $q(\{\mathrm{Sgn}(U_1^{(w)})\}_{w=1}^k, \phi_i) = 6$ or $q(\{\mathrm{Sgn}(U_1^{(w)})\}_{w=1}^k, \phi_i) = 0$. We just need to consider the first case, because when $q(\{\mathrm{Sgn}(U_1^{(w)})\}_{w=1}^k, \phi_i) = 0$, there exists a $j \in [m]$ such that $\overline{\phi_j} = \phi_i$, so $q(\{\mathrm{Sgn}(U_1^{(w)})\}_{w=1}^k, \phi_j) = 6$.

Without loss of generality, we assume that the index of the six entries in $\phi_i$ are $1, 2, 3, 4, 5, 6$. By the definition of $x_{4k+i}$, we know that $U_1 x_{4k+i} = \frac{k}{4} \sum_{j=1}^6 |U_1^{(z)}|$, and by (d3), we know that $U_2 x_{4k+i} = \frac{k}{4} \sum_{j=1}^6 |U_2^{(z)}|$.

Using (d2), we know that

$$0 < -k|U_2^{(1)}| + b_2 + (0.5 - \gamma)\|U_2\|_1 < \tfrac{k}{4} \sum_{j=1}^6 |U_2^{(z)}| + b_2 + (0.5 - \gamma)\|U_2\|_1. \tag{6}$$

Without loss of generality, we assume $U_1^{(1)} > 0$. Since $\mathcal{F}(x_{2k+1} - (0.5 - \gamma)\mathrm{Sgn}(U_1)) - 1.5 > 0$, we have

$$
\begin{aligned}
& 0 \\
< \quad & \sigma(U_1(x_{2k+1} - (0.5 - \gamma)\mathrm{Sgn}(U_1)) + b_1) - \sigma(U_2(2x_{2k+1} - (0.5 - \gamma)\mathrm{Sgn}(U_1)) + b_2) + c \\
= \quad & \sigma(2k|U_1^{(1)}| + b_1 - (0.5 - \gamma)\|U_1\|_1) - \sigma(2k|U_2^{(1)}| + b_2 - (0.5 - \gamma)\|U_2\|_1) + c \quad (by\ (d3)) \\
\leq \quad & \sigma(2k|U_1^{(1)}| + b_1 - (0.5 - \gamma)\|U_1\|_1) - (2k|U_2^{(1)}| + b_2 - (0.5 - \gamma)\|U_2\|_1) + c \\
= \quad & (2k|U_1^{(1)}| + b_1 - (0.5 - \gamma)\|U_1\|_1) - (2k|U_2^{(1)}| + b_2 - (0.5 - \gamma)\|U_2\|_1) + c. \quad (by\ (d4))
\end{aligned}
\tag{7}
$$

So $0 < (2k|U_1^{(1)}| + b_1 - (0.5 - \gamma)\|U_1\|_1) - (2k|U_2^{(1)}| + b_2 - (0.5 - \gamma)\|U_2\|_1) + c$. If $U_1^{(1)} < 0$. We only need to consider $x_{3k+1}$, and the others are the same. The conclusions are the same for $U_1^{(i)}, i = 1, \ldots, 6$.

Then because $\mathcal{F}(x_{4k+i} + (0.5 - \gamma)\mathrm{Sgn}(U_1)) - 1.5 < 0$, we have

$$
\begin{aligned}
& 0 \\
> \quad & \sigma(U_1(x_{4k+i} + (0.5 - \gamma)\mathrm{Sgn}(U_1)) + b_1) - \sigma(U_2(x_{4k+i} + (0.5 - \gamma)\mathrm{Sgn}(U_1)) + b_2) + c \\
= \quad & \sigma(U_1 x_{4k+i} + b_1 + (0.5 - \gamma)\|U_1\|_1) - \sigma(U_2 x_{4k+i} + b_2 + (0.5 - \gamma)\|U_2\|_1) + c \quad (by\ (d3)) \\
= \quad & \sigma(\tfrac{k}{4} \sum_{j=1}^6 |U_1^{(z)}| + b_1 + (0.5 - \gamma)\|U_1\|_1) - \sigma(\tfrac{k}{4} \sum_{j=1}^6 |U_2^{(z)}| + b_2 + (0.5 - \gamma)\|U_2\|_1) + c \\
= \quad & \sigma(\tfrac{k}{4} \sum_{j=1}^6 |U_1^{(z)}| + b_1 + (0.5 - \gamma)\|U_1\|_1) - (\tfrac{k}{4} \sum_{j=1}^6 |U_2^{(z)}| + b_2 + (0.5 - \gamma)\|U_2\|_1) + c \quad (by\ (6)) \\
\geq \quad & \tfrac{k}{4} \sum_{j=1}^6 |U_1^{(z)}| + b_1 + (0.5 - \gamma)\|U_1\|_1 - (\tfrac{k}{4} \sum_{j=1}^6 |U_2^{(z)}| + b_2 + (0.5 - \gamma)\|U_2\|_1) + c \\
= \quad & \tfrac{1}{6} \sum_{j=1}^6 (2k|U_1^{(z)}| + b_1 - (0.5 - \gamma)\|U_1\|_1 - 2k|U_2^{(z)}| - b_2 + (0.5 - \gamma)\|U_2\|_1 + c) \\
& - \tfrac{k}{12} \sum_{k=1}^6 (|U_1^{(z)}| - |U_2^{(z)}|) + (1 - 2\gamma)(\|U_1\|_1 - \|U_2\|_1) \\
\geq \quad & - \tfrac{k}{12} \sum_{k=1}^6 (|U_1^k| - |U_2^k|) + (1 - 2\gamma)(\|U_1\|_1 - \|U_2\|_1) \quad (by\ (7)) \\
> \quad & 0 \quad (by\ (d5))
\end{aligned}
$$

a contradiction, and Step 2.1 is proved.

**Step 2.2. Assuming $c < 0$, we will show that $J = \{x_i = \mathrm{Sgn}(U_1^{(i)})\}_{i=1}^k$ is a solution to the reversible 6-SAT problem $\varphi(k, m)$.** The proof is divided into six steps: (e1) - (e6).

(e1): There must be $s_1 = s_2 = 1$.

Just use Lemma B.1.

(e2): $U_1^{(q)} U_2^{(q)} < 0$ for any $q \in [k]$.

We just need to prove it for $q = 1$. First we prove that $U_1^{(1)} \neq 0$. If not, that is $U_1^{(1)} = 0$. Without loss of generality, let $U_2^{(1)} \leq 0$. Since $\mathcal{F}(x_1) - 1.5 < 0$ and $\mathcal{F}(x_{2k+1}) - 1.5 > 0$, we have

$$
\begin{aligned}
0 \\
> \quad & \mathcal{F}(x_1) - 1.5 \\
= \quad & \sigma(U_1 x_1 + b_1) + \sigma(U_2 x_1 + b_2) + c \\
= \quad & \sigma(b_1) + \sigma(k U_2^{(1)} + b_2) + c \quad (by\ U_1^{(1)} = 0) \\
\geq \quad & \sigma(b_1) + \sigma(2k U_2^{(1)} + b_2) + c \quad (by\ U_2^{(1)} \leq 0) \\
= \quad & \sigma(2k U_1^{(1)} + b_1) + \sigma(2k U_2^{(1)} + b_2) + c \quad (by\ U_1^{(1)} = 0) \\
= \quad & \sigma(U_1 x_{2k+1} + b_1) - \sigma(U_2 x_{2k+1} + b_2) + c \\
= \quad & \mathcal{F}(x_{2k+1}) - 1.5 \\
> \quad & 0
\end{aligned}
$$

which is a contradiction, and hence $U_1^{(1)} \neq 0$. When $U_2^{(1)} \geq 0$, we just need to consider $x_{k+1}$ and $x_{3k+1}$.

Now we prove (e2), let $h = 1 + k$ if $U_1^{(1)} > 0$, or $h = 1$ if $U_1^{(1)} < 0$. Then, we have that $\mathcal{F}(x_{h+2k}) - 1.5 > 0$ and $\mathcal{F}(x_h) - 1.5 < 0$, which means

$$
\begin{aligned}
& \sigma(U_1 x_{h+2k} + b_1) + \sigma(U_2 x_{h+2k} + b_2) - c \\
= \quad & \sigma(-2k|U_1^{(1)}| + b_1) + \sigma(-2k U_2^{(1)} \mathrm{Sgn}(U_1^{(1)}) + b_2) - c \\
> \quad & 0
\end{aligned}
\tag{8}
$$

and

$$
\begin{aligned}
& \sigma(U_1 x_h + b_1) + \sigma(U_2 x_h + b_2) - c \\
= \quad & \sigma(-k|U_1^{(1)}| + b_1) + \sigma(-k U_2^{(1)} \mathrm{Sgn}(U_1^{(1)}) + b_2) - c \\
< \quad & 0.
\end{aligned}
\tag{9}
$$

These two inequalities illustrate that $\sigma(-k|U_1^{(1)}| + b_1) + \sigma(-k U_2^{(1)} \mathrm{Sgn}(U_1^{(1)}) + b_2) < \sigma(-2k|U_1^{(1)}| + b_1) + \sigma(-2k U_2^{(1)} \mathrm{Sgn}(U_1^{(1)}) + b_2)$. Furthermore, since $\sigma(-k|U_1^{(1)}| + b_1) \geq \sigma(-2k|U_1^{(1)}| + b_1)$, we have $\sigma(-k U_2^{(1)} \mathrm{Sgn}(U_1^{(1)}) + b_2) < \sigma(-2k U_2^{(1)} \mathrm{Sgn}(U_1^{(1)}) + b_2)$, which means $(-k U_2^{(1)} \mathrm{Sgn}(U_1^{(1)}) + b_2) < (-2k U_2^{(1)} \mathrm{Sgn}(U_1^{(1)}) + b_2)$. Then $U_2^{(1)} \mathrm{Sgn}(U_1^{(1)}) < 0$, that is $U_1^{(1)} U_2^{(1)} < 0$, which is what we want.

(e3): $k|U_2^{(q)}| > (1 - 2\gamma)||U_2||_1$ for any $q \in [k]$.

We just need to prove it for $q = 1$. Let $h = 1$ if $U_2^{(1)} > 0$, or $h = k + 1$ if $U_2^{(1)} < 0$. We have $\mathcal{F}(x_{h+2k} - (0.5 - \gamma)\mathrm{Sgn}(U_2)) - 1.5 > 0$ and $\mathcal{F}(x_h + (0.5 - \gamma)\mathrm{Sgn}(U_2)) - 1.5 < 0$, which means

$$
\begin{aligned}
& \sigma(U_1(x_{h+2k} - (0.5 - \gamma)\mathrm{Sgn}(U_2)) + b_1) + \sigma(U_2(x_{h+2k} - (0.5 - \gamma)\mathrm{Sgn}(U_2)) + b_2) - c \\
= \quad & \sigma(-2k|U_1^{(1)}| + (0.5 - \gamma)||U_1||_1 + b_1) + \sigma(2k|U_2^{(1)}| - (0.5 - \gamma)||U_2||_1 + b_2) - c \quad (by\ (e2)) \\
> \quad & 0
\end{aligned}
\tag{10}
$$

and

$$
\begin{aligned}
& \sigma(U_1(x_h + (0.5 - \gamma)\mathrm{Sgn}(U_2)) + b_1) + \sigma(U_2(x_h + (0.5 - \gamma)\mathrm{Sgn}(U_2)) + b_2) - c \\
= \quad & \sigma(-k|U_1^{(1)}| - (0.5 - \gamma)||U_1||_1 + b_1) + \sigma(k|U_2^{(1)}| + (0.5 - \gamma)||U_2||_1 + b_2) - c \quad (by\ (e2)) \\
< \quad & 0.
\end{aligned}
\tag{11}
$$

Next, consider two situations:

**(e3.1)**: If $-2k|U_1^{(1)}| + (0.5-\gamma)||U_1||_1 + b_1 \le 0$, then we can prove that $k|U_2^{(1)}| > (1-2\gamma)||U_2||_1$.

By equation 10 and the fact $-2k|U_1^{(1)}| + (0.5-\gamma)||U_1||_1 + b_1 \le 0$, we have $\sigma(2k|U_2^{(1)}| - (0.5-\gamma)||U_2||_1 + b_2) - c > 0$.

By equation 11, we have $\sigma(k|U_2^{(1)}| + (0.5-\gamma)||U_2||_1 + b_2) - c \le \sigma(-k|U_1^{(1)}| - (0.5-\gamma)||U_1||_1 + b_1) + \sigma(k|U_2^{(1)}| + (0.5-\gamma)||U_2||_1 + b_2) - c < 0$.

So $\sigma(k|U_2^{(1)}| + (0.5-\gamma)||U_2||_1 + b_2) < c < \sigma(2k|U_2^{(1)}| - (0.5-\gamma)||U_2||_1 + b_2)$, which means $(k|U_2^{(1)}| + (0.5-\gamma)||U_2||_1 + b_2) < (2k|U_2^{(1)}| - (0.5-\gamma)||U_2||_1 + b_2)$. Then we get $(1-2\gamma)||U_2||_1 < k|U_2^{(1)}|$. This is what we want.

**(e3.2)**: If $-2k|U_1^{(1)}| + (0.5-\gamma)||U_1||_1 + b_1 > 0$, then we can prove $-2k|U_2^{(1)}| + (0.5-\gamma)||U_2||_1 + b_2 \le 0$ and $k|U_2^{(1)}| > (1-2\gamma)||U_2||_1$.

Since $\mathcal{F}(x_h - (0.5-\gamma)\text{Sgn}(U_2)) = -1 < 0$, we have that

$$
\begin{aligned}
& \sigma(U_1(x_h - (0.5-\gamma)\text{Sgn}(U_2)) + b_1) + \sigma(U_2(x_h - (0.5-\gamma)\text{Sgn}(U_2)) + b_2) - c \\
= \ & \sigma(-k|U_1^{(1)}| + (0.5-\gamma)||U_1||_1 + b_1) + \sigma(k|U_2^{(1)}| - (0.5-\gamma)||U_2||_1 + b_2) - c \quad (by \ (e2)) \\
< \ & 0.
\end{aligned}
\tag{12}
$$

Since $-2k|U_1^{(1)}| + (0.5-\gamma)||U_1||_1 + b_1 > 0$, it holds $-k|U_1^{(1)}| + (0.5-\gamma)||U_1||_1 + b_1 > 0$. Then by equation 10 and equation 12 and $-2k|U_1^{(1)}| + (0.5-\gamma)||U_1||_1 + b_1 > 0$, we have that

$$
\begin{aligned}
& \sigma(-k|U_1^{(1)}| + (0.5-\gamma)||U_1||_1 + b_1) + \sigma(k|U_2^{(1)}| - (0.5-\gamma)||U_2||_1 + b_2) - c \\
= \ & (-k|U_1^{(1)}| + (0.5-\gamma)||U_1||_1 + b_1) + \sigma(k|U_2^{(1)}| - (0.5-\gamma)||U_2||_1 + b_2) - c \\
< \ & 0 \\
< \ & \sigma(-2k|U_1^{(1)}| + (0.5-\gamma)||U_1||_1 + b_1) + \sigma(2k|U_2^{(1)}| - (0.5-\gamma)||U_2||_1 + b_2) - c \\
= \ & (-2k|U_1^{(1)}| + (0.5-\gamma)||U_1||_1 + b_1) + \sigma(2k|U_2^{(1)}| - (0.5-\gamma)||U_2||_1 + b_2) - c
\end{aligned}
\tag{13}
$$

which means $k|U_1^{(1)}| < \sigma(2k|U_2^{(1)}| - (0.5-\gamma)||U_2||_1 + b_2) - \sigma(k|U_2^{(1)}| - (0.5-\gamma)||U_2||_1 + b_2) \le k|U_2^{(1)}|$ (Use $\sigma(x) - \sigma(y) \le |x-y|$ here). So $|U_1^{(1)}| < |U_2^{(1)}|$.

Similarly, if $-2k|U_2^{(1)}| + (0.5-\gamma)||U_2||_1 + b_2 > 0$, then we have $|U_1^{(1)}| > |U_2^{(1)}|$. But $|U_1^{(1)}| < |U_2^{(1)}|$ and $|U_1^{(1)}| > |U_2^{(1)}|$ cannot stand simultaneously, so $-2k|U_2^{(1)}| + (0.5-\gamma)||U_2||_1 + b_2 > 0$ can not stand. Then we have $-2k|U_2^{(1)}| + (0.5-\gamma)||U_2||_1 + b_2 \le 0$.

Now using equation 10 and equation 11, we have

$$
\begin{aligned}
& \sigma(-2k|U_1^{(1)}| + (0.5-\gamma)||U_1||_1 + b_1) + \sigma(2k|U_2^{(1)}| - (0.5-\gamma)||U_2||_1 + b_2) - c \\
= \ & (-2k|U_1^{(1)}| + (0.5-\gamma)||U_1||_1 + b_1) + \sigma(2k|U_2^{(1)}| - (0.5-\gamma)||U_2||_1 + b_2) - c \\
> \ & 0 \\
> \ & \sigma(-k|U_1^{(1)}| - (0.5-\gamma)||U_1||_1 + b_1) + \sigma(k|U_2^{(1)}| + (0.5-\gamma)||U_2||_1 + b_2) - c \\
\ge \ & (-k|U_1^{(1)}| - (0.5-\gamma)||U_1||_1 + b_1) + \sigma(k|U_2^{(1)}| + (0.5-\gamma)||U_2||_1 + b_2) - c
\end{aligned}
$$

which means $(-k|U_1^{(1)}| - (0.5 - \gamma)||U_1||_1 + b_1) - (-2k|U_1^{(1)}| + (0.5 - \gamma)||U_1||_1 + b_1) < \sigma(2k|U_2^{(1)}| - (0.5 - \gamma)||U_2||_1 + b_2) - \sigma(k|U_2^{(1)}| + (0.5 - \gamma)||U_2||_1 + b_2)$. Since $-2k|U_2^{(1)}| + (0.5 - \gamma)||U_2||_1 + b_2 \le 0$, similar to (e3.1), we have $(1 - 2\gamma)||U_1||_1 < k|U_1^{(1)}|$.

So we can obtain

$$
\begin{aligned}
0 & \\
< \quad & -(1 - 2\gamma)||U_1||_1 + k|U_1^{(1)}| \quad (by \ equc3) \\
= \quad & (-k|U_1^{(1)}| - (0.5 - \gamma)||U_1||_1 + b_1) - (-2k|U_1^{(1)}| + (0.5 - \gamma)||U_1||_1 + b_1) \\
< \quad & \sigma(2k|U_2^{(1)}| - (0.5 - \gamma)||U_2||_1 + b_2) - \sigma(k|U_2^{(1)}| + (0.5 - \gamma)||U_2||_1 + b_2)
\end{aligned}
$$

which implies $\sigma(2k|U_2^{(1)}| - (0.5 - \gamma)||U_2||_1 + b_2) > 0$. Then we have

$$
\begin{aligned}
0 & \\
< \quad & \sigma(2k|U_2^{(1)}| - (0.5 - \gamma)||U_2||_1 + b_2) - \sigma(k|U_2^{(1)}| + (0.5 - \gamma)||U_2||_1 + b_2) \\
= \quad & (2k|U_2^{(1)}| - (0.5 - \gamma)||U_2||_1 + b_2) - \sigma(k|U_2^{(1)}| + (0.5 - \gamma)||U_2||_1 + b_2) \\
\le \quad & (2k|U_2^{(1)}| - (0.5 - \gamma)||U_2||_1 + b_2) - (k|U_2^{(1)}| + (0.5 - \gamma)||U_2||_1 + b_2) \\
= \quad & k|U_2^{(1)}| - (1 - 2\gamma)||U_2||_1.
\end{aligned}
$$

This is what we want.

(e4): $k|U_1^{(q)}| > (1 - 2\gamma)||U_1||_1$ for any $q \in [k]$.

Similar to (e3).

(e5): $J = \{x_i = \mathrm{Sgn}(U_1^{(i)})\}_{i=1}^k$ is the solution to the reversible 6-SAT problem $\varphi(k, m)$.

If not, as said in (d6), there is an $i \in [m]$ such that $q(\{\mathrm{Sgn}(U_1^{(w)})\}_{w=1}^k, \phi_i) = 6$. And there is a $j \in [k]$ such that $\overline{\phi}_j = \phi_i$.

Without loss of generality, we assume that the indexes of the six entries in $\phi_i$ are $1, 2, 3, 4, 5, 6$. By the definition of $x_{4k+i}$, we know that $U_1 x_{4k+i} = \frac{k}{4} \sum_{j=1}^6 |U_1^{(z)}|$, and by (e2), we know that $U_2 x_{4k+i} = -\frac{k}{4} \sum_{j=1}^6 |U_2^{(z)}|$. By the definition of $x_{4k+j}$, we know that $U_1 x_{4k+j} = -\frac{k}{4} \sum_{j=1}^6 |U_1^{(z)}|$, and by (e2), we know that $U_2 x_{4k+j} = \frac{k}{4} \sum_{j=1}^6 |U_2^{(z)}|$.

As said in (e3.2), we have $-2k|U_1^{(j)}| + (0.5 - \gamma)||U_1||_1 + b_1 < 0$ or $-2k|U_2^{(j)}| + (0.5 - \gamma)||U_2||_1 + b_2 < 0$ standing for any $z \in [k]$. Let the last stand for $z = 7$. If the first one stands, it is similar.

Now we will show that

$$
\begin{aligned}
& \sigma(2k|U_1^{(7)}| - (0.5 - \gamma)||U_1||_1 + b_1) \\
< \quad & \sigma(\tfrac{k}{4} \sum_{j=1}^6 |U_1^{(z)}| + (0.5 - \gamma)||U_1||_1 + b_1) + \sigma(-\sum_{j=1}^6 |U_2^{(z)}| + (0.5 - \gamma)||U_1||_1 + b_1) \quad (14) \\
< \quad & \sigma(2k|U_1^{(7)}| - (0.5 - \gamma)||U_1||_1 + b_1),
\end{aligned}
$$

which lead to a contradiction.

(e5.1): We prove that $\sigma(\tfrac{k}{4} \sum_{j=1}^6 |U_1^{(z)}| + (0.5 - \gamma)||U_1||_1 + b_1) + \sigma(-\tfrac{k}{4} \sum_{j=1}^6 |U_2^{(z)}| + (0.5 - \gamma)||U_1||_1 + b_1) < \sigma(2k|U_1^{(7)}| - (0.5 - \gamma)||U_1||_1 + b_1)$.

Let $h = 7$ if $U_1^{(7)} > 0$, and $h = k + 7$ if $U_1^{(7)} < 0$. Because $\mathcal{F}(x_{2k+h} - (0.5 - \gamma)\text{Sgn}(U_1)) - 1.5 > 0$ and $-2k|U_2^{(7)}| + (0.5 - \gamma)||U_2||_1 + b_2 - 1.5 < 0$, we have

$$
\begin{aligned}
& \sigma(U_1(x_{h+2k} - (0.5 - \gamma)\text{Sgn}(U_1)) + b_1) + \sigma(U_2(x_{h+2k} - (0.5 - \gamma)\text{Sgn}(U_1)) + b_2) + c \\
= \; & \sigma(2k|U_1^{(7)}| - (0.5 - \gamma)||U_1||_1 + b_1) + \sigma(-2k|U_2^{(7)}| + (0.5 - \gamma)||U_2||_1 + b_2) + c \quad (by \ (e2)) \\
= \; & \sigma(2k|U_1^{(7)}| - (0.5 - \gamma)||U_1||_1 + b_1) + c \\
> \; & 0.
\end{aligned}
\tag{15}
$$

Because $\mathcal{F}(x_{4k+i} + (0.5 - \gamma)\text{Sgn}(U_1)) - 1.5 < 0$, we have that:

$$
\begin{aligned}
& \sigma(U_1(x_{4k+i} + (0.5 - \gamma)\text{Sgn}(U_1)) + b_1) + \sigma(U_2(x_{4k+i} + (0.5 - \gamma)\text{Sgn}(U_1)) + b_2) + c \\
= \; & \sigma(\tfrac{k}{4}\sum_{j=1}^{6}|U_1^{(z)}| + (0.5 - \gamma)||U_1||_1 + b_1) + \sigma(-\sum_{j=1}^{6}|U_2^{(z)}| \\
& + (0.5 - \gamma)||U_1||_1 + b_1) + c \ (by \ (e2)) \\
< \; & 0.
\end{aligned}
\tag{16}
$$

By equation 15 and equation 16, it holds $\sigma(\tfrac{k}{4}\sum_{j=1}^{6}|U_1^{(z)}| + (0.5 - \gamma)||U_1||_1 + b_1) + \sigma(-\sum_{j=1}^{6}|U_2^{(z)}| + (0.5 - \gamma)||U_1||_1 + b_1) < \sigma(2k|U_1^{(7)}| - (0.5 - \gamma)||U_1||_1 + b_1)$. This is what we want.

(e5.2) We prove that $\sigma(\tfrac{k}{4}\sum_{j=1}^{6}|U_1^{(z)}| + (0.5 - \gamma)||U_1||_1 + b_1) + \sigma(-\sum_{j=1}^{6}|U_2^{(z)}| + (0.5 - \gamma)||U_1||_1 + b_1) > \sigma(2k|U_1^{(7)}| - (0.5 - \gamma)||U_1||_1 + b_1)$.

By (e4), we have that:

$$
\begin{aligned}
& 2k|U_1^{(7)}| - (0.5 - \gamma)||U_1||_1 \\
= \; & 2k(||U_1||_1 - \sum_{z \neq 7}|U_1^{(z)}|) - (0.5 - \gamma)||U_1||_1 \\
< \; & 2k(||U_1||_1 - (k-1)(1 - 2\gamma)||U_1||_1/k) - (0.5 - \gamma)||U_1||_1 \quad (by \ (e4)) \\
= \; & (1.5 + 4\gamma k - 3\gamma)||U_1||_1 \\
< \; & (1.5(1 - 2\gamma) + (0.5 - \gamma))||U_1||_1 \quad (by \ \gamma < 1/(10k)) \\
< \; & \tfrac{k}{4}\sum_{j=1}^{6}|U_1^{(z)}| + (0.5 - \gamma)||U_1||_1 \quad (by \ (e4)).
\end{aligned}
$$

So $\sigma(\tfrac{k}{4}\sum_{j=1}^{6}|U_1^{(z)}| + (0.5 - \gamma)||U_1||_1 + b_1) > \sigma(2k|U_1^{(7)}| - (0.5 - \gamma)||U_1||_1 + b_1)$. Then $\sigma(\tfrac{k}{4}\sum_{j=1}^{6}|U_1^{(z)}| + (0.5 - \gamma)||U_1||_1 + b_1) + \sigma(-\sum_{j=1}^{6}|U_2^{(z)}| + (0.5 - \gamma)||U_1||_1 + b_1) > \sigma(2k|U_1^{(7)}| - (0.5 - \gamma)||U_1||_1 + b_1)$. (e5.2) is proved.

From (e5.1) and (e5.2), the assumption is wrong and (e5) is proved. $\square$

### B.2 PROOF OF THEOREM 4.3

*Proof.* It suffices to show that there exists a dataset $\mathcal{D}$ such that if $\mathcal{F}$ has width less than $n$ and memorizes $\mathcal{D}$ (that is $\widehat{\mathcal{F}}(x) = y$ for $(x, y) \in \mathcal{D}$), then $\text{RA}_{\mathcal{D}}(\mathcal{F}, 0.4\lambda_{\mathcal{D}}) \leq 1 - \frac{1}{n+1}$; that is, $\mathcal{F}$ is not a robust memorization of $\mathcal{D}$ with budget $0.4\lambda_{\mathcal{D}}$.

Denote by $\mathbf{1}$ the vector all whose entries are 1 and $\mathbf{1}_k$ the vector whose $k$-th entry is 1 and all other entries are 0. Without loss of generality, let $N$ satisfy $(n + 1)|N$. We define a dataset $\mathcal{D} = \{x_i, y_i\}_{i=0}^{N-1}$ with separation bound 1 as follows:

(1) $x_0 = 0\mathbf{1}$ and $y_0 = 0$; $x_i = \mathbf{1}_i$ and $y_i = 1$ for $i \in [n]$;

(2) for $i = k(n+1), \ldots, k(n+1) + n$ and $k = 1, \ldots, \frac{N}{n+1} - 1$, $x_i = x_{\bar{i}} + \mathbf{1}$ and $y_i = y_{\bar{i}}$, where $\bar{i} = i$ mod $(n+1)$. It is easy to see that $\lambda_{\mathcal{D}} = 1$.

Let $\mathcal{F} : \mathbb{R}^n \to \mathbb{R}$ be a network that memorizes the dataset $\mathcal{D}$ defined above and let $W_1$ be the weight matrix of the first layer of $\mathcal{F}$. Then $W_1 \in \mathbb{R}^{K \times n}$. Since $\mathcal{F}$ has a width smaller than $n$, we have $K < n$. We will show that there exists an $s$ in $[n]$ such that $\exists \delta_0, \delta_s \in \mathbb{R}^n$, satisfying $||\delta_0||_\infty < 0.4, ||\delta_s||_\infty < 0.4$, and $W_1(x_0 + \delta_0) = W_1(x_s + \delta_s)$.

Since $K < n$, $W_1 \in \mathbb{R}^{K \times n}$ is not of full row rank and, therefore, there exists a vector $v \in \mathbb{R}^n$ such that $W_1 v = 0$ and $||v||_\infty = 1$. For such a $v$, let $|v^{(s)}| = 1$ for some $s \in [n]$. We define $\delta_0, \delta_s \in \mathbb{R}^n$ as follows:

$\delta_0^{(s)} = 1/3$ and $\delta_0^{(k)} = -v^{(s)}v^{(k)}/3$ for $k \neq s$;     $\delta_s^{(s)} = 0$ and $\delta_s^{(k)} = v^{(s)}v^{(k)}/3$ for $k \neq s$.

It is clear that $||\delta_0||_\infty = \frac{1}{3} < 0.4$ and $||\delta_s||_\infty = \frac{1}{3} < 0.4$. Also, $x_s + \delta_s - x_1 - \delta_0 = \frac{2}{3}v^{(s)}v$. Thus, $W_1(x_0 + \delta_0) - W_1(x_s + \delta_s) = W_1(x_0 + \delta_0 - x_s - \delta_s) = W_1(\frac{2}{3}v^{(s)}v) = 0$.

It is easy to see that for any $x, z \in \mathbb{R}^n$, $W_1 x = W_1 z$ implies $\mathcal{F}(x) = \mathcal{F}(z)$. Since $W_1(x_0 + \delta_0) = W_1(x_s + \delta_s)$, we have $\mathcal{F}(x_0 + \delta_0) = \mathcal{F}(x_s + \delta_s)$. Since $\mathcal{F}$ memorizes $\mathcal{D}$, we have $\widehat{\mathcal{F}}(x_0) = 0, \widehat{\mathcal{F}}(x_s) = 1$. Therefore, $\widehat{\mathcal{F}}(x_0 + \delta_0) \neq 0$ or $\widehat{\mathcal{F}}(x_s + \delta_s) \neq 1$ must be valid. In other words, $\mathcal{F}$ cannot be robust at $x_0$ or $x_s$ for the robust budget 0.4. Similarly, $\mathcal{F}$ cannot be robust for at least one point in $\{x_i\}_{i=k(n+1)}^{k(n+1)+n}$ for $k \in \{1, \ldots, \frac{N}{n+1} - 1\}$. In summary, $\mathcal{F}$ cannot be robust for at least $\frac{N}{n+1}$ points in $\mathcal{D}$, so $\mathrm{RA}_{\mathcal{D}}(\mathcal{F}, 0.4) \leq 1 - \frac{1}{n+1}$. $\square$

### B.3   PROOF OF PROPOSITION 4.5

*Proof.* It is easy to construct a dataset $\{x_i\}_{i=1}^N \subset \mathbb{R}^n$ such that $\cup_{i=2}^N \mathbb{B}_\infty(x_i, \mu) = \mathbb{B}_\infty(x_1, 2\mu + \lambda) \setminus \mathbb{B}_\infty(x_1, \lambda)$. Then, we let $\mathcal{D} = \{(x_1, 1)\} \cup \{(x_2, 2)\}_{i=2}^N$. It is easy to see that $\mathcal{D}$ satisfies $\lambda_{\mathcal{D}} \geq \lambda$, so the first part of the proposition is proved.

The rest of the proof is similar to that of Theorem 4.3, so we just give a sketch of the proof. Let $\mathcal{F} \in \mathbf{H}_{n,*,d,*}$ be a network and let $W_1$ be the weight matrix of the first layer of $\mathcal{F}$. If $d < n$, then $W_1$ is not of full row rank, so there exists a $v \in \mathbb{R}^n$ such that $\mathcal{F}(x) = \mathcal{F}(x + kv)$ for any $x \in \mathbb{R}^n$ and $k \in \mathbb{R}$. We take $x = x_1$ and $kv \in \mathbb{B}_\infty(0, 2\mu + \lambda) \setminus \mathbb{B}_\infty(0, \lambda)$, so $\mathcal{F}(x_1) = \mathcal{F}(x_1 + kv)$. If $\mathcal{F}$ is a robust memorization of $\mathcal{D}$ with budget $\mu$, then it must hold $|\mathcal{F}(x_1) - 1| < 0.5$ and $|\mathcal{F}(x_1 + kv) - 2| < 0.5$, because $\cup_{i=2}^N \mathbb{B}_\infty(x_i, \mu) = \mathbb{B}_\infty(x_1, 2\mu + \lambda) \setminus \mathbb{B}_\infty(x_1, \lambda)$, which is in contradiction to $\mathcal{F}(x_1) = \mathcal{F}(x_1 + kv)$. So $\mathcal{F}$ is not a robustness memorization of $\mathcal{F}$ with budget $\mu$. The proposition is proved. $\square$

### B.4   PROOF OF (2) OF PROPOSITION 4.6

*Proof.* Let us first consider $n = 1$. Let $\mathcal{D}$ be $\{(x_i, y_i)\}_{i=1}^N$, where $x_i = i \in \mathbb{R}^1$ and $y_1 = 1, y_2 = 3, y_3 = 5$; $y_i = 2$ if $i > 1$ and $i$ are even; otherwise $y_i = 4$. Let $\mu = 0.4$. It is easy to see that $\lambda_{\mathcal{D}} = 1 > 2\mu$. Let $\mathcal{F}(x) = U_2 \sigma(U_1 x + B_1) + b_2$ be a network with depth 2 and width $w$, where $U_1 = (u_1, \ldots, u_w)^\tau \in \mathbb{R}^{w \times 1}$, $B_1 = (b_1, \ldots, b_w)^\tau \in \mathbb{R}^{w \times 1}, U_2 \in \mathbb{R}^{1 \times w}, B_2 \in \mathbb{R}$.

It suffices to show that if $\mathcal{F}$ is a robust memorization of $\mathcal{D}$ with radius $\mu$, then $w > N$.

We will show that, for any $k \in \{4, 5, \ldots, N - 1\}$ there exist $i, j \in [w]$ and $i \neq j$ so that $u_i x + b_i = 0$ and $u_j x + b_j = 0$ have solutions in $(k, k+1)$. Also note that for different $k$, the corresponding $i, j$ must be different. Thus $w \geq 2(N - 4) > N$, which is what we want. The proof of such a conclusion is given below.

We prove only the case $k = 4$, and other cases can be proved similarly. We know that $\mathcal{F}(5 + \epsilon) \in (3.5, 4.5)$ and $\mathcal{F}(4 + \epsilon) \in (1.5, 2.5)$ when $\epsilon \in [-0.4, 0.4]$, because $\mathcal{F}$ is a robust memorization of $\mathcal{D}$. Thus, we have $\frac{\mathcal{F}(4.6) - \mathcal{F}(4.4)}{4.6 - 4.4} \geq \frac{3.5 - 2.5}{4.6 - 4.4} = 5$, so there exists an interval $A \subset (4.4, 4.6)$ such that the slope of $\mathcal{F}$ in $A$ is at least 5. We consider four cases.

**(c1):** The intervals $(4, 4.4)$ and $(4.6, 5)$ are linear regions of $\mathcal{F}$.

Since $|\frac{\mathcal{F}(4) - \mathcal{F}(4.4)}{4.4 - 4}| \leq |\frac{2.5 - 1.5}{4.4 - 4}| = 2.5$, the absolute value of the slope of $\mathcal{F}$ in $(4, 4.4)$ is at most 2.5. Similarly, the absolute value of the slope of $\mathcal{F}$ in $(4.6, 5)$ is at most 2.5.

Note that $A = (a_l, a_r) \subset (4.4, 4.6)$ and that the slope of $\mathcal{F}$ in $A$ is at least 5. Since $\mathcal{F}$ has the same slope in the same linear region, $A$ and $(4, 4.4)$ are not in the same linear region, which means that there exists an $i \in [w]$ such that the active states of $\sigma(u_i x + b_i)$ are different in $A$ and $(4, 4.4)$. In other words, $u_i x + b_i = 0$ has a solution in $(4.4, a_l)$. Similarly, there is a $j \in [w]$ such that $u_j x + b_j = 0$ has a solution in $(a_r, 4.6)$. This is what we want.

**(c2):** The interval $(4, 4.4)$ is a linear region of $\mathcal{F}$ and the interval $(4.6, 5)$ is not a linear region of $\mathcal{F}$.

Because $(4.6, 5)$ is not a linear region of $\mathcal{F}$, there must be a $j \in [w]$ that makes $u_j x + b_j = 0$ have a solution in $(4.6, 5)$. As proved in case c1, there exists an $i$ such that $u_i x + b_i = 0$ has a solution in $(4.4, a_l)$. This is what we want.

**(c3):** The interval $(4.6, 5)$ is a linear region of $\mathcal{F}$ and the interval $(4, 4.4)$ is not a linear region of $\mathcal{F}$. This case can be proved similar to case (c2).

**(c4):** Both the intervals $(4.6, 5)$ and $(4, 4.4)$ are not linear regions of $\mathcal{F}$. This case can be proved similar to case (c2).

For $n > 1$, we just need to take $x_i = (i, 0, 0, \ldots, 0) \in \mathbb{R}^n$, and the proof is the same. $\qquad\square$

### B.5 Proof of Theorem 4.8

*Proof.* It suffices to show that for any $\mu < 0.5\lambda_{\mathcal{D}}$, there exists a network with depth $2N + 1$, width $3n + 1$, and $O(Nn)$ non-zero parameters, which can robustly memorize $\mathcal{D}$ with robust budget $\mu$.

Let $\mathcal{D} = \{(x_i, y_i)\}_{i=1}^N \subset \mathbb{R}^n \times [L]$. Let $C \in \mathbb{R}_+$ satisfy $C > |x_i^{(j)}| + \mu > 0$ for all $i \in [N]$ and $j \in [n]$. $\mathcal{F}$ will be defined in three steps for an input $x \in \mathbb{R}^n$.

**Step 1**. The first layer has width $3n + 1$ and is used to check whether $x \in \mathbb{B}(x_1, \mu)$. Specifically, $x \in \mathbb{B}_{\infty}(x_1, \mu)$ if and only if $\mathcal{F}_1^j(x) = 0$ for all $j \in [2n]$. The second layer has width $n + 2$ and computes $E_1(x)$ in Property 2 given below. The two layers are given below.

> (1-1.1) $\mathcal{F}_1^0(x) = 0$;
>
> (1-1.2) $\mathcal{F}_1^j(x) = \sigma(x_1^{(j)} - x^{(j)} - \mu)$, $\mathcal{F}_1^{n+j}(x) = \sigma(x^{(j)} - x_1^{(j)} - \mu)$, where $j \in [n]$;
>
> (1-1.3) $\mathcal{F}_1^{2n+j}(x) = \sigma(x^{(j)} + C)$, where $j \in [n]$;
>
> (1-2.1) $\mathcal{F}_2^0(x) = 0$;
>
> (1-2.2) $\mathcal{F}_2^1(x) = \sigma(y_1 - \frac{y_1}{\lambda_{\mathcal{D}} - 2\mu} \sum_{k=1}^{2n} \mathcal{F}_1^k(x))$;
>
> (1-2.3) $\mathcal{F}_2^{j+1}(x) = \sigma(\mathcal{F}_1^{2n+j}(x))$, where $j \in [n]$.

**Step 2**. For $i = 2, 3, \ldots, N$, the $(2i-1)$-th layer has width $3n + 1$ and is used to check whether $x \in \mathbb{B}(x_i, \mu)$. Specifically, $x \in \mathbb{B}_\infty(x_i, \mu)$ if and only if $\mathcal{F}_{2i-1}^j(x) = 0$ for all $j \in [2n]$. The $2i$-th layer has width $n + 2$ and is used to calculate $E_i(x)$ in Property 2 given below.

(i-1.1) $\mathcal{F}_{2i-1}^0(x) = \sigma(\mathcal{F}_{2i-2}^0(x) + \mathcal{F}_{2i-2}^1(x))$;

(i-1.2) $\mathcal{F}_{2i-1}^j(x) = \sigma((x_i^{(j)} + C) - \mathcal{F}_{2i-2}^{j+1}(x) - \mu)$ and $\mathcal{F}_{2i-1}^{n+j}(x) = \sigma(\mathcal{F}_{2i-2}^{j+1}(x) - (x_i^{(j)} + C) - \mu)$, where $j \in [n]$;

(i-1.3) $\mathcal{F}_{2i-1}^{2n+j}(x) = \sigma(\mathcal{F}_{2i-2}^{j+1}(x))$, where $j \in [n]$;

(i-2.1) $\mathcal{F}_{2i}^0(x) = \sigma(\mathcal{F}_{2i-1}^0(x))$;

(i-2.2) $\mathcal{F}_{2i}^1(x) = \sigma(y_i - \frac{y_i}{\lambda_\mathcal{D}-2\mu}\sum_{k=1}^{2n}\mathcal{F}_{2i-1}^k(x) - \mathcal{F}_{2i-1}^0(x))$;

(i-2.3) $\mathcal{F}_{2i}^{j+1}(x) = \sigma(\mathcal{F}_{2i-1}^{2n+j}(x))$, where $j \in [n]$.

**Step 3**. The output layer is $\mathcal{F}(x) = \mathcal{F}_{2N}^0(x) + \mathcal{F}_{2N}^1(x)$.

Next, we will show that $\mathcal{F}$ has the following properties.

**Property 1**. $\mathcal{F}_{2i}^{j+1}(x) = x^{(j)} + C$ for $i \in [N]$, $j \in [n]$, and $x \in \mathbb{B}_\infty(x_i, \mu)$, that is, $\mathcal{F}_i^{2n+j}(x)$ for $j \in [n]$ are used to maintain the value $x^{(j)}$.

From (1-1.3) and (1-2.3), since $C + x_i^{(j)} > \mu > 0$ for all $i \in [N]$ and $j \in [n]$, we have $\mathcal{F}_2^{j+1}(x) = \mathcal{F}_1^{2n+j}(x) = \sigma(x^j + C) = x^j + C$. From (i-2.3) and (i-1.3), we have $\mathcal{F}_{2i}^{j+1}(x) = \sigma(\mathcal{F}_{2i-1}^{2n+j}(x)) = \sigma(\mathcal{F}_{2i-2}^{j+1}(x)) = \cdots = \sigma(\mathcal{F}_2^{j+1}(x)) = x^{(j)} + C$, for all $i \in [N]$ and $j \in [n]$. Property 1 is proved.

**Property 2**. Let $E_i(x) = y_i - \frac{y_i}{\lambda_\mathcal{D}-2\mu}\sum_{j=1}^{2n}\mathcal{F}_{2i-1}^j(x)$ for $i \in [N]$. Then $E_i(x) = y_i$ for $x \in \mathbb{B}_\infty(x_i, \mu)$, and $E_i(x) < y_i$ for $x \notin \mathbb{B}_\infty(x_i, \mu)$.

Due to Property 1, for $j \in [n]$, step (i-1.2) becomes

$$\mathcal{F}_{2i-1}^j(x) = \sigma((x_i^{(j)} + C) - \mathcal{F}_{2i-2}^{j+1}(x) - \mu) = \sigma(x_i^{(j)} - x^{(j)} - \mu)$$
$$\mathcal{F}_{2i-1}^{n+j}(x) = \sigma(\mathcal{F}_{2i-2}^{j+1}(x) - (x_i^{(j)} + C) - \mu) = \sigma(x^{(j)} - x_i^{(j)} - \mu).$$

Then $x \in \mathbb{B}_\infty(x_i, \mu)$ if and only if $\sigma(x_i^{(j)} - x^{(j)} - \mu) = \sigma(x^{(j)} - x_i^{(j)} - \mu) = 0$, or equivalently $\mathcal{F}_{2i-1}^j(x) = 0$ for $j \in [2n]$. Thus, $E_i(x) = y_i$ for $x \in \mathbb{B}_\infty(x_i, \mu)$. If $x \notin \mathbb{B}_\infty(x_i, \mu)$, then $||x_i - x - \mu||_\infty > 0$ or $||x - x_i - \mu||_\infty > 0$, which means that $\mathcal{F}_{2i-1}^j(x) > 0$ for at least one $j \in [2n]$. Since $\mathcal{F}_i^j(x) \geq 0$ for all $i$ and $j$, we have $E_i(x) < y_i$.

**Property 3**. If $x \in \mathbb{B}_\infty(x_k, \mu)$ for $y_k \neq y_i$, then $E_i(x) \leq 0$.

Since $x \in \mathbb{B}_\infty(x_k, \mu)$ and $y_k \neq y_i$, we have $||x_i - x - \mu||_\infty \geq \lambda_\mathcal{D} - 2\mu > 0$ or $||x - x_i - \mu||_\infty \geq \lambda_\mathcal{D} - 2\mu > 0$, because the separation bound is $\lambda_\mathcal{D}$. Then $\mathcal{F}_{2i-1}^j(x) \geq \lambda_\mathcal{D} - 2\mu$ for at least one $j \in [2n]$ and thus $E_i(x) \leq y_i - \frac{y_i}{\lambda_\mathcal{D}-2\mu}\mathcal{F}_{2i-1}^j(x) \leq y_i - \frac{y_i}{\lambda_\mathcal{D}-2\mu}(\lambda_\mathcal{D} - 2\mu) = 0$.

**Property 4**. $\mathcal{F}(x) = \max_{i \in [N]}\{E_i(x), 0\}$ for $x \in \mathbb{R}^n$.

Since $\max\{x, y\} = x + \sigma(y - x)$ for $x, y \in \mathbb{R}$ and $\mathcal{F}_i^j(x) \geq 0$ for all $i$ and $j$, we have that

$$\begin{aligned}
\sigma(\mathcal{F}_{2i}^0(x) + \mathcal{F}_{2i}^1(x)) \quad &= \mathcal{F}_{2i}^0(x) + \mathcal{F}_{2i}^1(x) \\
&= \sigma(\mathcal{F}_{2i-1}^0(x)) + \sigma(E_i(x) - \mathcal{F}_{2i-1}^0(x)) \\
&= \max\{\mathcal{F}_{2i-1}^0(x), E_i(x)\} \\
&= \max\{\sigma(\mathcal{F}_{2i-2}^0(x) + \mathcal{F}_{2i-2}^1(x)), E_i(x)\}.
\end{aligned}$$

Using the above equation repeatedly, we have $\mathcal{F}(x) = \sigma(\mathcal{F}_{2N}^0(x) + \mathcal{F}_{2N}^1(x)) = \max_{i=1}^N\{E_i(x), \mathcal{F}_2^0(x)\} = \max_{i=1}^N\{E_i(x), 0\}$.

We now show that $\mathcal{F}$ satisfies the conditions of the theorem. Let $x \in \mathbb{B}_\infty(x_s, \mu)$ be $s \in [N]$. By Property 2, $E_s(x) = y_s$; and if $i \neq s$ and $y_i = y_s$, then $E_i(x) < y_s$. By Property 3, if $y_i \neq y_s$, then $E_i(x) \leq 0$. By Property 4, $\mathcal{F}(x) = \max_{i\in[N]}\{E_i(x), 0\} = E_s(x) = y_s$; that is, $\mathcal{F}$ is robust at $x_s$ with budget $\mu$.

The network $\mathcal{F}$ has width $3n + 1$ and depth $2N$. We now estimate the number of non-zero parameters. For $i \in [N]$, constructions (i-1.1) and (i-2.1) need 3 parameters; (i-1.2) needs $8n$ parameters; (i-1.3) and (i-2.3) need $2n$ parameters; (i-2.2) need $2n + 2$ parameters. In total, $(N - 1)(12n + 5) + 2$ parameters are needed. Finally, $\mathcal{F}$ can clearly be constructed in polynomial time. $\qquad\square$

### B.6 MORE ON THE NUMBER OF PARAMETERS AND PROOF OF PROPOSITION 4.10

In this subsection, we give more detailed explanation on Remark 4.9.

**Definition B.2.** Let $\mathcal{P}_{n,N,L}^G$ be the minimum $K$ such that the hypothesis space $\mathbf{H} = \{\mathcal{F} : \mathbb{R}^n \to \mathbb{R} : \mathrm{para}(\mathcal{F}) = K\}$ is an optimal robust memorization for any dataset in $\mathcal{D}_{n,L,N}$. $\mathcal{P}_{n,N,L}^R$ can be defined similarly for the following *strict optimal robust memorization networks* that satisfy $\mathcal{F}(\overline{x}) = y_i$ for all $\overline{x} \in \mathbb{B}_\infty(x_i, \mu)$ and $i \in [N]$.

The upper bound given in the proof of Theorem 4.8 is for $\mathcal{P}_{n,N,L}^R$. Proposition 4.10 implies that $\mathcal{P}_{n,N,L}^R$ and $\mathcal{P}_{n,N,L}^G$ are essentially the same, as shown by the following proposition and Corollary B.4.

**Proposition B.3.** *We have that* $0 \leq \mathcal{P}_{n,N,L}^R - \mathcal{P}_{n,N,L}^G \leq 6L$.

*Proof.* It is easy to see that $\mathcal{P}_{n,N,L,\mu}^R \geq \mathcal{P}_{n,N,L,\mu}^G$, because if a network is a strict robust memorization, then it is also a robust memorization for the same dataset and the same $\mu$. This proves the left side of the inequality in Proposition B.3. The right side follows from Proposition 4.10, whose proof is given below. $\qquad\square$

By Theorem 4.8 and Li et al. (2022), we have that $O(\sqrt{Nn}) \leq \mathcal{P}_{n,N,L}^R \leq O(Nn)$. Since $L \ll Nn$ for most dataset, Proposition B.3 implies

**Corollary B.4.** $\mathcal{P}_{n,N,L}^R \simeq \mathcal{P}_{n,N,L}^G = O(Nn)$, *if* $L \ll Nn$.

We now prove Proposition 4.10.

*Proof.* Since $\mathcal{F}$ is a robust memorization of $\mathcal{D}$ with budget $\mu$, it holds that $\mathcal{F}(x) \in (y_i - 0.5, y_i + 0.5)$ for all $x \in \mathbb{B}_\infty(x_i, \mu)$ and $i \in [N]$. Since $\mathbb{B}_\infty(x_i, \mu)$ is a closed set, there exists an $\epsilon > 0$ such that $\mathcal{F}(x) \in [y_i - 0.5 + \epsilon, y_i + 0.5 - \epsilon]$ for all $x \in \mathbb{B}_\infty(x_i, \mu)$, $i \in [N]$, and a small $\epsilon \in \mathbb{R}_{>0}$.

We claim that there exists a network $\mathcal{G}$ with $\mathrm{depth}(\mathcal{G}) = 2$ and $\mathrm{width}(\mathcal{G}) = 2L$, such that $\mathcal{G}(k + v) = k$ for any $k \in [N]$ and $v \in [-0.5 + \epsilon, 0.5 - \epsilon]$. As a consequence, we have $\mathcal{G}(\sigma(\mathcal{F}(x))) = \mathcal{G}(\mathcal{F}(x)) = y_i$ for all $x \in \mathbb{B}_\infty(x_i, \mu)$ and $i \in [N]$, and the proposition follows.

$\mathcal{G}$ is given below

$$\mathcal{G}(x) = \frac{1}{2\epsilon} \sum_{i=1}^{L} (\sigma(x - 0.5 - i + \epsilon) - \sigma(x - 0.5 - i - \epsilon)) + 1.$$

It is easy to see that:

(c1) $\sigma(x - 0.5 - i + \epsilon) - \sigma(x - 0.5 - i - \epsilon) = 0$ when $x \leq 0.5 + i - \epsilon$;

(c2) $\sigma(x - 0.5 - i + \epsilon) - \sigma(x - 0.5 - i - \epsilon) = 2\epsilon$ when $x \geq 0.5 + i + \epsilon$.

So for $k \in [L]$ and $x_0 \in [k - 0.5 + \epsilon, k + 0.5 - \epsilon]$, we have

(d1) $\sigma(x_0 - 0.5 - i + \epsilon) - \sigma(x_0 - 0.5 - i - \epsilon) = 0$ when $k \leq i$;

(d2) $\sigma(x_0 - 0.5 - i + \epsilon) - \sigma(x_0 - 0.5 - i - \epsilon) = 2\epsilon$ when $k \geq i + 1$;

and thus

$$\begin{aligned}
&\mathcal{G}(x_0) \\
&= \tfrac{1}{2\epsilon} \sum_{i=1}^{L} (\sigma(x_0 - 0.5 - i + \epsilon) - \sigma(x_0 - 0.5 - i - \epsilon)) + 1 \\
&= \tfrac{1}{2\epsilon} \sum_{i=1}^{k-1} (\sigma(x_0 - 0.5 - i + \epsilon) - \sigma(x_0 - 0.5 - i - \epsilon)) + 1 \\
&= \tfrac{1}{2\epsilon} \sum_{i=1}^{k-1} (2\epsilon) + 1 \\
&= k - 1 + 1 = k.
\end{aligned}$$

Thus, $\mathcal{G}(k + v) = k$ for any $k \in [N]$ and $v \in [-0.5 + \epsilon, 0.5 - \epsilon]$. The claim and hence the proposition is proved. $\qquad\square$

## B.7 OPTIMAL ROBUST MEMORIZATION FOR POSITIVE NORM

In this section, we compute the optimal robust memorization networks for the $L_p$ norm with $p \geq 1$, that is, we extend Theorem 4.8 from $\infty$-norm to $L_p$ norm.

For any $p > 1$, $\mu > 0$, and $x \in \mathbb{R}^n$, let $\mathbb{B}_p(x, \mu) = \{\overline{x} \in \mathbb{R}^n : ||x - \overline{x}||_p = (\sum_{i=1}^{n} |x_i - \overline{x}_i|^p)^{1/p} \leq \mu\}$. The *robust accuracy* of a network $\mathcal{F}$ on $\mathcal{D}$ with respect to a given *robust budget* $\mu \in \mathbb{R}_+$ is

$$\mathrm{RA}_{\mathcal{D}}^p(\mathcal{F}, \mu) = \mathbb{P}_{(x,y) \sim \mathcal{D}}(\forall \tilde{x} \in \mathbb{B}_p(x, \mu), |\mathcal{F}(\tilde{x}) - y| < 0.5).$$

The *p-separation bound* for a dataset $\mathcal{D}$ is defined to be

$$\lambda_{\mathcal{D}}^p = \min\{||x_i - x_j||_p : (x_i, y_i), (x_j, y_j) \in \mathcal{D} \text{ and } y_i \neq y_j\}.$$

The problem of **$p$-robust memorization for a given dataset** $\mathcal{D} \in \mathcal{D}_{n,L,N}$ **with budget** $\mu$ is to construct a network $\mathcal{F} : \mathbb{R}^n \to \mathbb{R}$ satisfying $\mathrm{RA}_{\mathcal{D}}^p(\mathcal{F}, \mu) = 1$. A network hypothesis space $\mathbf{H}$ is said to be an $p$-**optimal robust memorization** for a dataset $\mathcal{D}$, if for any $\mu < \lambda_{\mathcal{D}}^p/2$, there exists an $\mathcal{F} \in \mathbf{H}$ such that $\mathrm{RA}_{\mathcal{D}}^p(\mathcal{F}, \mu) = 1$. Then, we have:

**Theorem B.5.** *For any dataset $\mathcal{D} \in \mathcal{D}_{n,L,N}$, the hypothesis space $\mathbf{H}_{n,O(N),O(n),O(Nn)}$ is a 1-optimal robust memorization for $\mathcal{D}$.*

**Proof Sketch**. The proof is similar to that of Theorem 4.8.

**Step 1:** For any $i \in [N]$, calculate $|x^{(j)} - x_i^{(j)}|$ for all $j \in [n]$ at first and then calculate $|x^{(j)} - x_i^{(j)}|$ for all $j \in [n]$. Let $\mu$ be a given robustness radius and $\mu < \lambda_{\mathcal{D}}^1/2$.

**Step 2:** Calculate $\sum_{j=1}^{n} |x^{(j)} - x_i^{(j)}| - (\lambda_{\mathcal{D}}^1/2 + \mu)/2$.

**Step 3:** Use $\max_{i\in[N]}\{y_i - \frac{4}{3\lambda_\mathcal{D}^1-6\mu}L\sigma(\sum_{j=1}^n|x^{(j)}-x_i^{(j)}|-(\lambda_\mathcal{D}^1/2+\mu)/2)\}$ as the label of $x$.

Let $x\in\mathbb{B}_1(x_w,\mu)$. Since $\mu<\lambda_\mathcal{D}^1/2$, we have $y_w - \frac{4}{3\lambda_\mathcal{D}^1-6\mu}L\sigma(\sum_{j=1}^n|x^{(j)}-x_w^{(j)}|-(\lambda_\mathcal{D}^1/2+\mu)/2)\geq y_w - \frac{4}{3\lambda_\mathcal{D}^1-6\mu}L\sigma(\mu-(\lambda_\mathcal{D}^1/2+\mu)/2)=y_w$.

For all $y_j\neq y_w$, there must be $||x-x_j||_1\geq||x_j-x_w||_1-||x_w-x||_1\geq\lambda_\mathcal{D}^1-\mu$, so

$$
\begin{aligned}
& y_j - \frac{4}{3\lambda_\mathcal{D}^1-6\mu}L\sigma(\textstyle\sum_{j=1}^n|x^{(j)}-x_j^{(j)}|-(\lambda_\mathcal{D}^1/2+\mu)/2)\\
\leq\ & y_j - \frac{4}{3\lambda_\mathcal{D}^1-6\mu}L\sigma(\lambda_\mathcal{D}^1-\mu-(\lambda_\mathcal{D}^1/2+\mu)/2)\\
=\ & y_j - \frac{4}{3\lambda_\mathcal{D}^1-6\mu}L\sigma(3\lambda_\mathcal{D}^1/4-3\mu/2)\\
=\ & y_j - \frac{4}{3\lambda_\mathcal{D}^1-6\mu}L(3\lambda_\mathcal{D}^1/4-3\mu/2)\\
=\ & y_j - L\\
\leq\ & 0
\end{aligned}
$$

We thus have $y_w = y_w - \frac{4}{3\lambda_\mathcal{D}^1-6\mu}L\sigma(\sum_{j=1}^n|x^{(j)}-x_w^{(j)}|-(\lambda_\mathcal{D}^1/2+\mu)/2) \leq \max_{i\in[N]}\{y_i - \frac{4}{3\lambda_\mathcal{D}^1-6\mu}L\sigma(\sum_{j=1}^n|x^{(j)}-x_i^{(j)}|-(\lambda_\mathcal{D}^1/2+\mu)/2)\} = \max_{i\in[N],y_i=y_w}\{y_i - \frac{4}{3\lambda_\mathcal{D}^1-6\mu}L\sigma(\sum_{j=1}^n|x^{(j)}-x_i^{(j)}|-(\lambda_\mathcal{D}^1/2+\mu)/2)\} \leq \max_{i\in[N],y_i=y_w}\{y_i\}=y_w$, which means $\max_{i\in[N]}\{y_i-\frac{4}{3\lambda_\mathcal{D}^1-6\mu}L\sigma(\sum_{j=1}^n|x^{(j)}-x_i^{(j)}|-(\lambda_\mathcal{D}^1/2+\mu)/2)\}=y_w$. The theorem is proved.

**Theorem B.6.** *For any dataset $\mathcal{D}\in\mathcal{D}_{n,L,N}$ satisfies $D\subset[-\Delta,\Delta]^n$ where $\Delta\geq1$, $p\in\mathbb{N}_{>1}$, and $\lambda_\mathcal{D}^p/2>\gamma>0$, there is a network with width $O(n)$, depth $O(Np(\log(\frac{n}{\gamma^p})+p\log\Delta+\log p))$, and $O(Nnp(\log(n/\gamma^p)+p\log\Delta+\log p))$ parameters, which is a $p$-robust memorization for $\mathcal{D}$ with radius $\lambda_\mathcal{D}^p/2-\gamma$.*

The proof of this theorem needs the following lemma.

**Lemma B.7** (Proposition 3.5 of Elbrächter et al. (2021)). *For all $p\in\mathbb{N}^+$, $\Delta\geq1$, and $\epsilon<0.5$, there is a network $\mathcal{G}:\mathbb{R}\to\mathbb{R}$ with width 9 and depth $O(p(\log(1/\epsilon)+p\log(\Delta)+\log(p)))$ such that $|\mathcal{G}(x)-x^p|\leq\epsilon$ for all $x\in[-\Delta,\Delta]$.*

**Proof Sketch** The proof is similar to that of Theorem 4.8.

**Step 1:** For any $i\in[N]$, calculate $|x^{(j)}-x_i^{(j)}|$ for all $j\in[n]$ at first and then calculate $\mathcal{G}(|x^{(j)}-x_i^{(j)}|)$ for all $j\in[n]$, where $\mathcal{G}(x)$ is obtained from Lemma B.7 and satisfies $|\mathcal{G}(x)-x^p|\leq\epsilon$ for all $x\in[-10\Delta,10\Delta]$ and $\epsilon=\frac{(\lambda_\mathcal{D}^p/2-\gamma/3)^p-(\lambda_\mathcal{D}^p/2-\gamma)^p}{n}$. Since $\frac{(\lambda_\mathcal{D}^p/2-\gamma/3)^p-(\lambda_\mathcal{D}^p/2-\gamma)^p}{n}\geq\frac{(2/3\gamma)^p}{n}$, by Lemma B.7, $\mathcal{G}$ has depth $O(p(\log(\frac{n}{\gamma^p})+p\log(\Delta)+\log(p)))$.

**Step 2:** Calculate $\sum_{j=1}^n\mathcal{G}(|x^{(j)}-x_i^{(j)}|)-(\lambda_\mathcal{D}^p/2-\gamma/3)^p$. We will show that if $||x-x_i||_p\leq\lambda_\mathcal{D}^p/2-\gamma$, then $\sum_{j=1}^n\mathcal{G}(|x^{(j)}-x_i^{(j)}|)-(\lambda_\mathcal{D}^p/2-\gamma/3)^p\leq0$; if $||x-x_i||_p\geq\lambda_\mathcal{D}^p/2+\gamma$, then $\sum_{j=1}^n\mathcal{G}(|x^{(j)}-x_i^{(j)}|)-(\lambda_\mathcal{D}^p/2-\gamma/3)^p\geq2(\frac{1}{3\gamma})^p$. Thus, Step 2 follows from (2.1) and (2.2) to be proved in the following.

(2.1): Assume $||x-x_i||_p\leq\lambda_\mathcal{D}^p/2-\gamma$. Since $\sum_{i=1}^n\mathcal{G}(|x-x_i|)-n\epsilon\leq||x-x_i||_p^p$, we have that

$$
\sum_{j=1}^n\mathcal{G}(|x^{(j)}-x_i^{(j)}|)-(\lambda_\mathcal{D}^p/2-\gamma/3)^p\leq||x-x_i||_p^p+n\epsilon-(\lambda_\mathcal{D}^p/2-\gamma/3)^p\leq(\lambda_\mathcal{D}^p/2-\gamma)^p+n\epsilon-(\lambda_\mathcal{D}^p/2-\gamma/3)=0.
$$

(2.2): Assume $||x - x_i||_p \geq \lambda_{\mathcal{D}}^p/2 + \gamma$. Since $\sum_{i=1}^{n} \mathcal{G}(|x - x_i|) + n\epsilon \geq ||x - x_i||_p^p$, we have

$$
\begin{aligned}
& \sum_{j=1}^{n} \mathcal{G}(|x^{(j)} - x_i^{(j)}|) - (\lambda_{\mathcal{D}}^p/2 - \gamma/3)^p \\
\geq \quad & ||x - x_i||_p^p - n\epsilon - (\lambda_{\mathcal{D}}^p/2 - \gamma/3)^p \\
\geq \quad & (\lambda_{\mathcal{D}}^p/2 + \gamma)^p - n\epsilon - (\lambda_{\mathcal{D}}^p/2 - \gamma/3)^p \\
= \quad & (\lambda_{\mathcal{D}}^p/2 + \gamma)^p + (\lambda_{\mathcal{D}}^p/2 - \gamma)^p - 2(\lambda_{\mathcal{D}}^p/2 - \gamma/3)^p \\
\geq \quad & 2(\lambda_{\mathcal{D}}^p/2)^p - 2(\lambda_{\mathcal{D}}^p/2 - \gamma/3)^p \\
\geq \quad & 2(\tfrac{1}{3}\gamma)^p.
\end{aligned}
$$

**Step 3:** Use $\max_{i \in [N]}\{y_i - (3\gamma)^p L\sigma(\sum_{j=1}^{n} \mathcal{G}(|x^{(j)} - x_i^{(j)}|) - (\lambda_{\mathcal{D}}^p/2 - \gamma/3)^p)\}$ as the label of $x$.

Let $x \in \mathbb{B}_p(x_w, \lambda_{\mathcal{D}}^p/2 - \gamma)$. By (2.1), we have $y_w - (3\gamma)^p L\sigma(\sum_{j=1}^{n} \mathcal{G}(|x^{(j)} - x_w^{(j)}|) - (\lambda_{\mathcal{D}}^p/2 - \gamma/3)^p) = y_w$; and for all $y_j \neq y_w$, there must be $||x - x_j||_p \geq ||x_j - x_w||_p - ||x_w - x||_p \geq \lambda_{\mathcal{D}}^p/2 + \gamma$, so by (2.2), we have $y_j - (3\gamma)^p L\sigma(\sum_{j=1}^{n} \mathcal{G}(|x^{(j)} - x_j^{(j)}|) - (\lambda_{\mathcal{D}}^p/2 - \gamma/3)^p) = 0$.

Therefore, $y_w = y_w - (3\gamma)^p L\sigma(\sum_{j=1}^{n} \mathcal{G}(|x^{(j)} - x_w^{(j)}|) - (\lambda_{\mathcal{D}}^p/2 - \gamma/3)^p) \leq \max_{i \in [N]}\{y_i - (3\gamma)^p L\sigma(\sum_{j=1}^{n} \mathcal{G}(|x^{(j)} - x_i^{(j)}|) - (\lambda_{\mathcal{D}}^p/2 - \gamma/3)^p)\} = \max_{i \in [N], y_i = y_w}\{y_i - (3\gamma)^p L\sigma(\sum_{j=1}^{n} \mathcal{G}(|x^{(j)} - x_i^{(j)}|) - (\lambda_{\mathcal{D}}^p/2 - \gamma/3)^p)\} \leq \max_{i \in [N], y_i = y_w}\{y_i\} = y_w$, which means $\max_{i \in [N]}\{y_i - (3\gamma)^p L\sigma(\sum_{j=1}^{n} \mathcal{G}(|x^{(j)} - x_i^{(j)}|) - (\lambda_{\mathcal{D}}^p/2 - \gamma/3)^p)\} = y_w$, and the theorem is proved.

*Remark* B.8. When $p = 2$ and $\gamma = \lambda_{\mathcal{D}}/4$, our bound for the number of parameters in Theorem B.6 becomes $O(Nn \log(n/\lambda_{\mathcal{D}}))$. The result in (Li et al., 2022) is $O(Nn \log(n/\lambda_{\mathcal{D}}) + N\text{poly} \log(N/\lambda_{\mathcal{D}}))$. Our result is better.

## C  PROOFS FOR SECTION 5

### C.1  A LEMMA

The following lemma was given in Li et al. (2022), but without explicit information on width and depth, so we give an explicit construction.

**Lemma C.1.** *There exists a network $\mathcal{F} \in \mathbf{H}_{n, 2\log n, O(n), O(n)}$ such that $\mathcal{F}(x) = ||x||_\infty$; that is, there exists a network $\mathcal{F} : \mathbb{R}^n \to \mathbb{R}$ with depth $2\log n$, width $O(n)$, and $O(n)$ non-zero parameters such that $\mathcal{F}(x) = ||x||_\infty$.*

*Proof.* Let $e = \lceil \log_2 n \rceil$. Without loss of generality, we assume that $n = 2^e$. Then $\mathcal{F}$ has depth $2e$ and for $i \in [e+1]$, the $(2i-1)$-th layer has width $2^{e-i+2}$, and the $2i$-th layer has width $2^{e-i+1}$.

Denote $W_i$ and $b_i$ as the weight matrix and the bias of the $i$-th layer of $\mathcal{F}$. The first and second layers will change $x$ to $|x|$. The first layer has width $2^{e+1}$ and the second layer has width $2^e$, which are defined below.

$W_1^{2i,i} = 1$ and $W_1^{2i+1,i} = -1$; other entries of $W_1$ are 0. $b_1 = 0$.

$W_2^{i,2i} = 1$ and $W_2^{i,2i+1} = 1$; other entries of $W_2$ are 0. $b_2 = 0$.

Since $\sigma(x) + \sigma(-x) = |x|$ for any $x \in \mathbb{R}$, it is easy to check that $\mathcal{F}_2(x) = \sigma(W_2\sigma(W_1 x)) = |x|$.

For $i \in [e]$, the $(2i+1)$-th and the $(2i+2)$-th layers are defined below.

$\mathcal{F}_{2i+1}^{2m}(x) = \sigma(\mathcal{F}_{2i}^{2m}(x))$, where $m = 0, 1, \ldots, 2^{e-i} - 1$.

$\mathcal{F}_{2i+1}^{2m+1}(x) = \sigma(\mathcal{F}_{2i}^{2m+1}(x) - \mathcal{F}_{2i}^{2m}(x))$, where $m = 0, 1, \ldots, 2^{e-i} - 1$.

$\mathcal{F}_{2i+2}^{m}(x) = \sigma(\mathcal{F}_{2i+1}^{2m}(x) + \mathcal{F}_{2i+1}^{2m+1}(x))$, where $m = 0, 1, \ldots, 2^{e-i} - 1$.

For $i \in [e+1]$, using $\sigma(x - y) + y = \max\{x, y\}$ for any $x, y \in \mathbb{R}$, we have that

$$
\begin{aligned}
& \mathcal{F}_{2i+2}^{m}(x) \\
=\ & \sigma(\mathcal{F}_{2i+1}^{2m}(x) + \mathcal{F}_{2i+1}^{2m+1}(x)) \\
=\ & \mathcal{F}_{2i+1}^{2m}(x) + \mathcal{F}_{2i+1}^{2m+1}(x) \\
=\ & \sigma(\mathcal{F}_{2i}^{2m}(x)) + \sigma(\mathcal{F}_{2i}^{2m+1}(x) - \mathcal{F}_{2i}^{2m}(x)) \\
=\ & \mathcal{F}_{2i}^{2m}(x) + \sigma(\mathcal{F}_{2i}^{2m+1}(x) - \mathcal{F}_{2i}^{2m}(x)) \\
=\ & \max\{\mathcal{F}_{2i}^{2m}(x), \mathcal{F}_{2i}^{2m+1}(x)\}.
\end{aligned}
$$

The $(2e+2)$-th layer has width $1$ and is the output

$$
\begin{aligned}
\mathcal{F}(x) =\ & \mathcal{F}_{2e+2}^{1}(x) \\
=\ & \max\{\mathcal{F}_{2e}^{2}(x), \mathcal{F}_{2e}^{1}(x)\} \\
=\ & \max\{\mathcal{F}_{2e-2}^{4}(x), \mathcal{F}_{2e-2}^{3}(x), \mathcal{F}_{2e-2}^{2}(x), , \mathcal{F}_{2e-2}^{1}(x)\} \\
=\ & \ldots \\
=\ & \max\{\mathcal{F}_{2}^{2^{e}}(x), \mathcal{F}_{2}^{2^{e}-1}(x), \ldots, \mathcal{F}_{2}^{2}(x), , \mathcal{F}_{2}^{1}(x)\} \\
=\ & \|x\|_{\infty}.
\end{aligned}
$$

We now estimate the number of parameters. The first two layers need $4d$ non-zero parameters. For $i \in [e]$, the $(2i+1)$-th layer and the $(2i+2)$-th layer need $5 \cdot 2^{e-i}$ parameters. Therefore, we need $\sum_{i=1}^{e} 5 \cdot 2^{e-i} = O(2^e) = O(n)$ parameters. Then the lemma is proved. $\qquad\square$

## C.2 PROOF OF THEOREM 5.2

*Proof.* Let $\mathcal{D}$ be defined in equation 3 and $C \in \mathbb{R}_+$ satisfy $C + x_i^{(k)} - 0.5\lambda_{\mathcal{D}} > 0$ for all $i \in [N]$, $k \in [n]$. The network has $N(2\lceil \log(n) \rceil + 5) + 1$ hidden layers which will be defined below.

**Step 1.** The first layer has width $n + 1$: $\mathcal{F}_1^0(x) = 2$ and $\mathcal{F}_1^j(x) = \sigma(x^{(j)} + C) = x^{(j)} + C$, where $j \in [n]$.

**Step 2.** For $k \in [N]$, let $s_k = (2\lceil \log(n) \rceil + 5)(k-1) + 2$ and we will use the $s_k$-th layer to the $(s_k + 2\lceil \log(n) \rceil + 4)$-th layer to check if $\|x - x_k\|_{\infty} < 0.5\lambda_{\mathcal{D}}$. Step 2 consists of three sub-steps.

**Step 2a.** We use the $s_k$-th layer and the $(s_k + 1)$-th layer to calculate $|x - x_k|$. The $s_k$-th layer has width $3n + 1$ and is defined below.

$\mathcal{F}_{s_k}^0(x) = \sigma(\mathcal{F}_{s_k-1}^0(x))$;

$\mathcal{F}_{s_k}^j(x) = \sigma(\mathcal{F}_{s_k-1}^j(x) - x_k^{(j)} - C)$, where $j \in [n]$;

$\mathcal{F}_{s_k}^{n+j}(x) = \sigma(-\mathcal{F}_{s_k-1}^j(x) + x_k^{(j)} + C)$, where $j \in [n]$;

$\mathcal{F}_{s_k}^{2n+j}(x) = \sigma(\mathcal{F}_{s_k-1}^j(x))$, where $j \in [n]$.

The $(s_k + 1)$-th layer has width $2n + 1$ and is defined below.

$$\mathcal{F}^0_{s_k+1}(x) = \sigma(\mathcal{F}^0_{s_k}(x));$$

$$\mathcal{F}^j_{s_k+1}(x) = \sigma(\mathcal{F}^j_{s_k}(x) + \mathcal{F}^{n+j}_{s_k}(x)), \text{ where } j \in [n];$$

$$\mathcal{F}^{n+j}_{s_k+1}(x) = \sigma(\mathcal{F}^{2n+j}_{s_k}(x)), \text{ where } j \in [n].$$

The $s_k$-th layer needs $5n+1$ non-zeros parameters and $(s_k+1)$-th layer needs $3n+1$ non-zeros parameters.

**Step 2b.** Lemma C.1 is used to calculate $||x - x_k||_\infty$. According to Lemma C.1, there exists a network $\mathcal{H} : \mathbb{R}^n \to \mathbb{R}$ with $2\lceil \log(n) \rceil$ hidden layers, width $O(n)$, and $O(n)$ non-zero parameters to compute $\mathcal{H}(x) = ||x||_\infty$ for $x \in \mathbb{R}^n$. Since $\mathcal{H}$ has $2\lceil \log(n) \rceil$ hidden layers, we set the output of the $(s_k + 2\lceil \log(n) \rceil + 1)$-th layer to be

$$\mathcal{F}^0_{s_k+2\lceil \log(n) \rceil +1}(x) = \sigma(\mathcal{F}^0_{s_k+1}(x));$$

$$\mathcal{F}^1_{s_k+2\lceil \log(n) \rceil +1}(x) = \mathcal{H}(\mathcal{F}^1_{s_k+1}(x), \ldots, \mathcal{F}^n_{s_k+1}(x)) = ||\mathcal{F}_{s_k+1}(x)||_\infty;$$

$$\mathcal{F}^{j+1}_{s_k+2\lceil \log(n) \rceil +1}(x) = \sigma(\mathcal{F}^{n+j}_{s_k+1}(x)), \text{ where } j \in [n].$$

**Step 2c.** Use the $(s_k+2\lceil \log(n) \rceil +2)$-th to the $(s_k+2\lceil \log(n) \rceil +4)$-th layers to check if $||x - x_k||_\infty < 0.5\lambda_\mathcal{D}$. The $(s_k + 2\lceil \log(n) \rceil + 2)$-th layer has width $n + 4$ and is defined below.

$$\mathcal{F}^0_{s_k+2\lceil \log(n) \rceil +2}(x) = \sigma(\mathcal{F}^0_{s_k+2\lceil \log(n) \rceil +1}(x));$$

$$\mathcal{F}^1_{s_k+2\lceil \log(n) \rceil +2}(x) = \sigma(-\tfrac{2}{\lambda_\mathcal{D}}\mathcal{F}^1_{s_k+2\lceil \log(n) \rceil +1}(x) + 1);$$

$$\mathcal{F}^2_{s_k+2\lceil \log(n) \rceil +2}(x) = \sigma(\mathcal{F}^0_{s_k+2\lceil \log(n) \rceil +1}(x) - 2);$$

$$\mathcal{F}^3_{s_k+2\lceil \log(n) \rceil +2}(x) = \sigma(-\mathcal{F}^0_{s_k+2\lceil \log(n) \rceil +1}(x) + 2);$$

$$\mathcal{F}^{j+3}_{s_k+2\lceil \log(n) \rceil +2}(x) = \sigma(\mathcal{F}^{j+1}_{s_k+2\lceil \log(n) \rceil +1}(x)), \text{ where } j \in [n].$$

The $(s_k + 2\lceil \log(n) \rceil + 3)$-th layer has width $n + 3$ and is defined below.

$$\mathcal{F}^0_{s_k+2\lceil \log(n) \rceil +3}(x) = \sigma(\mathcal{F}^0_{s_k+2\lceil \log(n) \rceil +2}(x) + y_k\mathcal{F}^1_{s_k+2\lceil \log(n) \rceil +2}(x));$$

$$\mathcal{F}^1_{s_k+2\lceil \log(n) \rceil +3}(x) = \sigma(\mathcal{F}^1_{s_k+2\lceil \log(n) \rceil +2});$$

$$\mathcal{F}^2_{s_k+2\lceil \log(n) \rceil +3}(x) = \sigma(\mathcal{F}^1_{s_k+2\lceil \log(n) \rceil +2} - (\mathcal{F}^2_{s_k+2\lceil \log(n) \rceil +2}(x) + \mathcal{F}^3_{s_k+2\lceil \log(n) \rceil +2}(x)));$$

$$\mathcal{F}^{j+2}_{s_k+2\lceil \log(n) \rceil +3}(x) = \sigma(\mathcal{F}^{j+3}_{s_k+2\lceil \log(n) \rceil +2}(x)), \text{ where } j \in [n].$$

The $(s_k + 2\lceil \log(n) \rceil + 4)$-th layer has width $n + 1$ and is defined as

$$\mathcal{F}^0_{s_k+2\lceil \log(n) \rceil +4}(x) = \sigma(\mathcal{F}^0_{s_k+2\lceil \log(n) \rceil +3}(x) - y_k(\mathcal{F}^1_{s_k+2\lceil \log(n) \rceil +3}(x) - \mathcal{F}^2_{s_k+2\lceil \log(n) \rceil +3}(x)));$$

$$\mathcal{F}^j_{s_k+2\lceil \log(n) \rceil +4}(x) = \sigma(\mathcal{F}^{j+2}_{s_k+2\lceil \log(n) \rceil +3}(x)), \text{ where } j \in [n].$$

It is easy to check that if $\mathcal{F}^j_{s_k+1}(x) = |x^{(j)} - x_k^{(j)}|$. Then

$$\mathcal{F}^1_{s_k+2\lceil \log(n) \rceil +2}(x) = \sigma(-\frac{2}{\lambda_\mathcal{D}}\mathcal{F}^1_{s_k+2\lceil \log(n) \rceil +1}(x) + 1) > 0$$

if and only if $||x - x_k||_\infty < 0.5\lambda_\mathcal{D}$. These three layers need $3n + 16$ non-zeros parameters.

**Step 3.** The output is $\mathcal{F}(x) = \mathcal{F}^0_{s_N+2\lceil \log(n) \rceil +4}(x) - 2$. The network $\mathcal{F}$ has width $O(n)$, depth $O(N\log(n))$, and $O(Nn\log(n))$ non-zeros parameters.

We now show that $\mathcal{F}$ satisfies the condition of the theorem; that is, $\mathcal{F}$ memorizes $\mathcal{D}$ and satisfies $\text{Lip}_\infty(\mathcal{F}) = \frac{2}{\lambda_\mathcal{D}}$. The proof will be given by proving four properties.

**Property 1.** $\mathcal{F}^j_{s_k-1}(x) = x^{(j)} + C$ for $j \in [n]$ and $k \in [N]$. When $k = 1$, $s_k - 1 = 1$. By Step 1, we have $\mathcal{F}^j_{s_1-1}(x) = \mathcal{F}^j_1(x) = x^{(j)} + C$. When $k > 1$, we have

$$
\begin{aligned}
\mathcal{F}^j_{s_{k+1}-1}(x) &= \sigma(\mathcal{F}^j_{s_k+2\lceil\log(n)\rceil+4}(x)) = \sigma(\mathcal{F}^{j+2}_{s_k+2\lceil\log(n)\rceil+3}(x)) \\
&= \sigma(\mathcal{F}^{j+3}_{s_k+2\lceil\log(n)\rceil+2}(x)) = \sigma(\mathcal{F}^{j+1}_{s_k+2\lceil\log(n)\rceil+1}(x)) \\
&= \sigma(\mathcal{F}^{n+j}_{s_k+1}(x)) = \sigma(\mathcal{F}^{2n+j}_{s_k}(x)) = \sigma(\mathcal{F}^j_{s_k-1}(x)) = \mathcal{F}^j_{s_k-1}(x).
\end{aligned}
$$

Then, $\mathcal{F}^j_{s_{k+1}-1}(x) = \mathcal{F}^j_{s_k-1}(x) = \cdots = \mathcal{F}^j_{s_1-1}(x) = \mathcal{F}^j_1(x) = x^{(j)} + C$.

**Property 2.** $\mathcal{F}^j_{s_k+1}(x) = |x^{(j)} - x^{(j)}_k|$ and $\mathcal{F}^1_{s_k+2\lceil\log(n)\rceil+1}(x) = ||x - x_k||_\infty$ for $j \in [n]$.

Since $\sigma(x) + \sigma(-x) = |x|$ for any $x \in \mathbb{R}$, from Step 2a, $\mathcal{F}^j_{s_k+1}(x) = |\mathcal{F}^j_{s_k-1}(x) - x^{(j)}_k - C|$ for $j \in [n]$. By Property 1, $\mathcal{F}^j_{s_k-1}(x) = x^{(j)} + C$ for $j \in [n]$. Then, $\mathcal{F}^j_{s_k+1}(x) = |x^{(j)} - x^{(j)}_k|$ for $j \in [n]$. From Step 2b, we have that $\mathcal{F}^1_{s_k+2\lceil\log(n)\rceil+1}(x) = ||x - x_k||_\infty$ for $j \in [n]$.

**Property 3.** $\mathcal{F}^0_{s_k+2\lceil\log(n)\rceil+4}(x) = 2 + y_{w_k}\sigma(1 - \frac{2}{\lambda_\mathcal{D}}||x - x_{w_k}||_\infty)$, where $w_k = \text{argmin}_{i\in[k]}||x - x_i||_\infty$.

We prove the property by induction on $k$. We first show that the statement is valid for $k = 1$. We have that $w_k = 1$ and $\mathcal{F}^0_{s_1+2\lceil\log(n)\rceil+2}(x) = \mathcal{F}^0_{s_1+2\lceil\log(n)\rceil+1}(x) = \mathcal{F}^0_{s_1+1}(x) = \mathcal{F}^0_{s_1}(x) = \mathcal{F}^0_{s_1-1}(x) = 2$. From Step 2c and Property 2,

$$
\begin{aligned}
&\mathcal{F}^0_{s_1+2\lceil\log(n)\rceil+3}(x) \\
&= \sigma(\mathcal{F}^0_{s_1+2\lceil\log(n)\rceil+2}(x) + y_1\mathcal{F}^1_{s_1+2\lceil\log(n)\rceil+2}(x)) \\
&= \sigma(2 + y_1\sigma(1 - \frac{2}{\lambda_\mathcal{D}}\mathcal{F}^1_{s_1+2\lceil\log(n)\rceil+1}(x))) \\
&= 2 + y_1\sigma(-\frac{2}{\lambda_\mathcal{D}}\mathcal{F}^1_{s_1+2\lceil\log(n)\rceil+1}(x) + 1) \\
&= 2 + y_1\sigma(1 - \frac{2}{\lambda_\mathcal{D}}||x - x_0||_\infty).
\end{aligned}
$$

Since $\mathcal{F}^2_{s_1+2\lceil\log(n)\rceil+2}(x) = \sigma(\mathcal{F}^0_{s_1+2\lceil\log(n)\rceil+1}(x) - 2) = \sigma(2 - 2) = 0$ and $\mathcal{F}^3_{s_1+2\lceil\log(n)\rceil+2}(x) = \sigma(-\mathcal{F}^0_{s_1+2\lceil\log(n)\rceil+1}(x) + 2) = \sigma(2 - 2) = 0$, we have $\mathcal{F}^2_{s_1+2\lceil\log(n)\rceil+3}(x) = \sigma(\mathcal{F}^1_{s_1+2\lceil\log(n)\rceil+2} - (\mathcal{F}^2_{s_1+2\lceil\log(n)\rceil+2}(x) + \mathcal{F}^3_{s_1+2\lceil\log(n)\rceil+2}(x))) = \sigma(\mathcal{F}^1_{s_1+2\lceil\log(n)\rceil+2}) = \mathcal{F}^1_{s_1+2\lceil\log(n)\rceil+3}$. Then

$$
\begin{aligned}
&\mathcal{F}^0_{s_1+2\lceil\log(n)\rceil+4}(x) \\
&= \sigma(\mathcal{F}^0_{s_1+2\lceil\log(n)\rceil+3}(x) - y_1(\mathcal{F}^1_{s_1+2\lceil\log(n)\rceil+3}(x) - \mathcal{F}^2_{s_1+2\lceil\log(n)\rceil+3}(x))) \\
&= \mathcal{F}^0_{s_1+2\lceil\log(n)\rceil+3}(x) \\
&= 2 + y_1\sigma(1 - \frac{2}{\lambda_\mathcal{D}}||x - x_0||_\infty).
\end{aligned}
$$

We have proved the statement for $k = 1$.

Assume that the statement is valid for $k - 1$; that is, $\mathcal{F}^0_{s_{k-1}+2\lceil\log(n)\rceil+4}(x) = 2 + y_{w_{k-1}}\sigma(1 - \frac{2}{\lambda_\mathcal{D}}||x - x_{w_{k-1}}||_\infty)$. We have $\mathcal{F}^0_{s_k+2\lceil\log(n)\rceil+2}(x) = \mathcal{F}^0_{s_k+2\lceil\log(n)\rceil+1}(x) = \mathcal{F}^0_{s_k+1}(x) = \mathcal{F}^0_{s_k}(x) = \mathcal{F}^0_{s_k-1}(x) = 2 + y_{w_{k-1}}\sigma(1 - \frac{2}{\lambda_\mathcal{D}}||x - x_{w_{k-1}}||_\infty) \geq 1$, and we also have $\mathcal{F}^1_{s_k+2\lceil\log(n)\rceil+2}(x) = \sigma(-\frac{2}{\lambda_\mathcal{D}}\mathcal{F}^1_{s_k+2\lceil\log(n)\rceil+1}(x) +$

$1) \leq 1$. Then

$$
\begin{aligned}
&\mathcal{F}^0_{s_k+2\lceil\log(n)\rceil+3}(x)\\
=~& \sigma(\mathcal{F}^0_{s_k+2\lceil\log(n)\rceil+2}(x) + y_k\mathcal{F}^1_{s_k+2\lceil\log(n)\rceil+2}(x))\\
=~& \sigma(\mathcal{F}^0_{s_k+2\lceil\log(n)\rceil+2}(x) + y_k\sigma(1 - \tfrac{2}{\lambda_\mathcal{D}}\mathcal{F}^1_{s_k+2\lceil\log(n)\rceil+1}(x)))\\
=~& \mathcal{F}^0_{s_k+2\lceil\log(n)\rceil+2}(x) + y_k\sigma(1 - \tfrac{2}{\lambda_\mathcal{D}}\mathcal{F}^1_{s_k+2\lceil\log(n)\rceil+1}(x))\\
=~& \mathcal{F}^0_{s_k-1}(x) + y_k\sigma(1 - \tfrac{2}{\lambda_\mathcal{D}}\mathcal{F}^1_{s_k+2\lceil\log(n)\rceil+1}(x))\\
=~& \mathcal{F}^0_{s_k-1}(x) + y_k\mathcal{F}^1_{s_k+2\lceil\log(n)\rceil+2}(x).
\end{aligned}
\tag{17}
$$

Since $\mathcal{F}^2_{s_k+2\lceil\log(n)\rceil+2}(x) = \sigma(\mathcal{F}^0_{s_k+2\lceil\log(n)\rceil+1}(x) - 2)$ and $\mathcal{F}^3_{s_k+2\lceil\log(n)\rceil+2}(x) = \sigma(-\mathcal{F}^0_{s_k+2\lceil\log(n)\rceil+1}(x) + 2)$, we have

$$
\begin{aligned}
&\mathcal{F}^2_{s_k+2\lceil\log(n)\rceil+3}(x)\\
=~& \sigma(\mathcal{F}^1_{s_k+2\lceil\log(n)\rceil+2} - (\mathcal{F}^2_{s_k+2\lceil\log(n)\rceil+2}(x) + \mathcal{F}^3_{s_k+2\lceil\log(n)\rceil+2}(x)))\\
=~& \sigma(\mathcal{F}^1_{s_k+2\lceil\log(n)\rceil+2} - |\mathcal{F}^0_{s_k+2\lceil\log(n)\rceil+1}(x) - 2|).
\end{aligned}
$$

Then

$$
\begin{aligned}
&\mathcal{F}^0_{s_k+2\lceil\log(n)\rceil+4}(x)\\
=~& \sigma(\mathcal{F}^0_{s_k+2\lceil\log(n)\rceil+3}(x) - y_k(\mathcal{F}^1_{s_k+2\lceil\log(n)\rceil+3}(x) - \mathcal{F}^2_{s_k+2\lceil\log(n)\rceil+3}(x)))\\
=~& \sigma(\mathcal{F}^0_{s_k-1}(x) + y_k\mathcal{F}^1_{s_k+2\lceil\log(n)\rceil+2}(x)\\
&\quad -y_k(\mathcal{F}^1_{s_k+2\lceil\log(n)\rceil+2}(x) - \sigma(\mathcal{F}^1_{s_k+2\lceil\log(n)\rceil+2}(x) - |\mathcal{F}^0_{s_k-1}(x) - 2|))).
\end{aligned}
$$

We divide the proof into two cases.

**Case 1.** If $x \notin \mathbb{B}_\infty(x_k, 0.5\lambda_\mathcal{D})$, then $w_k = w_{k-1}$ and $\mathcal{F}^1_{s_k+2\lceil\log(n)\rceil+2}(x) = \sigma(-\tfrac{2}{\lambda_\mathcal{D}}\mathcal{F}^1_{s_k+2\lceil\log(n)\rceil+1}(x) + 1) = \sigma(1 - \tfrac{2}{\lambda_\mathcal{D}}\|x - x_k\|_\infty) = 0$ and

$$
\begin{aligned}
&\mathcal{F}^0_{s_k+2\lceil\log(n)\rceil+4}(x)\\
=~& \sigma(\mathcal{F}^0_{s_k-1}(x) + y_k\mathcal{F}^1_{s_k+2\lceil\log(n)\rceil+2}(x)\\
&\quad -y_k(\mathcal{F}^1_{s_k+2\lceil\log(n)\rceil+2}(x) - \sigma(\mathcal{F}^1_{s_k+2\lceil\log(n)\rceil+2}(x) - |\mathcal{F}^0_{s_k-1}(x) - 2|)))\\
=~& \mathcal{F}^0_{s_k-1}(x)\\
=~& \mathcal{F}^0_{s_{k-1}+2\lceil\log(n)\rceil+4}(x)\\
=~& 2 + y_{w_{k-1}}\sigma(1 - \tfrac{2}{\lambda_\mathcal{D}}\|x - x_{w_{k-1}}\|_\infty)\\
=~& 2 + y_{w_k}\sigma(1 - \tfrac{2}{\lambda_\mathcal{D}}\|x - x_{w_k}\|_\infty).
\end{aligned}
$$

**Case 2.** If $x \in \mathbb{B}_\infty(x_k, 0.5\lambda_\mathcal{D})$, then $\mathcal{F}^1_{s_k+2\lceil\log(n)\rceil+2}(x) = \sigma(-\tfrac{2}{\lambda_\mathcal{D}}\mathcal{F}^1_{s_k+2\lceil\log(n)\rceil+1}(x) + 1) = \sigma(1 - \tfrac{2}{\lambda_\mathcal{D}}\|x - x_k\|_\infty) \geq 0$ and using equation 17:

$$\mathcal{F}^0_{s_k+2\lceil\log(n)\rceil+4}(x)$$
$$= \sigma(\mathcal{F}^0_{s_k-1}(x) + y_k\mathcal{F}^1_{s_k+2\lceil\log(n)\rceil+2}(x)$$
$$-y_k(\mathcal{F}^1_{s_k+2\lceil\log(n)\rceil+2}(x) - \sigma(\mathcal{F}^1_{s_k+2\lceil\log(n)\rceil+2}(x) - |\mathcal{F}^0_{s_k-1}(x) - 2|)))$$
$$= \sigma(\mathcal{F}^0_{s_k-1}(x) + y_k\mathcal{F}^1_{s_k+2\lceil\log(n)\rceil+2}(x)$$
$$-y_k(\min\{\mathcal{F}^1_{s_k+2\lceil\log(n)\rceil+2}(x), |2 - \mathcal{F}^0_{s_k-1}(x)|\}))$$
$$= \sigma(2 + y_{w_{k-1}}\sigma(1 - \tfrac{2}{\lambda_\mathcal{D}}||x - x_{w_{k-1}}||_\infty) + y_k(1 - \tfrac{2}{\lambda_\mathcal{D}}||x - x_k||_\infty)$$
$$-y_k(\min\{1 - \tfrac{2}{\lambda_\mathcal{D}}||x - x_k||_\infty, \sigma(1 - \tfrac{2}{\lambda_\mathcal{D}}||x - x_{w_{k-1}}||_\infty)\})).$$

Consider two sub-cases:

**Case 2.1.** If $||x - x_{w_{k-1}}||_\infty > 0.5\lambda_\mathcal{D}$, then $w_k = k$ and hence

$$\mathcal{F}^0_{s_k+2\lceil\log(n)\rceil+4}(x)$$
$$= \sigma(2 + y_{w_{k-1}}\sigma(1 - \tfrac{2}{\lambda_\mathcal{D}}||x - x_{w_{k-1}}||_\infty) + y_k(1 - \tfrac{2}{\lambda_\mathcal{D}}||x - x_k||_\infty)$$
$$-y_k(\min\{1 - \tfrac{2}{\lambda_\mathcal{D}}||x - x_k||_\infty, \sigma(1 - \tfrac{2}{\lambda_\mathcal{D}}||x - x_{w_{k-1}}||_\infty)\}))$$
$$= \sigma(2 + y_k(1 - \tfrac{2}{\lambda_\mathcal{D}}||x - x_k||_\infty))$$
$$= 2 + y_k(1 - \tfrac{2}{\lambda_\mathcal{D}}||x - x_k||_\infty)$$
$$= 2 + y_{w_k}(1 - \tfrac{2}{\lambda_\mathcal{D}}||x - x_{w_k}||_\infty).$$

**Case 2.2.** If $||x - x_{w_{k-1}}||_\infty \le 0.5\lambda_\mathcal{D}$, then $y_{w_{k-1}} = y_k$ and hence

$$\mathcal{F}^0_{s_k+2\lceil\log(n)\rceil+4}(x)$$
$$= \sigma(2 + y_{w_{k-1}}\sigma(1 - \tfrac{2}{\lambda_\mathcal{D}}||x - x_{w_{k-1}}||_\infty) + y_k(1 - \tfrac{2}{\lambda_\mathcal{D}}||x - x_k||_\infty)$$
$$-y_k(\min\{1 - \tfrac{2}{\lambda_\mathcal{D}}||x - x_k||_\infty, \sigma(1 - \tfrac{2}{\lambda_\mathcal{D}}||x - x_{w_{k-1}}||_\infty)\}))$$
$$= \sigma(2 + y_{w_{k-1}}(1 - \tfrac{2}{\lambda_\mathcal{D}}||x - x_{w_{k-1}}||_\infty) + y_k(1 - 2/\lambda||x - x_k||_\infty)$$
$$-y_k(\min\{1 - \tfrac{2}{\lambda_\mathcal{D}}||x - x_k||_\infty, 1 - \tfrac{2}{\lambda_\mathcal{D}}||x - x_{w_{k-1}}||_\infty\}))$$
$$= 2 + y_k\max\{1 - \tfrac{2}{\lambda_\mathcal{D}}||x - x_k||_\infty, 1 - \tfrac{2}{\lambda_\mathcal{D}}||x - x_{w_{k-1}}||_\infty\}$$
$$= 2 + y_{w_k}\sigma(1 - \tfrac{2}{\lambda_\mathcal{D}}||x - x_{w_k}||_\infty).$$

The property is proved.

**Property 4.** $\mathcal{F}$ is a memorization $\mathcal{D}$ and has $\text{Lip}_\infty(\mathcal{F}) = \tfrac{2}{\lambda_\mathcal{D}}$.

By Property 3, the output is

$$\mathcal{F}(x) = \mathcal{F}^1_{s_N+2\lceil\log(n)\rceil+4}(x) - 2 = y_{w_N}\sigma(1 - \frac{2}{\lambda_\mathcal{D}}||x - x_{w_N}||_\infty)$$

where $w_N = \text{argmin}_{i\in[N]}||x - x_i||_\infty$.

If $x = x_s$, then $w_N = s$ and $\mathcal{F}(x) = y_s$; that is, $\mathcal{F}$ memorizes $\mathcal{D}$. If $x \in \mathbb{B}(x_s, 0.5\lambda_\mathcal{D})$ for some $s \in [N]$, then $w_N \in [N]$ and $\mathcal{F}(x) = y_{w_N}(1 - \tfrac{2}{\lambda_\mathcal{D}}||x - x_{w_N}||_\infty)$ such that the local $\text{Lip}_\infty(\mathcal{F}) = \tfrac{2}{\lambda_\mathcal{D}}$ over $\mathbb{B}(x_{w_N}, 0.5\lambda_\mathcal{D})$. If $x$ is not in $\cup_{i=1}^N\mathbb{B}(x_s, 0.5\lambda_\mathcal{D})$, then $||x - x_{w_N}||_\infty > 0.5\lambda_\mathcal{D}$. Therefore, $\mathcal{F}(x) = 0$ and the local $\text{Lip}_\infty(\mathcal{F}) = 0$. It is clear that the global Liptchitz constant is $\tfrac{2}{\lambda_\mathcal{D}}$, and the theorem is proved. □

### C.3 PROOF OF PROPOSITION 5.4

We first define $\mathcal{T} \in \mathcal{B}_{n,N}$, which is a binary classification data.

$$\mathcal{T} = \{(x_i, y_i)\}_{i=0}^n \subset \mathbb{R}^n \times \{-1, 1\} \tag{18}$$

where $x_0 = 0$, $y_0 = 1$, $x_i = \mathbf{1}_i$ and $y_i = -1$ for $i \in [n]$. It is easy to see that $\lambda_\mathcal{T} = 1$.

We first prove a lemma.

**Lemma C.2.** *If $\mathcal{F}$ is a network memorizing $\mathcal{T}$ and $\mathrm{Lip}_\infty(\mathcal{F}) = 2$, then $\mathcal{F}(x) = 1 - 2||x||_\infty$ for $x \in D = \{x \in \mathbb{R}^n : 0 \le x^{(i)} < 0.5, \forall i \in [n]\}$.*

*Proof.* For $x \in D$, let $k = \mathrm{argmax}_{i \in [n]}\{x^{(i)}\}$; that is, $||x||_\infty = x^{(k)}$. Let $z = x_k - x$, where $x_k = \mathbf{1}_k$ is defined in equation 18. Since $x_k = \mathbf{1}_k$, we have $z^{(i)} = x^{(i)} < 0.5$ when $i \ne k$ and $z^{(k)} = 1 - x^{(k)} > 0.5$; that is, $||z||_\infty = 1 - x^{(k)}$. Since $\mathrm{Lip}_\infty(\mathcal{F}) = 2$, we have

$$
\begin{aligned}
& \mathcal{F}(x_0) - \mathcal{F}(x_k) \\
= & (\mathcal{F}(x_0) - \mathcal{F}(x)) + (\mathcal{F}(x) - \mathcal{F}(x_k)) \\
\le & 2||x_0 - x||_\infty + 2||x_k - x||_\infty \\
= & 2x^{(k)} + 2||x_k - x||_\infty \\
= & 2x^{(k)} + 2(1 - x^{(k)}) \\
= & 2.
\end{aligned}
\tag{19}
$$

Since $\mathcal{F}$ memorizes $\mathcal{T}$, we have $\mathcal{F}(x_0) - \mathcal{F}(x_k) = 2$, which means that the inequality in equation 19 becomes an equation. Then $\mathcal{F}(x_0) - \mathcal{F}(x) = 2x^{(k)}$; that is $\mathcal{F}(x) = \mathcal{F}(x_0) - 2x^{(k)} = 1 - 2||x||_\infty$. $\square$

We now prove Proposition 5.4. Note that $\lambda_\mathcal{T} = 1$, which is omitted in the proof.

*Proof.* It suffices to show that if $\mathcal{F} \in \mathbf{H}$ is a memorization of $\mathcal{T}$ defined in equation 18, then $\mathrm{Lip}_\infty(\mathcal{F}) > 2$; that is, $\mathbf{H}$ is not an optimal robust memorization for $\mathcal{T}$ via Lipschitz.

Since $\mathcal{F}$ memorizes $\mathcal{T}$, we have $\mathrm{Lip}_\infty(\mathcal{F}) \ge 2$. Let $\mathcal{F}(x) = \sum_{i=1}^k a_i \sigma(U_i x + b_i) + Qx + b$ have the normal form equation 4.

Assume the contrary: $\mathrm{Lip}_\infty(\mathcal{F}) = 2$. By Lemma C.2, we know that $\mathcal{F}(x) = 1 - 2||x||_\infty$ over $D = \{x \in \mathbb{R}^n : 0 < x^{(i)} < 0.5, \forall i \in [n]\}$.

A point $t \in \mathbb{R}^n$ is called a T-point, if there exist a $j \in [n]$ and an $\epsilon \in \mathbb{R}_+$ such that $\epsilon < t^{(k)} < 0.5 - \epsilon$ for any $k \in [n]$ and $|t^{(j)}| - |t^{(i)}| > \epsilon > 0$ when $i \ne j$.

For $s \in [k]$, let $H_s$ be the hyperplane defined by $U_s x + b_s = 0$.

Then one of the following properties must be valid:

(p1) $H_s$ does not intersect $D$.

(p2) $H_s$ intersects $D$ and there exists a T-point $t \in D$ such that $U_s t + b_s = 0$.

(p3) $H_s$ intersects $D$ and there does not exist a T-point $t \in D$ such that $U_s t + b_s = 0$.

We first prove the following properties.

**(c1)** Property (p1) is not valid for some $s \in [k]$.

If (p1) is valid for all $s \in [k]$, then $D$ is inside a linear region of $\mathcal{F}$. This is impossible because, according to Lemma C.2, $\mathcal{F}(x) = 1 - 2||x||_\infty$ over $D$, and $||\cdot||_\infty$ is not linear over $D$.

**(c2)** Property (p2) is not valid for any $s \in [k]$.

Suppose (p2) is valid for $s$. Let $t \in D$ be a T-point satisfying $U_s t + b_s = 0$ and $\epsilon < t^{(k)} < 0.5 - \epsilon$ for all $k \in [n]$. Let $j = \operatorname{argmax}_{i \in [n]}\{t^{(i)}\}$, $|t^{(j)}| - |t^{(j')}| > \epsilon > 0$ when $j' \neq j$. By Lemma A.3, there exist two linear regions $R_1$ and $R_2$ whose boundary is $(U_s, b_s)$ and points $P \in R_1, Q \in R_2$ such that $||P - t||_\infty < \epsilon/3$ and $||Q - t||_\infty < \epsilon/3$. Then we have

**(c2.1)** $0.5 - 2\epsilon/3 > P^{(k)} > 2\epsilon/3$ and $0.5 - 2\epsilon/3 > Q^{(k)} > 2\epsilon/3$ for any $k \in [n]$, which implies $P, Q \in D$.

**(c2.2)** $j = \operatorname{argmax}_{i \in [n]}\{|P^{(i)}|\} = \operatorname{argmax}_{i \in [n]}\{|Q^{(i)}|\}$, $|P^{(j)}| - |P^{(j')}| > \epsilon/3 > 0$, $|Q^{(j)}| - |Q^{(j')}| > \epsilon/3 > 0$ when $j' \neq j$.

By Lemma C.2 and **(c2.1)**, we have $\mathcal{F}(x) = 1 - 2||x||_\infty = 1 - 2x^{(j)}$ over $\mathbb{B}_\infty(P, \epsilon/7) \cap R_1$, because for any $x \in \mathbb{B}_\infty(P, \epsilon/7)$, by **(c2.2)**, we have $x^{(j)} - x^{(j')} > (P^{(j)} - \epsilon/7) - (P^{(j')} + \epsilon/7) > \epsilon(1/3 - 2/7) > 0$ when $j' \neq j$.

Since $\mathcal{F}(x)$ is a linear function on $R_1$, we have $\mathcal{F}(x) = 1 - 2x^{(j)}$ on $R_1$, and the same is true for $R_2$. Thus, the normal vectors of $\mathcal{F}$ are the same for $R_1$ and $R_2$. On the other hand, since $R_1$ and $R_2$ have boundary $(U_s, b_s)$ and $U_s \neq 0$, the normal vectors of $\mathcal{F}$ over $R_1$ and $R_2$ are not the same, a contradiction.

**(c3)** Property (p3) is not valid for any $s \in [k]$.

If (p3) is valid, then find a point $t \in D \cap H_s$ and let $\epsilon = 0.5 - ||t||_\infty$. Since $t$ is not a T-point, there exist $a, b \in [n]$ such that $t^{(a)} = t^{(b)} = ||t||_\infty = 0.5 - \epsilon$. (If $a, b$ do not exist, we just need to take $\epsilon' = \min_{i \in [n]\, i \neq \operatorname{argmax}_{j \in [n]}\{|t^{(j)}|\}}\{0.5 - ||t||_\infty, |t^{(i)}|, ||t||_\infty - |t^{(i)}|\}$. Then $t$ is a T-point for $j = \operatorname{argmax}_{j \in [n]}\{|t_j|\}$ and $\epsilon'$.)

We will find a T-point $t_1$ near $t$ such that $t_1 \in D$ and $U_s t_1 + b_s = 0$, which means that (p3) is not correct. Three cases are considered.

**(c3.1)** $U_s^{(k)} = 0$ for $k \in [n]$. In this case, let $t_1 = t + (0.5 - \epsilon/2)\mathbf{1}_k - t^{(k)}\mathbf{1}_k$. We have $U_s t_1 + b_s = U_s(t + (0.5 - \epsilon/2)\mathbf{1}_k - t^{(k)}\mathbf{1}_k) + b_s = U_s t + b_s = 0$ and it is easy to see that $t_1^{(k)} = 0.5 - \epsilon/2 > t^{(c)} = t_1^{(c)}$ when $c \neq k$, so $t_1$ is a T-point.

**(c3.2)** $U_s^{(k)} \neq 0$ for all $k \in [n]$ and $\operatorname{Sgn}(U_s^{(b)}) = \operatorname{Sgn}(U_s^{(a)})$, let $|U_s^{(b)}| > |U_s^{(a)}|$. In this case, let $t_1 = t + \epsilon/2\mathbf{1}_a - \frac{U_s^{(a)}}{U_s^{(b)}}\epsilon/2\mathbf{1}_b$. Then $U_s t_1 + b_s = U_s(t + \epsilon/2\mathbf{1}_a - \frac{U_s^{(a)}}{U_s^{(b)}}\epsilon/2\mathbf{1}_b) + b_s = U_s(\epsilon/2\mathbf{1}_a - \frac{U_s^{(a)}}{U_s^{(b)}}\epsilon/2\mathbf{1}_b) = U_s^{(a)}\epsilon/2 - U_s^{(b)}\epsilon/2\frac{U_s^{(a)}}{U_s^{(b)}} = 0$. We also have $0.5 > t_1^{(a)} = t^{(a)} + \epsilon/2 = t^{(b)} + \epsilon/2 > t^{(b)} > t^{(b)} - \frac{U_s^{(a)}}{U_s^{(b)}}\epsilon/2 = t_1^{(b)}$, because $\operatorname{Sgn}(U_s^{(b)}) = \operatorname{Sgn}(U_s^{(a)})$, and $t_1^{(a)} = t^{(a)} + \epsilon/2 > t^{(a)} \geq t^{(k)} = t_1(k)$ when $k \neq a, b$. Thus, $t_1$ is a T-point.

**(c3.3)** $U_s^{(k)} \neq 0$ for all $k \in [n]$ and $\operatorname{Sgn}(U_s^{(b)}) \neq \operatorname{Sgn}(U_s^{(a)})$. Let $U_s^{(a)} > 0 > U_s^{(b)}$ and $c \neq a, b$, such that $U_s^{(c)} > 0$. Let $t_1 = t + (\epsilon/2\mathbf{1}_a - \epsilon/2\frac{U_s^{(a)}}{U_s^{(c)}}\mathbf{1}_c)\eta$, where $\eta \in (0, 1)$ and make $\eta\epsilon/2\frac{U_s^{(a)}}{U_s^{(c)}} < t^c$. Then $U_s t_1 + b_s = U_s(t + \eta\epsilon/2\mathbf{1}_a - \eta\frac{U_s^{(a)}}{U_s^{(c)}}\epsilon/2\mathbf{1}_c) + b_s = U_s(\eta\epsilon/2\mathbf{1}_a - \eta\epsilon/2\frac{U_s^{(a)}}{U_s^{(c)}}\mathbf{1}_c) = \eta U_i^{(a)}\epsilon/2 - \eta U_s^{(c)}\epsilon/2\frac{U_s^{(a)}}{U_s^{(c)}} = 0$. We also have $0.5 > t_1^{(a)} = t^{(a)} + \eta\epsilon/2 \geq t^{(c)} + \eta\epsilon/2 > t^{(c)} - \eta\frac{U_s^{(a)}}{U_s^{(c)}}\epsilon/2 = t_1^{(c)}$, because $U_s^{(c)}, U_s^{(a)} > 0$, and

$t_1^{(a)} = t^{(a)} + \eta\epsilon/2 > t^{(a)} \geq t^{(k)} = t_1^{(k)}$ when $k \neq a, c$. Thus, $t_1$ is a T-point. If $U_s^{(b)} < 0$, we only need to let $t_1 = t + \eta(\epsilon/21_b - \frac{U_s^{(b)}}{U_s^{(c)}}\epsilon/21_c)$ and the proof is the same.

By **(c1)**, (p1) is not valid for some $i$. By **(c2)** and **(c3)**, (p2) and (p3) are not valid for all $i$. Thus, we reach a contradiction, because for any $s \in [k]$, one of (p1) , (p2) , (p3) must be valid. Therefore, the assumption $\text{Lip}_\infty(\mathcal{F}) = 2$ is wrong. The lemma is proved. $\qquad\square$

### C.4 PROOF OF PROPOSITION 5.5

Let $\mathcal{T}$ be the dataset defined in equation 18. We first give a technical lemma, whose proof is given in Appendix C.4.2.

**Lemma C.3.** *Let $C \in \mathbb{N}_+$ and $\epsilon \in \mathbb{R}_+$ satisfy $\epsilon^{1/3} < \min\{\frac{1}{120}, \frac{1}{100n^2}, \frac{1}{2C}\}$ and $k \leq C$; and let $\mathcal{F}(x) = \sum_{i=1}^k a_i \sigma(U_i x + b_i) + Qx + b$ be a network with normal form equation 4, which memorizes $\mathcal{T}$ and $\text{Lip}_\infty(\mathcal{F}) \leq 2 + \epsilon$. Then, for any $i, j \in [n]$ and $i \neq j$, there exists an $s \in [k]$ such that*

$$\frac{0.5 + 4\epsilon^{1/3}}{0.5 - 5\epsilon^{1/3}} \geq \frac{|U_s^{(i)}|}{|U_s^{(j)}|} \geq \frac{0.5 - 5\epsilon^{1/3}}{0.5 + 4\epsilon^{1/3}} \text{ and } \frac{|U_s^{(q)}|}{\max(|U_s^{(i)}|, |U_s^{(j)}|)} < \frac{12\epsilon^{1/3}}{1 - 2\epsilon^3} \text{ for } q \neq i, j; \quad (20)$$

*and the linear equation $U_s x + b_s = 0$ has a solution in the $\mathbb{B}_\infty(0, 2)$.*

#### C.4.1 PROOF OF PROPOSITION 5.5

*Proof.* Let $V = (1, 2, \ldots, n)^T \in \mathbb{R}^n$, $M = [\frac{N}{n+1}]$, and $\mathcal{T}$ be defined in equation 18. We define $\mathcal{D}$ as:

$$\mathcal{D} = \mathcal{T}_0 \cup \mathcal{T}_1 \cup \mathcal{T}_2 \cup \mathcal{T}_3 \cdots \cup \mathcal{T}_M,$$

where $\mathcal{T}_0 = \{(-V, 1), (-2V, 1), \ldots, (-(N - (n+1)M)V, 1)\}$ and $T_k = \mathcal{T} + 16knV = \{(x, y) : x = \tilde{x} + 16nkV, \ y = \hat{y}, \ (\tilde{x}, \hat{y}) \in \mathcal{T}\}$ for $k \in [M]$. We see that $\lambda_\mathcal{D} = 1$ and is omitted from the rest of the proof.

We prove Proposition 5.5 (1); that is,

**(1) For any $\mu < 0.5$, there exists a network $\mathcal{F}$ with depth 2 and width $4nM$, which is a robust memorization of $\mathcal{D}$.**

For $\epsilon < 0.5$, define a network $\mathcal{G}_\epsilon : \mathbb{R} \to \mathbb{R}$:

$$\mathcal{G}_\epsilon(x) = \sigma(x - 0.5 + \epsilon) - \sigma(x - 0.5 - \epsilon) - (\sigma(x - 1.5 + \epsilon) - \sigma(x - 1.5 - \epsilon)).$$

It is easy to see that $\mathcal{G}_\epsilon$ has depth 2, width 4, and satisfies

(c1) $\mathcal{G}_\epsilon(x) = 0$ if $x \leq 0.5 - \epsilon$ and $x \geq 1.5 + \epsilon$.

(c2) $\mathcal{G}_\epsilon(x) = 2\epsilon$ if $0.5 + \epsilon \leq x \leq 1.5 - \epsilon$.

For $i \in [n]$, define $\mathcal{G}_\epsilon^i(x) : \mathbb{R}^n \to \mathbb{R}$ as $\mathcal{G}_\epsilon^i(x) = \mathcal{G}_\epsilon(x^{(i)})$ and define $\mathcal{G} : \mathbb{R}^n \to \mathbb{R}$ as

$$\mathcal{G}(x) = -\frac{2}{1 - 2\mu} \sum_{i=1}^n \mathcal{G}_{0.5-\mu}^i(x) + 1$$

where $\mu \in (0, 0.5)$. Then $\mathcal{G}$ is the network with depth 2 and width $4n$. Using (c1) and (c2), we know that $\mathcal{G}$ is a robust memorization of $\mathcal{T}$ with budget $\mu$. Let network $\mathcal{F} : \mathbb{R}^n \to \mathbb{R}$ be defined as:

$$\mathcal{F}(x) = -\frac{2}{1 - 2\mu} (\sum_{i=1}^n \sum_{j=1}^M \mathcal{G}_{0.5-\mu}^i(x - 16njV)) + 1.$$

Obviously, $\mathcal{F}$ has depth 2 and width $4nM$. For any $k \in [M]$ and $x \in \mathbb{B}_\infty(\mathcal{T}_k, \mu)$, since $\mathcal{T}_k = \mathcal{T} + 16knV$, there exists an $\tilde{x} \in \mathbb{B}_\infty(\mathcal{T}, \mu)$ that satisfies $x = \tilde{x} + 16knV$. Therefore,

$$
\begin{aligned}
&\mathcal{F}(x) \\
=\ & -\tfrac{2}{1-2\mu}\left(\sum_{i=1}^n \sum_{j=1}^M \mathcal{G}_{0.5-\mu}^i(x - 16njV)\right) + 1 \\
=\ & -\tfrac{2}{1-2\mu}\left(\sum_{i=1}^n \sum_{j=1}^M \mathcal{G}_{0.5-\mu}^i(\tilde{x} - 16n(k-j)V)\right) + 1 \\
=\ & -\tfrac{2}{1-2\mu}\sum_{i=1}^n \mathcal{G}_{0.5-\mu}^i(\tilde{x}) + 1 \,(\text{ by (c1)}) \\
=\ & \mathcal{G}(\tilde{x}).
\end{aligned}
$$

Since $G$ is a robust memorization of $\mathcal{T}$ with budget $\mu$, $\mathcal{F}$ is a robust memorization of $\mathcal{T}_k(k \in [M])$ with budget $\mu$. Now we show that $\mathcal{F}$ is a robust memorization of $T_0$ with budget $\mu$. It is easy to see that $\mathcal{F}(\tilde{x}) = 1$ for any $\tilde{x} \in \mathbb{B}_\infty(\mathcal{T}_0, \mu)$, because all the entries of $\tilde{x}$ are negative, so $\mathcal{G}_{0.5-\mu}^i(x - 16njV)$ are always 0 for any $i \in [n]$ and $j \in [M]$. Thus, $\mathcal{F}$ is a robust memorization of $\mathcal{D}$ with budget $\mu$ and Proposition 5.5(1) is proved.

We now prove Proposition 5.5(2); that is

**(2) There exists a $\mu < 0.5$, such that, for any $\mathcal{F}$ with depth 2 and width $4nM$, if $\mathcal{F}$ is a memorization of $\mathcal{D}$, then $\mathrm{Lip}_\infty(\mathcal{F}) > \frac{1}{\mu}$.**

Let $\mu$ and $\epsilon$ satisfy $1/\mu = 2 + \epsilon^{1/3}$ and $\epsilon^{1/3} < \min\{\frac{1}{120}, \frac{1}{100n^2}, \frac{1}{8nM}\}$.

Assume $\mathcal{F} = W\sigma(Ux + b) + b_1$ is a network with depth 2 and width $4nM$, which is a memorization of $\mathcal{D}$ and $\mathrm{Lip}_\infty(\mathcal{F}) \le 1/\mu$. For any $k \in [M]$, $\mathcal{T}_k = \mathcal{T} + 16knV$ is contained in $\mathbb{B}_\infty(0, 2) + 16knV$. So using Lemma C.3 to $\mathcal{T}_k$, for the any $(i, j)$, $i, j \in [n]$, there exists an $s \in \mathbb{N}_+$ satisfying

$$
\begin{aligned}
&\tfrac{0.5+4\epsilon^{1/3}}{0.5-5\epsilon^{1/3}} \ge \tfrac{|U_s^{(i)}|}{|U_s^{(j)}|} \ge \tfrac{0.5-5\epsilon^{1/3}}{0.5+4\epsilon^{1/3}} \text{ and } \tfrac{|U_s^{(q)}|}{\max(|U_s^{(i)}|, |U_s^{(j)}|)} < \tfrac{12\epsilon^{1/3}}{1-2\epsilon^3} \text{ for } q \ne i, j \\
&\text{and } U_s x + b_s = 0 \text{ has zeros in } \mathbb{B}_\infty(0, 2) + 16nkv.
\end{aligned}
\tag{21}
$$

We claim that different combinations $(k, i, j)$, where $k \in [M]$, $i < j$, and $i, j \in [n]$, correspond to different $s$, which implies that the width of $\mathcal{F}$ is at least $\frac{n-1}{2}nM > 4nM$ since we assumed $n > 9$, which leads to a contradiction and a proof for Proposition 5.5(2).

We prove the claim below. Let $(k, i, j)$ and $(k_1, i_1, j_1)$ correspond to the same $s$. Firstly, we show that $i = i_1$ and $j = j_1$. From $\epsilon^{1/3} < \frac{1}{120}$, we can deduce $\frac{0.5-5\epsilon^{1/3}}{0.5+4\epsilon^{1/3}} > 2/3$ and $\frac{12\epsilon^{1/3}}{1-2\epsilon^3} < 1/3$. Without loss of generality, assume $|U_s^{(i)}| \le |U_s^{(j)}|$. Then, from equation 21, we have $\frac{|U_s^{(i)}|}{|U_s^{(j)}|} \ge 2/3$ and $\frac{|U_k^{(q)}|}{|U_s^{(j)}|} < 1/3$ for $q \ne i, j$. As a consequence, for different pairs $(i, j), i, j \in [n]$, the corresponding $s$ must be different, so $i = i_1$ and $j = j_1$.

We now show $k = k_1$. If $k \ne k_1$, let $k < k_1$. By equation 21, we have $0 = U_s(x + 16nkV) + b_s = U_s(\tilde{x} + 16nk_1V) + b_s$ for some $x, \tilde{x} \in \mathbb{B}_\infty(0, 2)$, which implies $|U_s(x - \tilde{x})| = |16n(k - k_1)(U_sV)|$. Firstly, we have

$$
\begin{aligned}
&|U_s(x - \tilde{x})| \\
\le\ & \|U_s\|_1 \|(x - \tilde{x})\|_\infty \\
\le\ & 4\|U_s\|_1 \\
\le\ & 4(|U_s^{(i)}| + |U_s^{(j)}| + \tfrac{12\epsilon^{1/3}}{1-2\epsilon^{1/3}}(d-2)\max\{|U_s^{(j)}|, |U_s^{(i)}|\})\,(\text{ by } equation\ 21) \\
\le\ & 4|U_s^{(j)}|((1 + \tfrac{0.5+4\epsilon^{1/3}}{0.5-5\epsilon^{1/3}} + \tfrac{0.5+4\epsilon^{1/3}}{0.5-5\epsilon^{1/3}}\tfrac{12\epsilon^{1/3}}{1-2\epsilon^{1/3}}(d-2)) \\
\le\ & 4|U_s^{(j)}|(1 + 1.17 + 0.119(n-2))\,(\text{use } \epsilon^{1/3} < 1/120) \\
\le\ & 4n|U_s^{(j)}|.
\end{aligned}
$$

Consider that we have assumed $i < j$, then

$$
\begin{aligned}
& |16n(k - k_1)(U_s V)| \\
\geq\ & 16n|U_s v| \\
\geq\ & 16n(j|U_s^{(j)}| - i|U_s^{(i)}| - \sum_{p \neq i,j} p|U_s^{(p)}|) \\
\geq\ & 16n(j|U_s^{(j)}| - i\frac{0.5+4\epsilon^{1/3}}{0.5-5\epsilon^{1/3}}|U_s^{(j)}| - \sum_{p \neq i,j} p\frac{12\epsilon^{1/3}}{1-2\epsilon^{1/3}} \max\{|U_s^{(j)}|, |U_s^{(i)}|\}) \\
\geq\ & 16n(j|U_s^{(j)}| - i\frac{0.5+4\epsilon^{1/3}}{0.5-5\epsilon^{1/3}}|U_s^{(j)}| - n^2\frac{12\epsilon^{1/3}}{1-2\epsilon^{1/3}} \max\{|U_s^{(j)}|, |U_s^{(i)}|\}) \\
\geq\ & 16n(j|U_s^{(j)}| - i\frac{0.5+4\epsilon^{1/3}}{0.5-5\epsilon^{1/3}}|U_s^{(j)}| - n^2\frac{12\epsilon^{1/3}}{1-2\epsilon^{1/3}}\frac{0.5+4\epsilon^{1/3}}{0.5-5\epsilon^{1/3}}|U_s^{(j)}|) \\
\geq\ & 16n|U_s^{(j)}|(j - (j-1)\frac{0.5+4\epsilon^{1/3}}{0.5-5\epsilon^{1/3}} - n^2\frac{12\epsilon^{1/3}}{1-2\epsilon^{1/3}}\frac{0.5+4\epsilon^{1/3}}{0.5-5\epsilon^{1/3}}|) \\
\geq\ & 16n|U_s^{(j)}|(n - (n-1)\frac{0.5+4\epsilon^{1/3}}{0.5-5\epsilon^{1/3}} - n^2\frac{12\epsilon^{1/3}}{1-2\epsilon^{1/3}}\frac{0.5+4\epsilon^{1/3}}{0.5-5\epsilon^{1/3}}|) \\
\geq\ & 16n|U_s^{(j)}|(n - (n-1)(1 + 20\epsilon^{1/3}) - 24n^2\epsilon^{1/3}(1 + 20\epsilon^{1/3}))\ (\text{use } \epsilon^{1/3} < 1/120) \\
\geq\ & 16n|U_s^{(j)}|(1 - 20\epsilon^{1/3}(n-1) - 29n^2\epsilon^{1/3})(\ \text{use } \epsilon^{1/3} < 1/120) \\
\geq\ & 16n|U_s^{(j)}|(1 - 49/100)\ (\text{use } \epsilon^{1/3} < 1/100n^2) \\
\geq\ & 8n|U_s^{(j)}|.
\end{aligned}
$$

So, it always holds $|U_s(x - \tilde{x})| \leq 4d|U_s^{(j)}| < 8n|U_s^{(j)}| \leq |16n(k - k_1)(U_s V)|$, which means $|U_s(x - \tilde{x})| = |16n(k - k_1)(U_s V)|$ is not valid, and $(k, i, j)$ and $(k_1, i_1, j_1)$ cannot correspond to the same $s$. Thus $k = k_1$. The claim and hence the proposition are proved. $\qquad\square$

*Remark* C.4. From the proof, we see that replacing $\mathbf{H}$ in the Proposition 5.5 by $\mathbf{H}(C_w) = \{\mathcal{F} : \mathbb{R}^n \to \mathbb{R} : \text{depth}(\mathcal{F}) = 2, \text{width}(\mathcal{F}) = C_w\}$, where $C_w \in \mathbb{N}$ satisfies $4n[\frac{N}{n+1}] \leq C_{wid} < \frac{(n-1)n}{2}[\frac{N}{n+1}]$, Proposition 5.5 is still valid.

### C.4.2 Proof of Lemma C.3

*Proof.* In the proof, the fact $\lambda_\mathcal{T} = 1$ is used. First, we introduce several notations.

(1) For a linear region $R$ of $\mathcal{F}$, let $\mathcal{F}(x) = \sum_{i=1}^n l_i x^{(i)} + c$ over $R$ with $N_{\mathcal{F},R} = (l_1, \ldots, l_n)$ as the normal vector.

(2) For $t \in \mathbb{R}_+$, let $R_i(t)$ be the set of linear regions $R$ of $\mathcal{F}$ such that the normal vector $N_{\mathcal{F},R} = (l_1, \ldots, l_n)$ satisfies $|l_i| \geq t$ and denote

$$P_i(t) = \{x \in \mathbb{R}^n : \exists R \in R_i(t) \text{ such that } x \in R\}.$$

(3) For $a, b \in \mathbb{R}^n$, use $a \to b$ to denote the directed segment from $a$ to $b$. For $i \in [n]$, the $H_i$-length of $a \to b$ is defined as $|a^{(i)} - b^{(i)}|$.

(4) Assume $a_i \in \mathbb{R}^n, i = 1, \ldots, k$ for $k \geq 3$. Use $a_1 \to a_2 \to \cdots \to a_k$ to denote the polyline segment $a_1 \to a_2 \to \cdots \to a_{k-1} \to a_k$.

Without loss of generality, assume $i = 1, j = 2$. For convenience, let

$$\theta = \frac{1}{\epsilon + 2}, \epsilon_0 = 2 + \epsilon - \epsilon^{1/3}, T = 1 - \epsilon^{1/3}.$$

We will prove a series of properties which will lead to a proof of the lemma.

**(p1).** For a linear region $R$ of $\mathcal{F}$, $N_{\mathcal{F},R} = (l_1, \ldots, l_n)$ satisfies $\sum_{i=1}^n |l_i| \leq \frac{1}{\theta} = 2 + \epsilon$.

There exist $x \in \mathbb{R}^n$ and $y = x - \delta \text{Sgn}(N_{\mathcal{F},R}) \in \mathbb{R}^n, \delta \in \mathbb{R}_{>0}$, such that $\frac{1}{\theta} \geq \text{Lip}_\infty(\mathcal{F}) \geq \frac{|\mathcal{F}(x)-\mathcal{F}(y)|}{||x-y||_\infty} = \frac{|\sum_{i=1}^n l_i(x^{(i)}-y^{(i)})|}{||x-y||_\infty} = |\sum_{i=1}^n l_i \frac{(x^{(i)}-y^{(i)})}{||x-y||_\infty}| = |\sum_{i=1}^n l_i \frac{\delta \text{Sgn}(l_i)}{\delta}| = |\sum_{i=1}^n l_i \text{Sgn}(l_i)| = \sum_{i=1}^n |l_i|$, from which we have $\sum_{i=1}^n |l_i| \leq \frac{1}{\theta}$. **(p1)** is proved.

Let $V = (1, v_1, \ldots, v_{n-1})^T \in \mathbb{R}^n$, where $|v_i| < T$ for $i \in [n-1]$. By Lemma A.2, the interior of a linear region is an open set of dimension $n$, but an edge of linear region is of dimension $\leq n-1$, so for almost all $v_i$, the polyline segment $B : x_0 \to x_0 + \theta V \to x_1$ is in the interior of linear regions of $\mathcal{F}$, except a finite number of points. We assume $V$ is chosen to satisfy the above condition.

**(p2).** By definition, $P_1(\epsilon_0) = \cup_{i=1}^h R_i$ is a set of linear regions of $\mathcal{F}$. By Lemma A.2, a linear region is a convex polyhedron. Denote the directed line segment $x_0 \to x_0 + \theta V = 0 \to \theta V$ by $D$. Then $D_i = D \cap R_i$ is also a directed line segment, and hence $D \cap P_1(\epsilon_0) = \cup_{i=1}^h D_i$. Then it holds

$$H_{\epsilon_0} \triangleq H_1\text{-length}(D \cap P_1(\epsilon_0)) = \sum_{i=1}^h H_1\text{-length}(D_i) \geq \theta - \epsilon^{1/3} = \frac{1}{\epsilon+2} - \epsilon^{1/3}. \tag{22}$$

For a linear region $R$, if $tV \in R$, let $U(t) = \mathcal{F}(x_0 + tV) = \mathcal{F}(tV) = (\sum_{i=2}^n l_i(tV)v_i + l_1(tV))t + b(tV)$, where $N_{\mathcal{F},R} = (l_1(tV), \ldots, l_n(tV))$ for $t \in \mathbb{R}$. Then $U(t)$ is a piecewise linear function in $t$ and $l_i(tV)$ is piecewise constant for each $i$. When the segment $D$ is in the interior of linear regions except a finite number of points, we can calculate the derivatives of $U(t)$; that is, $U'(t) = \frac{\nabla \mathcal{F}(tV)}{\nabla t} = \sum_{i=2}^n (l_i(tV)v_i) + l_1(tV)$.

We thus have

$$\begin{aligned}
& \mathcal{F}(x_0) - \mathcal{F}(x_0 + \theta V) = \mathcal{F}(0) - \mathcal{F}(\theta V) = U(0) - U(\theta) \\
= & \int_0^\theta U'(t)dt \\
= & \int_0^\theta l_1(tV) + \sum_{i=2}^n l_i(tV)v_i dt \\
\leq & \int_0^\theta l_1(tV) + \sum_{i=2}^n |l_i(tV)|T dt \text{ (by } |v_i| \leq T) \\
\leq & \int_0^\theta l_1(tV) + (2 + \epsilon - l_1(tV))T dt \text{ by (p1).}
\end{aligned} \tag{23}$$

Since $\text{Lip}_\infty(\mathcal{F}) \leq 2 + \epsilon$ and $\theta < 0.5$, we have $\mathcal{F}(x_0 + \theta V) - \mathcal{F}(x_1) \leq (2+\epsilon)||x_1 - \theta V||_\infty = (2+\epsilon)(1-\theta)$. Thus by equation 23, we have

$$\begin{aligned}
& \int_0^\theta (l_1(tV) + (2 + \epsilon - l_1(tV))T)dt + (2 + \epsilon)(1 - \theta) \\
\geq & (\mathcal{F}(x_0) - \mathcal{F}(x_0 + \theta V)) + (\mathcal{F}(x_0 + \theta V) - \mathcal{F}(x_1)) \\
= & 2.
\end{aligned}$$

Since $T < 1$, we have that

(k1) if $l_1(tV) \geq \epsilon_0$, then $l_1(tV) + (2 + \epsilon - l_1(tV))T \leq 2 + \epsilon$;

(k2) if $l_1(tV) < \epsilon_0$, then $l_1(tV) + (2 + \epsilon - l_1(tV))T \leq \epsilon_0 + (2 + \epsilon - \epsilon_0)T$.

Note that the $H_1$-length of $D$ is $\theta$. Since the $H_1$-length of $D \cap P_1(\epsilon_0)$ is $H_{\epsilon_0}$, the $H_1$-length of $D \setminus D \cap P_1(\epsilon_0)$ is $\theta - H_{\epsilon_0}$. Then we have

$$\begin{aligned}
& \int_{t=0}^\theta (l_1(tV) + (2 + \epsilon - l_1(tV))T)dt \\
\leq & \int_{t=0}^\theta (2 + \epsilon)I(l_1(tV) \geq \epsilon_0) + (\epsilon_0 + (2 + \epsilon - \epsilon_0)T)I(l_1(tV) < \epsilon_0)dt \\
= & (2 + \epsilon)H_{\epsilon_0} + (\epsilon_0 + (2 + \epsilon - \epsilon_0)T)(\theta - H_{\epsilon_0}),
\end{aligned}$$

so

$$(2 + \epsilon)H_{\epsilon_0} + (\epsilon_0 + (2 + \epsilon - \epsilon_0)T)(\theta - H_{\epsilon_0}) + (2 + \epsilon)(1 - \theta) \geq 2,$$

from which we can deduce $H_{\epsilon_0} \geq \theta - \frac{\epsilon}{(2+\epsilon-\epsilon_0)(1-T)}$.

Since $\epsilon_0 = 2 + \epsilon - \epsilon^{1/3}$ and $T < 1 - \epsilon^{1/3}$, we have that $H_{\epsilon_0} \geq \theta - \frac{\epsilon}{(2+\epsilon-\epsilon_0)(1-T)} = \theta - \frac{\epsilon}{\epsilon^{1/3}(1-T)} = \theta - \frac{\epsilon^{2/3}}{1-T} \geq \theta - \frac{\epsilon^{2/3}}{\epsilon^{1/3}} = \theta - \epsilon^{1/3}$. Property **(p2)** is proved.

Consider the directed segments $D_1 : x_0 \to x_0 + \theta E_1 = 0 \to \theta E_1$ and $D_2 : x_0 \to x_0 + \theta E_2 = 0 \to \theta E_2$, where $E_1 = (1, T', \epsilon_1, \ldots, \epsilon_{n-2})^T$, $E_2 = (T', 1, \epsilon_1, \ldots, \epsilon_{n-2})^T$, $T' \in (1 - 2\epsilon^{1/3}, 1 - \epsilon^{1/3})$, and $|\epsilon_i| < \epsilon^{1/3}\theta$ for $i \in [n-2]$. Similar to **(p2)**, $E_1$ and $E_2$ are chosen such that $D_1$ and $D_2$ are in the interior of linear regions of $\mathcal{F}$, except a finite number of points.

Let $I = ((1/2 - 3\epsilon^{1/3})\theta, (1/2 + 3\epsilon^{1/3})\theta)$ and denote the length of $I$ as $|I|$. From $\epsilon^{1/3} < \frac{1}{120}$, we have $\theta > 1/3$, so $|I| > 2\epsilon^{1/3}$. Let $I_i$ be the set of $\eta \in I$ such that $\eta E_i$ is in $P_i(\epsilon_0)$ for $i = 1, 2$. By property **(p2)**, the $H_1$-length of the segment $D_1 \setminus P_1(\epsilon_0)$ is at most $\epsilon^{1/3}$ and the total $H_2$-length of the segments $D_2 \setminus P_2(\epsilon_0)$ is at most $\epsilon^{1/3}$; that is, $|I/I_i| \leq \epsilon^{1/3}$. Then $|I_1 \cap I_2| = |I \setminus ((I/I_1) \cup (I/I_2))| \geq |I| - |I/I_1| - |I/I_2| > 0$; that is, $I_1 \cap I_2 \neq \emptyset$. We thus have

**(p3).** There exists an $\eta \in I_1 \cap I_2$, such that $q_i = \eta E_i \in P_i(\epsilon)$ for $i = 1, 2$.

Suppose $q_i$ is in the linear region $R_i$ and $N_{\mathcal{F},i}$ is the normal vector of $\mathcal{F}$ over $R_i$ for $i = 1, 2$. It is easy to see that $|N_{\mathcal{F},1}^{(1)}| \geq \epsilon_0, |N_{\mathcal{F},1}^{(2)}| \leq \epsilon^{1/3}, |N_{\mathcal{F},2}^{(2)}| \geq \epsilon_0$, and $|N_{\mathcal{F},2}^{(1)}| \leq \epsilon^{1/3}$. So $||N_{\mathcal{F},1} - N_{\mathcal{F},2}||_\infty \geq |N_{\mathcal{F},1}^{(1)} - N_{\mathcal{F},2}^{(1)}| \geq \epsilon_0 - \epsilon^{1/3} > 1$.

By Lemma A.2 and the fact that $\mathcal{F}$ is of width $\leq C$, there exists a $U_s$ such that

$$||U_s||_\infty \geq \frac{||N_{\mathcal{F},1} - N_{\mathcal{F},2}||_\infty}{C} > \frac{1}{C} \text{ and } S_s(R_1) \neq S_s(R_2),$$

where $S_s$ is defined in Definition A.1. We will show that $U_s$ satisfies the condition of the lemma.

There exists a $\lambda_0 \in (0, 1)$ such that $U_s(\lambda_0 q_1 + (1 - \lambda_0)q_2) + b_s = 0$. Let $q_3 = \lambda_0 q_1 + (1 - \lambda_0)q_2$. It is easy to check that:

(v1) $||q_3||_\infty \leq \max_{i=1,2} ||q_i||_\infty \leq \eta$.

(v2) $|q_3^{(j)}| \geq \min_{i=1,2} |q_i^{(j)}| \geq T'\eta$ for $j = 1, 2$.

Since $U_s q_3 + b_s = 0$, $U_s x + b_s = 0$ has solution in $\mathbb{B}_\infty(0, 2)$.

Therefore, it suffices to prove equation 20, We first prove

**(p4).** $\frac{0.5 + 4\epsilon^{1/3}}{0.5 - 5\epsilon^{1/3}}|U_s^{(2)}| < |U_s^{(1)}|$ is not valid.

If it is valid, let $q_t \in \mathbb{R}^n$ be defined as $q_t^{(1)} = q_3^{(1)} + \frac{U_i^{(2)}(-\theta + q_3^{(2)})}{U_s^{(1)}}$, $q_t^{(2)} = \theta$, and $q_t^{(k)} = q_3^{(k)}$ when $k > 2$. It is easy to see that, $U_s q_t + b_i = 0$.

By Lemma A.4, there exist $q_y$ and $v$ such that $||q_y - q_t||_\infty < 0.25\epsilon^{1/3}\theta$ and $||v||_\infty < 0.25\epsilon^{1/3}\theta$. Furthermore, set $p_\lambda = \lambda q_3 + (1 - \lambda)(q_y + v)$ and $q_\lambda = \lambda q_3 + (1 - \lambda)(q_y - v)$ for $\lambda \in [0, 1]$. Let $I_3$ be the set of $\lambda \in [0, 1]$, such that $p_\lambda$ is in the interior of a linear region $R_{1,\lambda}$, $q_\lambda$ is in the interior of a linear region $R_{2,\lambda}$, and $R_{1,\lambda}$ and $R_{2,\lambda}$ are neighboring linear regions with boundary $(U_s, b_s)$. Then $|I_3| \geq 0.5$.

We prove the properties **(p41)** and **(p42)** before proving **(p4)**.

**(p41).** The total $H_2$-length of the segments in $\{q_3 \to q_y + v\}/P_2(\epsilon_0)$ is $\leq \epsilon^{1/3}$. The total $H_2$-length of the segments $\{q_3 \to q_y - v\}/P_2(\epsilon_0)$ is $\leq \epsilon^{1/3}$.

We just prove the result for $q_3 \to q_y + v$, and the proof for $q_3 \to q_y - v$ is the same. Since

$$
\begin{aligned}
&|q_t^{(1)} - q_3^{(1)}| \\
=\ & \left| \frac{U_s^{(2)}(-\theta + q_3^{(2)})}{U_s^{(1)}} \right| \\
\leq\ & \frac{0.5 - 5\epsilon^{1/3}}{0.5 + 4\epsilon^{1/3}} |\theta - q_3^{(2)}| \\
\leq\ & \frac{0.5 - 5\epsilon^{1/3}}{0.5 + 4\epsilon^{1/3}} |\theta - T'\eta| \text{ (by (v2))} \\
<\ & \frac{0.5 - 5\epsilon^{1/3}}{0.5 + 4\epsilon^{1/3}} (1 - (0.5 - 3\epsilon^{1/3})(1 - 2\epsilon^{1/3}))\theta \\
<\ & (0.5 - 5\epsilon^{1/3})\theta
\end{aligned}
$$

and

$$
|q_3^{(2)}| \leq |\eta| \text{ (by (v1))} \leq (0.5 + 3\epsilon^{1/3})\theta,
$$

we have that

$$
\begin{aligned}
&|q_y^{(1)} + v^{(1)} - q_3^{(1)}| \\
\leq\ & |q_t^{(1)} - q_3^{(1)}| + \|v\|_\infty + |q_t^{(1)} - q_y^{(1)}| \\
\leq\ & 0.5\epsilon^{1/3}\theta + (0.5 - 5\epsilon^{1/3})\theta \\
=\ & (0.5 - 4.5\epsilon^{1/3})\theta
\end{aligned}
$$

and

$$
\begin{aligned}
&|q_y^{(2)} + v^{(2)} - q_3^{(2)}| \\
\geq\ & |q_t^2 - q_3^{(2)}| - \|v\|_\infty - |q_t^{(1)} - q_y^{(1)}| \\
\geq\ & (\theta - (0.5 + 3\epsilon^{1/3})\theta) - 0.5\epsilon^{1/3}\theta \\
=\ & -0.5\epsilon^{1/3}\theta + (0.5 - 3\epsilon^{1/3})\theta \\
=\ & (0.5 - 3.5\epsilon^{1/3})\theta,
\end{aligned}
\tag{24}
$$

which implies $|q_y^{(1)} + v^{(1)} - q_3^{(1)}| \leq \frac{0.5 - 4.5\epsilon^{1/3}}{0.5 - 3.5\epsilon^{1/3}} |q_y^{(2)} + v^{(2)} - q_3^{(2)}| \leq (1 - 2\epsilon^{1/3})|q_y^{(2)} + v^{(2)} - q_3^{(2)}|$.

For $j > 2$, we have $|q_y^{(j)} + v^{(j)} - q_3^{(j)}| \leq \|v\|_\infty + \|q_y - q_t\|_\infty + |q_t^{(j)} - q_3^{(j)}| \leq 0.5\epsilon^{1/3}\theta \leq (1 - 2\epsilon^{1/3})(0.5 - 3.5\epsilon^{1/3})\theta \leq (1 - 2\epsilon^{1/3})|q_y^{(2)} + v^{(2)} - q_3^{(2)}|$; that is, for any $j \neq 2$, we have

$$
\frac{|q_y^{(j)} + v^{(j)} - q_3^{(j)}|}{|q_y^{(2)} + v^{(2)} - q_3^{(2)}|} \leq 1 - 2\epsilon^{1/3}.
$$

Consider the polyline segment $x_0 \to q_3 \to q_y + v \to x_1$. The segment $q_3 \to q_y + v$ can be written as $q_3 \to q_3 + (q_y + v - q_3)$, and because $\frac{|q_y^{(j)} + v^{(j)} - q_3^{(j)}|}{|q_y^{(2)} + v^{(2)} - q_3^{(2)}|} \leq 1 - 2\epsilon^{1/3}$ for any $j \neq 2$, $q_y + v - q_3$ can be written as $(q_y^{(2)} + v^{(2)} - q_3^{(2)})(v_1, 1, v_2, \ldots, v_{n-1})^T \in \mathbb{R}^n$, where $v_i = \frac{q_y^{(i)} + v^{(i)} - q_3^{(i)}}{q_y^{(2)} + v^{(2)} - q_3^{(2)}}$ and $|v_i| = |\frac{q_y^{(i)} + v^{(i)} - q_3^{(i)}}{q_y^{(2)} + v^{(2)} - q_3^{(2)}}| < T = 1 - \epsilon^{1/3}$ for $i \in [n-1]$. So similar to Property **(p2)**, we have

$$
\begin{aligned}
2\ =\ & (\mathcal{F}(x_0) - \mathcal{F}(q_3)) + (\mathcal{F}(q_3) - \mathcal{F}(q_y + v)) + (\mathcal{F}(q_y + v) - \mathcal{F}(q_2)) \\
\leq\ & (2 + \epsilon)\|q_3\|_\infty + \int_0^{\|q_y + v - q_3\|_\infty} l_2(q_3 + t(q_y + v - q_3)) \\
& + (1 - \epsilon^{1/3})(2 + \epsilon - l_2(q_3 + t(q_y + v - q_3)))dt \\
& + (1 - \|q_y + v\|_\infty)(2 + \epsilon).
\end{aligned}
$$

Since $\frac{|q_y^{(j)} + v^{(j)} - q_3^{(j)}|}{|q_y^{(2)} + v^{(2)} - q_3^{(2)}|} \leq 1 - 2\epsilon^{1/3} < 1 - \epsilon^{1/3}$ for any $j \neq 2$. Similar to **(p2)**, we find that $\{q_3 \to q_y + v\}/P_2(2 + \epsilon - \epsilon^3)$ has $H_2$-length at most $\epsilon^{1/3}$. For $q_y - v$, we have the same result. This proves **(p41)**.

**(p42).** There exists a $\lambda \in [0,1]$ such that $R_{i,\lambda} \in R_2(\epsilon_0)$ are neighboring linear regions with boundary $(U_s, b_s)$, where $i = 1, 2$.

Let $I_4 \subset [0,1]$ be the set of $\lambda$ such that $R_{1,\lambda}$ and $R_{2,\lambda}$ are in $R_2(\epsilon_0)$ and are neighboring regions with boundary $(U_s, b_s)$. First, we prove (w1) and (w2).

(w1). There exists a set $I_5 \subset [0,1]$ with length at least $1 - \frac{\epsilon^{1/3}}{|q_3^{(2)} - (q_y^{(2)} + v^{(2)})|}$ such that, when $\lambda \in I_5$, $R_{1,\lambda}$ is in $R_2(\epsilon_0)$.

As defined in **(p4)**, $R_{1,\lambda}$ is the linear region containing $p_\lambda = \lambda q_3 + (1-\lambda)(q_y + v) = q_y + v + (-q_y - v + q_3)\lambda$, so $x \in [0,1] \setminus I_5$ if and only if $p_x \in \{q_3 \to q_y + v\} \setminus P_2(\epsilon_0)$. Then we just need to show that: the length of $[0,1] \setminus I_5$ is at most $\frac{\epsilon^{1/3}}{|q_3^{(2)} - (q_y^{(2)} + v^{(2)})|}$, which follows from the fact that the $H_2$-length of set $\{q_3 \to q_y + v\}/P_2(\epsilon_0)$ is at most $\epsilon^{1/3}$, as shown in **(p41)**.

(w2). There exists a set of intervals $I_6 \subset [0,1]$ with length at least $1 - \frac{\epsilon^{1/3}}{|q_3^{(2)} - (q_y^{(2)} - v^{(2)})|}$ such that, when $\lambda \in I_6$, $R_{2,\lambda}$ is in $R_2(\epsilon_0)$, which can be proved similar to (w1).

From the definitions of $I_3, I_4, I_5, I_6$, we have $I_4 = I_3 \cap I_5 \cap I_6 = I_3 \setminus (([0,1] \setminus I_5) \cup ([0,1] \setminus I_6))$. By properties (w1), (w2), Lemma A.4, and equation 24, $|I_4| = |I_3 \setminus (([0,1] \setminus I_5) \cup ([0,1] \setminus I_6))| \geq 0.5 - \frac{\epsilon^{1/3}}{|q_3^{(2)} - q_y^{(2)} - v^{(2)}|} - \frac{\epsilon^{1/3}}{|q_3^{(2)} - q_y^{(2)} + v^{(2)}|} \geq 0.5 - \frac{2\epsilon^{1/3}}{(0.5 - 3.5\epsilon^{1/3})\theta} > 0$. Thus there exists a $\lambda \in I_4$ and **(p42)** is proved.

We now prove **(p4)**. Let $v_{R_i}$ be the normal vector of $\mathcal{F}$ over $R_{i,\lambda}$. Then by property **(p42)**, we know that $R_{i,\lambda}$ are neighboring linear regions with boundary $(U_s, b_s)$, where $i = 1, 2$, so $||v_{R_1} - v_{R_2}||_\infty = ||U_s||_\infty > \frac{1}{C}$. On the other hand, still by **(p42)**, $R_{i,\lambda}$ are in $R_2(\epsilon_0)$, so we have $|v_{R_1}^{(k)} - v_{R_2}^{(k)}| < 2\epsilon^{1/3} < \frac{1}{C}$ when $k \neq 2$ and $|v_{R_1}^{(2)} - v_{R_2}^{(2)}| < (2 + \epsilon) - (\epsilon_0) = \epsilon^{1/3} < \frac{1}{C}$, which means $||v_{R_1} - v_{R_2}||_\infty < \frac{1}{C}$. A contradiction is obtained and **(p4)** is proved.

**(p5).** $\frac{0.5 + 4\epsilon^{1/3}}{0.5 - 5\epsilon^{1/3}}|U_s^{(1)}| \leq |U_s^{(2)}|$ is not valid. This can be proved similar to property **(p4)**.

**(p6).** If $\frac{0.5 + 4\epsilon^{1/3}}{0.5 - 5\epsilon^{1/3}} \geq \frac{|U_s^{(1)}|}{|U_s^{(2)}|} \geq \frac{0.5 - 5\epsilon^{1/3}}{0.5 + 4\epsilon^{1/3}}$, then there exists no $j > 2$ such that $\frac{|U_s^{(j)}|}{\max(|U_s^{(2)}|, |U_s^{(1)}|)} \geq \frac{12\epsilon^{1/3}}{1 - 2\epsilon^3}$.

Assume $|U_s^2| \geq |U_s^1|$ and define a point $q_t$: $q_t^{(1)} = \theta$, $q_t^{(2)} = q_3^{(2)} + (1 - 12\epsilon^{1/3})\frac{U_s^{(1)}(q_3^{(1)} - \theta)}{U_s^{(2)}}$, $q_t^{(j)} = q_3^{(j)} + 12\epsilon^{1/3}\frac{U_s^{(1)}(q_3^{(1)} - \theta)}{U_s^{(j)}}$ for a $j > 2$. Then **(p6)** can be proved similar to **(p4)**.

We now prove equation 20. Since properties **(p4)**, **(p5)**, **(p6)** are always false, it must hold that $\frac{0.5 + 4\epsilon^{1/3}}{0.5 - 5\epsilon^{1/3}} \geq \frac{|U_s^{(1)}|}{|U_s^{(2)}|} \geq \frac{0.5 - 5\epsilon^{1/3}}{0.5 + 4\epsilon^{1/3}}$ and $\frac{|U_s^{(j)}|}{\max(|U_s^{(2)}|, |U_s^{(1)}|)} < \frac{12\epsilon^{1/3}}{1 - 2\epsilon^3}$ for any $j > 2$. Then equation 20 and the lemma are proved. $\square$

