# OpenReview forum: "OPTIMAL ROBUST MEMORIZATION WITH RELU NEURAL NETWORKS"
_ICLR.cc/2024/Conference — ICLR 2024 spotlight_

### Official Review · Reviewer_LzNK · 2023-10-20

**Soundness:** 2 fair
**Presentation:** 3 good
**Contribution:** 2 fair
**Rating:** 5
**Confidence:** 4

**Summary:**

The paper studies the problem of robust memorization, namely exactly fitting the training data while keeping the same prediction in a small neighborhood of each training data. In its first result, it shows that it is NP-hard to compute a robust memorization of the simple network of depth 2 and width 2. It then provides a necessary condition on the width, depth and number of parameters for the existence of robust memorization. It further constructed a neural network which is a robust memorization.

**Strengths:**

It extends the prior results (Li et al 2022) on robust memorization from $\lambda/4$ to any value strictly less than $\lambda/2$.

The number of parameters in the constructed neural network which is a robust memorization does not depend on the separation bound $\lambda_D$ or the robust budget $\mu$.

The paper is well-organized and clearly written.

**Weaknesses:**

1. The term “optimal robust memorization” is inappropriate, and might be misleading and over-claims the significance of the work. Note that robust memorization with radius $< \lambda_D/2$ is not significantly different from that with radius $< \lambda_D/4$, except it is a bit larger neighborhood. This is because $\lambda_D$ is just the minimal distance, not necessarily the distance for every pair of data samples. Hence, even in the case of the so-called “optimal” $\lambda_D/2$-robust memorization, there are still many regions that are not covered by the robust-neighborhoods, and it can be non-robust in those areas. Therefore, $\lambda_D/2$-robust memorization does not really make much difference than the  $\lambda_D/4$-robust memorization. (The word “optimal” only reflects it is the largest radius in the minimal separation based analysis, however, as I mentioned above, it is far from an optimal robust memorization). Therefore, I would consider the contribution of this paper is on enlarging the robust memorization region, which is limited.

2. It seems to me that some results are not consistent. Proposition 4.7 part 2 already infers that depth 2 width 2 network is not a robust memorization. However, Theorem 4.1 claims that it is NP-hard. I hope the authors can clarify on this point.

3. The discussion below Theorem 1.1 is not quite correct. It somehow avoided the case that the absence of memorization implies the absence of robust memorization. Hence, it is not totally “cannot be deduced from each other”.

**Questions:**

I would like to see some intuition on why the number of non-zero parameters does not depend on $\lambda_D$, $L$ and $\mu$. Especially, a comparison with the prior work of Li. et. al. 2022.

---

> ### Author Response · Authors · 2023-11-12
> **Detailed Answer**
>
> Thanks for your valuable comments and questions. We give a detailed answer to them and hope that you could re-evaluate our paper based on these answers.
>
> 1. The term “optimal robust memorization” is inappropriate.
> This comment consists of two major issues.
>
> 1.1 ``Robust memorization with radius $\lambda/2$ is not significantly different from $\lambda/4$, except that it is a bit larger neighborhood''
>
> Answer:
> Please note that the difference between $\lambda/2$ and $\lambda/4$ is not only in the size of the robust neighborhood. Radius $\lambda/2$ is the largest possible robust neighborhood that can be achieved, if using the same radii for all samples, and it is in this sense we call our robust memorization optimal.
>
> 1.2 ``$\lambda/2$ is just the minimal distance, not necessarily the distance for every pair of data samples.''
>
> Answer:
> To the best of our knowledge, all papers that work on adversarial robustness use the same robust radius for all samples. It is a nice idea to use different radii for different samples, but this needs to consider $O(N^2)$ radii for $N$ samples and the number of parameters of the memorization network is expected to be larger.
>
> In summary, {\bf our optimal robust memorization is not indeed absolute optimal, but it is a type of optimal}. To make it more clear, we will change 'optimal' to 'optimal with respect to robust radius'.
>
> 2. It seems to me that some results are not consistent.  Proposition 4.7 part 2 already infers that depth 2 width 2 network is not a robust memorization. However, Theorem 4.1 claims that it is NP-hard.
>
> Answer:
> These two results are not contradictory.
>
> Proposition 4.7 part 2 shows that, {\bf there exists a dataset  D}, all networks with depth 2 and width 2 are not robustness memorizations for D.
>
> Th 4.1 shows that, there is no polynomial-time algorithm to decide whether there exists a robustness memorization with depth 2 and width 2 {\bf for any given dataset D}. Note that there do exist data sets D such that networks of depth 2 and width 2 are robustness memorizations for D.
>
> 3: The discussion below Theorem 1.1 is not quite correct. It somehow avoided the case that the absence of memorization implies the absence of robust memorization.
>
> Answer: We believe that The Discussion is correct. The purpose of this part is to show ``NP-hardness of computing robust memorization for a non-zero budget and the NP-hardness of computing memorization cannot be deduced from each other:''
> The two cases listed subsequently in the paper already serve our purpose.
>
> The reviewer is correct that we did not list the case ``absence of memorization implies the absence of robust memorization''. We will discuss this case in the revised version. But this case does not affect the correctness of our conclusion.
>
> 4: Intuition on why the number of non-zero parameters does not depend on $\lambda,\mu,L$.
>
> Answer: The conclusion of Li2022 is based on some approximate results of the Relu network for polynomials and $1/x$. However, it is obvious that Relu network is locally linear, its growth rate is lower than polynomial or $1/x$ when $x\to \infty$ or $x\to 0$, so this approximation must be within a certain range, and the number of parameters required for approximation is related to this range, so their conclusions need to be based on $\lambda,\mu$.
>
> Our work does not use the classical approximation theorem. We make full use of every parameter of the network and make the meaning of these parameter clear. For example, some parameters are used to measure the distance between the input and the point in the training set, so that $\lambda,\mu$ only affects the value of these parameters, but not the number of parameters;  some parameters are used to predict the label of inputs, so that $L$ only affect the value of these parameters, but not the number of parameters.

---

### Official Review · Reviewer_Sknc · 2023-10-28

**Soundness:** 2 fair
**Presentation:** 1 poor
**Contribution:** 3 good
**Rating:** 5
**Confidence:** 3

**Summary:**

The authors study memorization with neural networks and its connection to deep learning. It emphasizes the significance of "robust memorization," which hasn't been thoroughly explored. The passage mentions the NP-hardness of computing certain network structures for robust memorization and introduces the concept of "optimal robust memorization." It highlights the explicit construction of neural networks with specific parameter counts for optimal memorization. There's also a mention of a lower bound on network width and controlling the Lipschitz constant to achieve robust memorization in binary classification datasets. It is a technical paper addressing these aspects of neural network memorization and generalization. However, it does not provide a clear path for interested readers to understand them.

**Strengths:**

The strengths of the provided passage are its technical depth, problem formulation, explicit solutions, and mention of a lower bound. It delves into the complexities of neural network memorization, introduces a significant problem in deep learning, provides practical solutions, and hints at valuable insights for network design.

**Weaknesses:**

It is highly technical and requires a strong background in readers to grasp the meaning of this paper.

**Questions:**

The authors may think about the organization of this paper. ICLR may not be a suitable conference for this paper.

---

> ### Author Response · Authors · 2023-11-12
> **Learning theory is a topic of ICLR and our paper fits this topic very well**
>
> Thanks for your valuable comments. We give a detailed answer to them and hope that you could re-evaluate our paper based on these answers.
>
> 1. It is highly technical and requires a strong background in readers to grasp the meaning of this paper. The authors may think about the organization of this paper.
>
> Answer:
> Indeed, our paper is  technical and we tried our best to organize the paper to improve readability. For instance, we give informal description of the main results in Introduction Section; give proof sketches for most of the main results; and give remarks/explanations/comparisons after the main results.
> We believe that {\bf the main text of the paper should be understandable without difficulty}. We admit that the proofs in the Appendix are quite technical and tried our best to make the proofs more easily readable, including separate the long proofs into several parts and use figures for illustrations.
>
> 2.  ICLR may not be a suitable conference for this paper.
>
> Answer:
> Note that ``Learning theory'' is one of the official topics of ICLR and our paper fits this topic very well.

---

> ### Author Response · Authors · 2023-11-17
>
> Sorry, I mistakenly posted something here just now. Please ignore it.

---

> > ### Comment · Reviewer_Sknc · 2023-11-21
> > **Reply to authors**
> >
> > Thanks for answering my questions. I read your rebuttal and still keep my score. This paper is technical but may not be suitable for a conference paper.

---

### Official Review · Reviewer_MXq8 · 2023-11-01

**Soundness:** 4 excellent
**Presentation:** 4 excellent
**Contribution:** 3 good
**Rating:** 8
**Confidence:** 3

**Summary:**

The paper studies the complexity and necessary conditions of robust memorization for ReLU networks. Since it is NP-hard to decide whether there exists a small network which is a robust memorization of a given dataset with a robust budget, studying necessary conditions is very important. Two important results are given in the paper. Let n be the input dimension and N be the number of datapoints. First, under a reasonable setting, a network with width smaller than n can not be robust memorization for some dataset and robust budget. Furthermore, there exists a network with width $3n+1$ and depth $2N+1$, and $O(Nn)$ nonzero weights such that a robust memorization is achieved. However, in this case, the values of the parameters can go to infinity when the robust budget is increased. To address this case, the second important result of this paper utilizes a deeper network to guarantee a bounded Lipschitz constant of the network, provided that the underlying classification problems are binary. The depth in this case is increased by a factor of $log(n)$.

**Strengths:**

- Originality: Most existing memorization bounds are derived without optimal robustness. Given that importance of robustness, the proposed necessary conditions for ReLU networks are novel and interesting.

- Quality and clarity: This paper gives a comprehensive presentation on the existence of of ReLU networks that have robust memorization. The background knowledge is well-organized, and the theoretical results are presented in a flow that is easy to follow and understand.

- Significance: The family of ReLU networks is an important architecture and understanding the limitations of memorization is crucial. The new estimate $O(Nn)$ is an improvement over Theorem 2.2 in (Li et at., 2022) in the sense that it achieves stronger robustness with less number of parameters without assuming binary classification.

**Weaknesses:**

- The paper is mainly dedicated to the existence of robust training. No results on optimization or robust generalization are derived. Given that, the scope seems to be quite limited.

- Since overparameterization can often lead to powerful memorization and good generalization performance, the necessary conditions may have stronger implications if they are connected to generalization bounds. It is not clear in the paper that the constructions of ReLU networks for robust memorization would lead to robust generalization. I know the authors acknowledge this in the conclusion, but I think this is a very serious question.

- The main theorems 4.8 and 5.2 only guarantee the existence of optimal robust memorization. These results would be more useful if an optimization or constructive algorithm is given to find the optimal memorization.

**Questions:**

1. The Theorem 2.2 in (Li et al., 2022) is derived for $p>=2$. However, the bound given in Theorem 4.8 is only valid for the infinity norm. The authors may want to point out that in the paper. The bound given by Theorem B.3 seems to be a bit worse than the bound given by Theorem 2.2 in (Li et al., 2022). I think it would be also helpful to compare such a case in the main text. What are the main difficulties for deriving bounds under $p$-norm?

2. Given that the existence of optimal robust memorization is guaranteed under a ReLU network with bounded size, would it be possible to arrive at such a solution using any optimization algorithm? What would be the complexity of such an algorithm?

---

> ### Author Response · Authors · 2023-11-12
> **Detailed Answer**
>
> Thanks for your valuable comments and questions. We give a detailed answer and hope that we have addressed your concerns. The review is focused on four issues and we will answer them separately.
>
> 1. The main theorems 4.8 and 5.2 only guarantee the existence of optimal robust memorization. These results would be more useful if an optimization or constructive algorithm is given to find the optimal memorization.
>
> Answer:
> We did give constructive algorithms for the optimal robust memorization networks.
> Please see Appendix B.5 on page 28 (proof for Theorem 4.8) and Appendix C.2 on page 33 (proof for Theorem 5.2). We will make this clear in the revised version of the paper.
>
> 2. Given that the existence of optimal robust memorization is guaranteed under a ReLU network with bounded size, would it be possible to arrive at such a solution using any optimization algorithm? What would be the complexity of such an algorithm?
>
> Answer: As said in 1, we give direct constructive algorithms for the optimal robust memorization networks, and these are polynomial-time algorithms.
>
> To the best of our knowledge, all works that can give memorization with polynomial number of parameters are direct construction of the networks from the data sets, and no one used optimization methods such as SGD. This is because using SGD cannot in general to give theoretically guaranteed memorization networks.
>
> 3. Robust generalization. It is not clear in the paper that the constructions of ReLU networks for robust memorization would lead to robust generalization. I know the authors acknowledge this in the conclusion, but I think this is a very serious question.
>
> Answer:
>
> 3.1. Having  memorization neural networks with theoretically guaranteed robust generalization ability is one of the most important problems in adversarial learning, and worth studying.
> To the best of our knowledge, this is still an open problem, and all existing works on memorization do not have theoretical results on generalization.
>
> 3.2 Indeed, this work is primarily focused on the expressive power of neural networks, and no theoretical result on generalization is given.
> In our ongoing work, we link generalization to memorization. Under the assumption that the samples in the dataset are i.i.d. selected from the data distribution, we give generalization analysis of memorization networks. Moreover, for data distributions that satisfy certain assumptions, our Theorems 4.8 and 5.2 also guarantee generalization if the sample in dataset is sufficient.
>
> 4.  The Theorem 2.2 in (Li et al., 2022) is derived for  $p\ge2$ (actually it is $p\in\\{2, \infty\\}$). However, the bound given in Theorem 4.8 is only valid for the infinity norm. The authors may want to point out that in the paper. The bound given by Theorem B.3 seems to be a bit worse than the bound given by Theorem 2.2 in (Li et al., 2022). I think it would be also helpful to compare such a case in the main text. What are the main difficulties for deriving bounds under p-norm?
>
> Answer:  We are grateful that you read our Appendix. We answer this question in three parts.
>
> 4.1 It is not difficult to give an upper bound of robust memorization under the $L_p$ norm. Theorem B3 shows that the required number of parameters in relation to $p$ is no more that $O(Nnp^2\log(n/\gamma))$. For the case of $p=1$, Theorem B2 gives a good bound similar to Theorem 1.1. But it is desirable and difficult to have a bound which is not essentially dependent on p.
>
> 4.2 To come up with that in Li2022. When $p=2$ and $\gamma=\lambda_D^p/4$, our bound in Theorem B.3 becomes $O(Nn\log(n/\lambda_D^p))$, but the result in Li2022 is $O(Nn\log(n/\lambda_D^p)+O(Npoly\log(N/\lambda_D^p)))$. Our result is better than that of Li2022.
>
> 4.3 We focus mainly on the $L_\infty$ norm in this work, so the results on other norms are not mentioned in the main text. We will add these comparisons in the revised version.

---

> > ### Comment · Reviewer_MXq8 · 2023-11-17
> >
> > I would like to thank the authors for their detailed answers. Most of my concerns have been addressed.
> >
> > Regarding your claim "all existing works on memorization do not have theoretical results on generalization."
> >
> > 1. Could you further clarify or provide some references to support your claim?
> > 2. By avoiding memorizing training data, would it be possible to achieve higher generalization performance?
> > 3. Is the usual setting used to prove better generalization not in favor of memorization?

---

> > > ### Author Response · Authors · 2023-11-17
> > >
> > > We want to thank the reviewer again for these questions, which help us think more deeply on the relation between memorization and generalization.
> > >
> > > Question 1. Could you further clarify or provide some references to support your claim: "all existing works on memorization do not have theoretical results on generalization"?
> > >
> > > We answer this question from two perspectives.
> > >
> > > 1.1 Existing papers on memorization focus mainly on the number of parameters needed for memorization.
> > > For example, paper [1] gave the first neural network of memorization with sublinear parameters and [2] gave the optimal number of parameters for memorization networks. In these papers (and other similar papers), the memorization network is for "any dataset", and it was NOT assumed that the dataset is drwan i.i.d from the data distribution. As a consequence, it is not possible to give theoretical generalization guaranteed theory such as generalization bound, since such an assumption is a necessary condition in developing a generalization bound.
> > >
> > > 1.2 On the other hand, there exist works to study the relationship between generalization and memorization networks, mostly from experimental aspects.
> > > Paper [3] pointed out that memorization helps generalization on complex learning tasks, because data with the same label have quite diversified features and need to be nearly memorized.
> > > Paper [4] showed that, when the networks reach the interpolation threshold, larger networks tend to have more generalization ability.
> > >
> > > In [5], the numbers of parameters required for a well-robust-generalized memorization network was given, but the result is not in the form of the traditionally generalization bound theory that deduces accuracy on the whole data distribution from that on the training set.
> > >
> > > Question 2. By avoiding memorizing training data, would it be possible to achieve higher generalization performance?
> > >
> > > We answer the question from two perspectives.
> > >
> > > 2.1 There exist vast literature to achieve higher generalization performance practically. For instance, by early termination, dropout, adding regulations, etc. These methods are not directly related with memorization. One of the main idea to achieve higher generalization performance is make the function of the network more "smooth".
> > >
> > > 2.2 In most theories on generalization bound that guarantees generalization or practical training of networks, very small empirical errors are required or achieved, that is, the network should performing well on the training set. Very small empirical error is almost the same as that of memorization when the MSE loss is used. In this meaning, very small empirical training error or (approximately) memorization is certain necessary conditions for better generalization.
> > >
> > >
> > > Question 3. Is the usual setting used to prove better generalization not in favor of memorization?
> > >
> > > No, most assumptions are in favor of memorization.
> > > As we already said in 2.2, the generalization bound is to bound the difference between the population loss over the data distribution and the empirical loss over the training set. In other words, very small empirical error is certain necessary condition to guarantee generalization, and very small empirical error is almost the same as memorization when the MSE loss is used.
> > >
> > > Another major difference between the generalization bound theory and memorization is that "dataset is iid selected from distribution" is assumed in generalization bound. On the other hand, memorization works for any data distribution and hence also works for iid selected data.
> > >
> > > [1] Park S, Lee J, Yun C, et al. Provable memorization via deep neural networks using sub-linear parameters. 2021.
> > >
> > > [2] Vardi G, Yehudai G, Shamir O. On the optimal memorization power of relu neural networks. 2021.
> > >
> > > [3] Feldman V, Zhang C. What neural networks memorize and why: Discovering the long tail via influence estimation. 2020.
> > >
> > > [4] Khandelwal U, Levy O, Jurafsky D, et al. Generalization through memorization: Nearest neighbor language models. 2019.
> > >
> > > [5] Li B, Jin J, Zhong H, Hopcroft JE,  Wang L. Why robust generalization in deep learning is difficult: Perspective of expressive power. 2022.

---

> > > > ### Comment · Reviewer_MXq8 · 2023-11-22
> > > >
> > > > I extend my sincere thanks to the authors for their comprehensive responses. I appreciate the thorough addressing of my concerns, and I am now confident in affirming that this paper makes valuable contributions to the field of memorization.

---

> > > > > ### Author Response · Authors · 2023-11-22
> > > > >
> > > > > Thank you very much for all the insightful questions!

---

### Meta-Review · Area_Chair_cXCp · 2023-12-12

**Metareview:**

An existing literature in the theory of deep learning focuses on the phenomenon of memorization. Insofar as neural networks can fit the entirety of the training data, this literature characterizes how many parameters are required to accomplish the feat for a neural network of some given architecture as a function of the data dimension, number of examples etc. This paper extends the literature by asking about the sample complexity of robust "memorization" where the goal is not only to fit all of the training examples but all perturbations of those same examples according to some noise model. The paper did not receive the depth of peer review and discussion that it deserved. While the paper received three reviews with scores 8, 5, 5, the review from Rev MXq8 who awarded the 8 was far the most informative and most engaged. The paper was lauded for its originality, clarity, the strength of the demonstrated theoretical result, with a note that the O(nN) result was ("an improvement over Theorem 2.2 in (Li et at., 2022) in the sense that it achieves stronger robustness with less number of parameters"). The reviewer noted that one hope for this line of literature would be for it to tackle not only the robustness of memorization but generalization. At a bird's eye view, I wonder if this sort of robustness offers any sort of ladder towards making statements about generalization more broadly.

**Justification For Why Not Higher Score:**

Not enough to go on.

**Justification For Why Not Lower Score:**

The result appears to be a non-trivial advance in an established line of deep learning theory.

---

### Decision · Program_Chairs · 2024-01-16

Accept (spotlight)